# Genetic studies of paired metabolomes reveal enzymatic and transport processes at the interface of plasma and urine

Pascal Schlosser [1,2,20] ✉, Nora Scherer [1,3,20],
Franziska Grundner-Culemann [1], Sara Monteiro-Martins[1], Stefan Haug[1],
Inga Steinbrenner [1], Burulça Uluvar[1], Matthias Wuttke [1], Yurong Cheng[1],
Arif B. Ekici [4], Gergely Gyimesi[5], Edward D. Karoly[6], Fruzsina Kotsis[1,7],
Johanna Mielke[8], Maria F. Gomez [9], Bing Yu [10], Morgan E. Grams[11],
Josef Coresh [2], Eric Boerwinkle[10,12], Michael Köttgen [7,13],
Florian Kronenberg [14], Heike Meiselbach [15], Robert P. Mohney [6],
Shreeram Akilesh [16], GCKD Investigators*, Miriam Schmidts [13,17],
Matthias A. Hediger[5], Ulla T. Schultheiss[1,7], Kai-Uwe Eckardt [15,18],
Peter J. Oefner[19], Peggy Sekula [1], Yong Li [1] & Anna Köttgen [1,2,13] ✉

The kidneys operate at the interface of plasma and urine by clearing molecular waste products while retaining valuable solutes. Genetic studies of paired plasma and urine metabolomes may identify underlying processes. We conducted genome-wide studies of 1,916 plasma and urine metabolites and detected 1,299 significant associations. Associations with 40% of implicated metabolites would have been missed by studying plasma alone. We detected urine-specific findings that provide information about metabolite reabsorption in the kidney, such as aquaporin (AQP)-7-mediated glycerol transport, and different metabolomic footprints of kidney-expressed proteins in plasma and urine that are consistent with their localization and function, including the transporters NaDC3 (*SLC13A3*) and ASBT (*SLC10A2*). Shared genetic determinants of 7,073 metabolite–disease combinations represent a resource to better understand metabolic diseases and revealed connections of dipeptidase 1 with circulating digestive enzymes and with hypertension. Extending genetic studies of the metabolome beyond plasma yields unique insights into processes at the interface of body compartments.

The human kidney clears small molecular waste products from plasma while retaining valuable solutes such as amino acids to maintain metabolic homeostasis. After glomerular filtration of plasma to primary urine ultrafiltrate, its composition is modified in a highly coordinated process along the nephron. Hundreds of highly specialized transport proteins move solutes across the membranes of the cells lining the nephron to reabsorb important molecules while actively excreting toxic or unnecessary ones[1]. Many of these transport proteins as well as the enzymes responsible for generating or breaking down the transported metabolites have been identified through the study of human monogenic diseases. They represent attractive drug targets not only to treat kidney diseases but also metabolic diseases, as exemplified

A full list of affiliations appears at the end of the paper. e-mail: pascal.schlosser@uniklinik-freiburg.de; anna.koettgen@uniklinik-freiburg.de

**Fig. 1 | Overview of the study design.** Schematic representation of the genome-wide screens for plasma and urine metabolite levels and their follow-up analyses. Analyses based on data from plasma are presented in red, analyses based on data from urine in blue, comparative analyses of results based on plasma and urine are shown in a red–blue gradient, and matrix-independent analyses are in white. Icon credit, Servier Medical Art by Servier (licensed under a Creative Commons Attribution 3.0 Unported License). HMGD, Human Gene Mutation Database; HPA, Human Protein Atlas; HRC, Haplotype Reference Consortium; v8, version 8.

by inhibitors of the transporters SGLT2 and URAT1 (refs. 2,3). However, many transporters and enzymes, as well as their substrates and products in vivo remain to be characterized. We hypothesized that linking information from human genetic studies to plasma and urine metabolomes would provide new insights into the roles of these proteins in health and disease.

Genetic effects on metabolite levels in urine can reflect systemic processes such as genotype-dependent intestinal metabolite uptake or hepatic transformation reactions that are detected in urine because of the respective metabolites' filtration from plasma. They can also reflect kidney-specific processes, for example, the active production, reuptake or secretion of small molecules by the cells lining the nephron. Studies with paired plasma and urine metabolite measurements have the potential to distinguish between these processes.

Here, we study differences and similarities regarding genetic influences on metabolomes derived from two 'matrices', plasma and urine, to test the hypothesis that both provide complementary information. Through systematic integration of genome-wide genetic information with paired plasma and urine metabolite measurements from 5,023 participants in the German Chronic Kidney Disease (GCKD) study, we highlight underlying systemic as well as kidney-specific processes. We detect 1,299 genome-wide significant associations and show that

studying plasma alone would have missed associations with almost 40% of metabolites. We highlight examples of urine-specific associations, of footprints that kidney-expressed transporters leave in plasma and urine metabolomes and of previously undescribed systemic roles of a kidney-enriched enzyme. This study generates a rich resource for future experimental validation of yet uncharacterized enzymatic and transport processes that may represent a molecular link between genetic variants and human traits and diseases.

## Results

We performed genome-wide screens for genetic variants significantly associated with levels of 1,296 plasma and 1,399 urine metabolites (779 overlapping metabolites; Fig. 1). Metabolites were quantified by non-targeted mass spectrometry[4] in plasma and urine specimens from 5,023 participants in the GCKD study (Methods and Supplementary Tables 1 and 2).

### mGWAS identify 1,299 signals for 760 metabolites

Genome-wide association studies (GWAS) of plasma metabolite levels (mGWAS) yielded 677 regions that contained at least one significantly associated SNP ($P$ value $< 3.9 \times 10^{-11}$; Fig. 1 and Supplementary Table 3). For each metabolite and region, the SNP with the lowest association

*P* value was chosen as the index SNP, termed metabolite quantitative trait locus (mQTL; regional association plots in Supplementary Data 1). While we have previously shown that genetic effects on the urine metabolome are of comparable magnitude in persons with and without reduced kidney function[5], we now used data from the independent, population-based Atherosclerosis Risk in Communities (ARIC) study (Supplementary Table 1 and Supplementary Methods) to show that this also holds true for plasma mQTLs, as detailed in the Supplementary Results, Supplementary Table 4 and Extended Data Fig. 1.

We next compared our findings with those from seven large genetic studies of the plasma or serum metabolome[6–12] (Methods). We observed excellent correlations of genetic effects and high validation rates of published mQTLs with results from our study (Supplementary Table 5) and conversely of our plasma mQTLs with the results of the published studies (Supplementary Table 6 and Supplementary Fig. 1; details in the Supplementary Results). Not surprisingly, the majority (92.6%) of plasma mQTLs from our study were already reported in at least one of these up-to-17-fold-larger studies. There were, however, 50 mQTLs not reported as significant in any of these published studies, with 20 of them arising from previously unreported metabolites. Together, these comparisons underscored the validity and generalizability of our findings in plasma.

Across results from 1,399 GWAS of urine metabolite levels, we identified 622 mQTLs (*P* value < $3.6 \times 10^{-11}$; Fig. 1, Supplementary Table 3 and Supplementary Data 2). In comparison to our previous study of the urine metabolome[5], 64% of the now detected mQTLs (399 of 622) were not reported before, and the number of unique metabolites with at least one urine mQTL more than doubled. Investigation of the detected urine mQTL in the seven mGWAS of the circulating metabolome[6–12] underscored the additional discovery potential of urine: 56.6% (352 of 622) of urine mQTL were not significant in any of these studies, with 212 of these mQTL arising from urine metabolites not reported in the plasma or serum metabolomes (Supplementary Table 6). Comparisons of the urine mQTLs to their associations with levels of the respective circulating metabolites from both the GCKD as well as the seven published mGWAS and vice versa are detailed in the Supplementary Results, Supplementary Fig. 2 and Supplementary Table 5 and contain interesting examples of inversely correlated genetic effects that are consistent with the localization of the encoded proteins and enzymes at the apical membrane of kidney tubular epithelial cells.

Across both matrices, we identified 1,299 mQTLs from the results from 2,697 GWAS (Supplementary Table 3 and Fig. 2), 37 of which showed interaction with sex ($P_{interaction} < 3.8 \times 10^{-5}$; Supplementary Table 7) and are summarized in the Supplementary Results. Statistical fine mapping enabled prioritization of the most likely causal variants at each mQTL (Methods). Of 1,509 independent signals (Supplementary Table 8), 396 (26%) were fine mapped to credible sets of two to five SNPs, and 192 (13%) were mapped even to a single SNP, including 53 missense, one splice and one stop-lost variant (Supplementary Table 9). Smaller credible set size was significantly associated with lower minor allele frequency (MAF) of the independent index SNPs ($P = 2.3 \times 10^{-13}$) but not with the number of associated metabolites ($P > 0.8$). In summary, discovery GWAS of the plasma and urine metabolomes identified a wealth of significantly associated loci, the basis for subsequent characterizations.

## Differences in plasma and urine mQTL

The 1,299 mQTLs arose from 760 unique metabolites, of which 301 (40%) only showed genetic associations with their levels in plasma, 275 (36%) only showed associations with their levels in urine and 184 (24%) showed associations with their levels in both matrices (Supplementary Table 3). Estimated genome-wide heritability was similar for most matched urine and plasma metabolites (Extended Data Fig. 2). There were 41% (213 of 517) plasma-specific, 30% (183 of 620) urine-specific and 47% (364 of 779) shared metabolites with an mQTL (Fig. 3a). Among

the 364 shared metabolites with an mQTL, 49% (180) exclusively showed a significant genetic association in plasma (88 metabolites) or in urine (92 metabolites; Fig. 3a).

Whereas plasma mQTLs more likely arose from lipid superpathway metabolites than urine mQTLs (301 versus 97 metabolites), consistent with the lack of glomerular filtration of many lipids, urine mQTLs were more likely connected to nucleotide, peptide or unnamed metabolites (Fig. 3b). The power to detect significant associations for almost all metabolite superpathways was similar for plasma and urine (Extended Data Fig. 3). The variance in metabolite levels explained by plasma mQTLs ranged from 0.18% to 50.9% (median 1.3%) and by urine mQTLs ranged from 0.55% to 61.4% (median 2.0%; Supplementary Table 3).

## Plasma and urine mQTLs highlight distinct major genes

Pairwise colocalization testing between metabolite association signals at the same locus to detect shared genetic associations likely to arise from the same underlying causal variant identified 10,596 positive colocalizations (posterior probability for a shared causal variant (PP $H_4$) > 0.8; Methods) involving 1,162 mQTLs. Colocalizing associations were divided into four groups (Supplementary Table 10): those where the same genetic signal affected different metabolites in the same matrix ((1) 'intraplasma', *n* = 3,189; (2) 'intraurine', *n* = 3,155), the same metabolite in both plasma and urine ((3) 'intermatrix, same metabolite', *n* = 204) and different metabolites in plasma and urine ((4) 'intermatrix, different metabolite', *n* = 4,048).

We next asked whether there were central genes shaping the matrix-specific metabolome by assessing major differences between the genes underlying positive intramatrix colocalizations in plasma and urine. Combination of multiple complementary sources of evidence at each mQTL enabled prioritization of 282 most likely underlying genes[5,13,14] (Methods and Supplementary Table 11), of which the majority encode enzymes (*n* = 211, 75%), followed by transport proteins[15,16] (Fig. 2 and Supplementary Table 12). Whereas *FADS1* and *SLCO1B1* accounted for nearly half of the 3,189 intraplasma colocalizations, *NAT8* and the solute carrier (SLC)17A genes (mostly *SLC17A1*) made up >50% of the 3,155 intraurine colocalizations (Fig. 3c). This is consistent with *FADS1* encoding a central enzyme in polyunsaturated fatty acid metabolism[17] and the predominance of these lipid metabolites in plasma and with *NAT8* encoding an *N*-acetyltransferase highly expressed in the kidney that generates water-soluble molecules for excretion[18] and the abundance of *N*-acetylated metabolites in urine. Similarly, the organic anion transporter encoded by *SLCO1B1* and the solute transporters encoded by the SLC17A family show high and specific expression in liver and kidney, respectively, where they transport dozens of physiological and pharmacological substrates[19,20].

The direction and strength of association of almost all 204 'intermatrix, same metabolite' mQTLs was nearly identical in plasma and urine (Extended Data Fig. 4), consistent with these metabolites' filtration from plasma to urine. Observed differences in explained metabolite variance as well as effect direction are detailed in the Supplementary Results.

## mQTL share genetic associations with biomarkers and diseases

Pairwise colocalization analysis of mQTL summary statistics with those of 2,942 unique clinical biomarkers and diseases from the UK Biobank (Methods) identified 7,073 positive colocalizations (Supplementary Table 13). The corresponding metabolites may represent a molecular link between genetic variants and clinical endpoints, as detailed for genetic variants at the *CYP3A7* locus that colocalized with plasma androsterone sulfate levels and hypertension in the Supplementary Results.

With respect to kidney diseases, evidence for a shared genetic signal was detected between metabolite associations at *GSTM1* and kidney cancer[21]; *FMO4* and hypertensive chronic kidney disease (CKD);

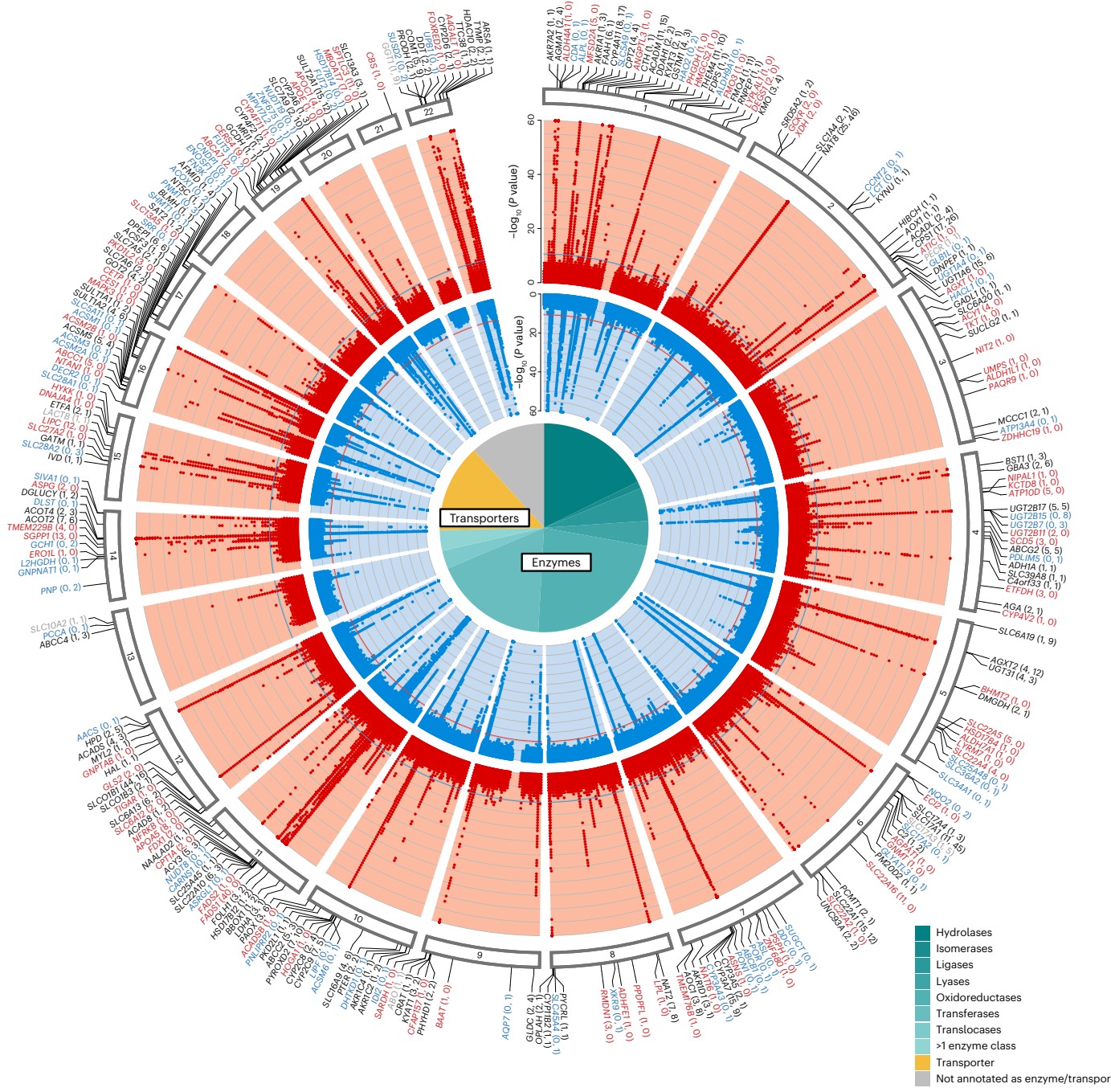

**Fig. 2 | Circular presentation of the 1,299 identified genetic associations with metabolite levels in plasma and urine.** The light red band shows the $-\log_{10}(P\text{ values})$ for genetic associations with metabolite levels in plasma by chromosomal position, and the light blue band shows the associations with metabolite levels in urine. Results from all 1,296 plasma and 1,401 urine GWAS traits are based on linear regressions and are overlaid in the respective bands, with $P$ values truncated at $1 \times 10^{-60}$. The horizontal lines (blue and red) indicate genome-wide significance ($P_{plasma} = 3.9 \times 10^{-11}$ and $P_{urine} = 3.6 \times 10^{-11}$). Supplementary Table 3 contains details about index SNP associations (mQTLs). Gene labels for significant loci were assigned based on mQTL annotations, colocalization analysis with gene expression and protein levels, and literature research (Methods). Black gene labels indicate genetic regions identified in both plasma and urine with intermatrix colocalization (PP $H_4 > 0.8$), gray labels indicate genetic regions identified in both plasma and urine without intermatrix colocalization, and red or blue labels indicate genetic regions exclusively identified in plasma or urine, respectively. The number of plasma and urine mQTLs annotated to a gene is given in parentheses (plasma, urine). The pie chart reflects the proportions of the 282 unique genes that were annotated as enzymes and transporters. Official gene symbols for PYCRL and ERO1L are PYCR3 and ERO1A, respectively.

*ALPL*, *CYP2D6* and *SLC34A1* as well as *ABCG2* and kidney stones; and *ABCC4* and urine retention.

Many colocalizations were detected with continuous markers of kidney (14.8%) and liver function (7.7%). Creatinine-based estimated glomerular filtration rate (eGFR) (eGFRcrea, 127 colocalizations) and

alanine aminotransferase (ALT, 86 colocalizations) as exemplary kidney and liver function markers often colocalized with metabolite levels in the expected matrix (Fig. 4a and Extended Data Fig. 5): for example, loci containing lipid metabolism-related genes such as *FADS1* or *LIPC* showed evidence of shared genetic architecture between plasma lipid

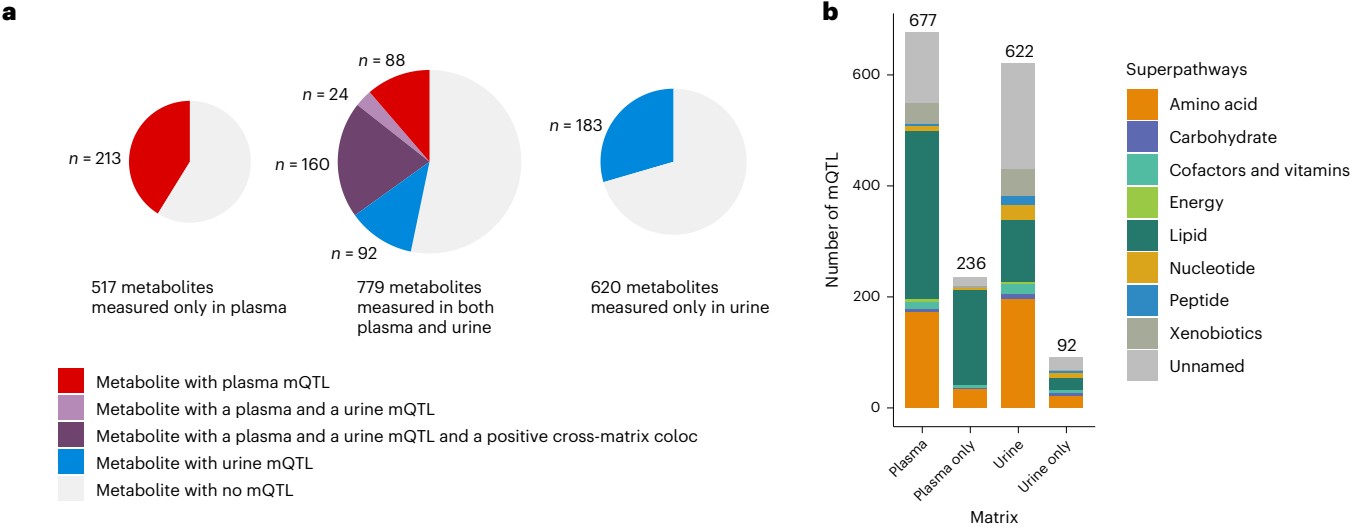

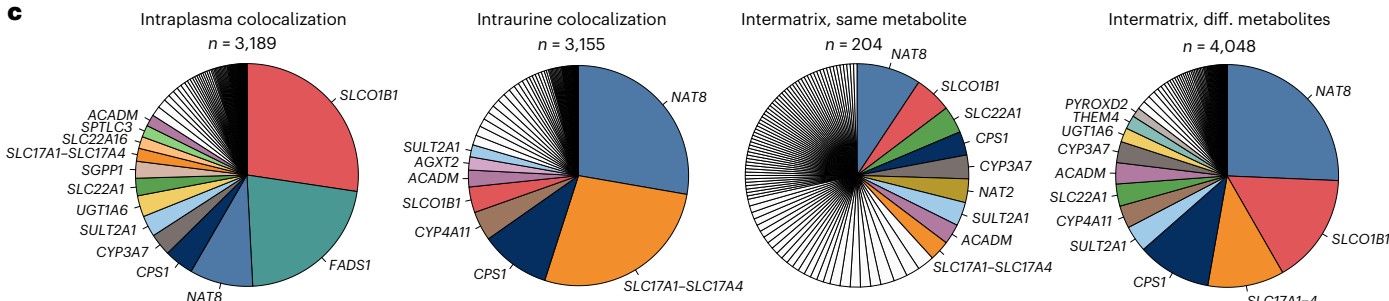

**Fig. 3 | Comparative analyses of mQTLs from plasma and urine. a**, Proportions and counts of metabolites with mQTL split by measurement matrix. **b**, mQTL annotation by metabolite superpathway for mQTLs identified in plasma, in urine and in plasma and urine only. mQTLs identified in plasma contained a higher proportion of lipids, whereas mQTLs identified in urine contained a higher proportion of unnamed molecules, peptides and nucleotides. **c**, Major genes underlying colocalizing association signals of metabolites in four different groups: intraplasma colocalizations (551 distinct mQTLs involved), intraurine colocalizations (497 distinct mQTLs involved), intermatrix colocalizations with the same metabolite (408 distinct mQTLs involved) and intermatrix colocalizations with different (diff.) metabolites (837 distinct mQTLs involved, distinguishing between mQTL in plasma and urine). For the smallest 'intermatrix, same metabolite' group, all genes assigned to >5 colocalizing regions are color coded and labeled. For the three other groups, all genes assigned to >50 colocalizing metabolite regions are color coded and labeled.

metabolite levels and liver but not kidney function. Likewise, several loci encoding transporters with important roles in the kidney such as *SLC34A1* or *SLC7A9* exhibited shared genetic architecture between urine levels of associated metabolites and kidney function. At the majority of loci, however, several mQTLs colocalized with kidney or liver function markers, some of which were detected from urine and some from plasma. These observations further emphasize the value of studying paired matrices. Metabolites most strongly connected to kidney function by correlation analyses and genetic evidence as well as Mendelian randomization studies are summarized in the Supplementary Results, Supplementary Table 14 and Extended Data Fig. 6.

We also searched for gene-level associations of the 282 prioritized genes (Supplementary Table 15) and for variant-level associations of the identified mQTL (Supplementary Table 16) with several thousands of phenotypes based on whole-exome-sequencing data from ~450,000 UK Biobank participants[22] (Methods). At the gene level, putative damaging rare variants in 28 genes were associated with at least one of 437 phenotypes at $P < 2 \times 10^{-9}$ (Supplementary Table 15 and the Methods). We observed both trait- or risk-increasing and -decreasing associations upon the genes' assumed loss of function (Fig. 4b), highlighting opportunities in which therapeutic target inhibition confers protection as exemplified by *ANGPTL3* and dyslipidemia, for which new drugs have recently gained approval[23]. While in this example plasma lipid levels can serve as the required intermediate biomarker for the clinical development pipeline, such biomarkers may be elusive for other potential targets and contained among results of this study. At the variant level, 14 coding variants were associated with genitourinary traits at $P < 1 \times 10^{-5}$ (Supplementary Table 16), including experimentally confirmed positive controls such as p.Gln141Lys in the transporter ABCG2 and serum urate[24].

**Tissue, pathway and murine phenotype enrichment**

Over-representation analyses of the 282 prioritized genes revealed a large number of significant gene ontology (GO) terms and Kyoto Encyclopedia of Genes and Genomes (KEGG) pathways (Supplementary Table 17), human tissues and cell types (Supplementary Tables 18 and 19), especially in kidney and liver, as well as metabolic homeostasis-related phenotypes in genetically manipulated mice (Supplementary Table 20 and Fig. 4c). The Supplementary Results contain details, including a focus on matrix-specific mQTLs (Extended Data Fig. 7).

**mQTLs from urine-specific metabolites: the *FUT2* locus**

mQTLs arising from the 187 urine-specific metabolites may highlight kidney-specific processes or systemic processes only detected in urine (Fig. 5a). Regarding kidney-specific processes, there were multiple

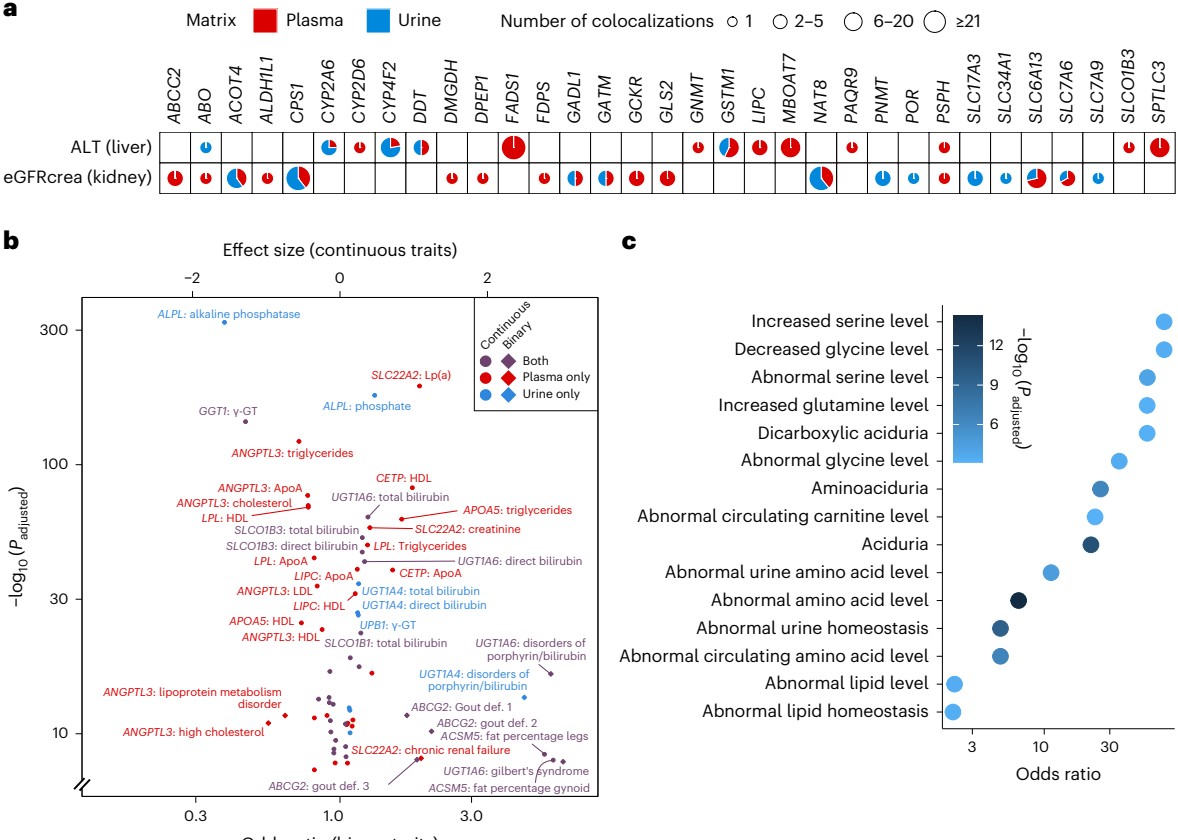

**Fig. 4 | Association of mQTLs and implicated genes with clinical biomarkers, diseases and phenotypes in genetically manipulated mice. a**, Colocalization of mQTLs with clinical markers of kidney (eGFR) and liver (ALT) function. mQTLs are represented by the implicated genes on the *x* axis. The size of the pie represents the total number of colocalizations grouped into four categories. The slices in each pie colored in red and blue represent the proportion of colocalizations of plasma and urine mQTLs with the respective markers. **b**, Effect size (continuous traits) and odds ratio (binary traits) estimates from tests comparing carriers and noncarriers of rare, presumed deleterious

mutations in a gene (Supplementary Table 15; gene-based testing; Methods). The matrix of the origin of mQTL-implicated genes is color coded (red, plasma; blue, urine; purple, both). ApoA, apolipoprotein A; γ-GT, gamma glutamyltransferase; HDL, high-density lipoprotein; LDL, low-density lipoprotein; Lp(a), lipoprotein A. **c**, Over-representation of the 282 genes identified in the mGWAS among phenotypes arising from genetically manipulated mice as part of the Mouse Genome Informatics resource. Only the 15 significant terms with the lowest *P* values are shown (Fisher's exact test); a full list is found in Supplementary Table 20.

examples of associations between variants in genes encoding transporters at the apical membrane of tubular cells with the levels of metabolites that they reabsorb from the urine ultrafiltrate (Supplementary Results). An example in which urine-specific metabolites serve as a readout of systemic processes were two mQTLs for galactosylglycerol and 1,6-anhydroglucose at *FUT2*. Both index SNPs are in high linkage disequilibrium (LD; $r^2 > 0.8$) with rs601338, at which the minor A allele encodes the stop-gain variant p.Trp154Ter (NP_000502.4) that was associated with higher levels of only these two urine metabolites. The encoded fucosyltransferase 2 is a ubiquitously expressed enzyme that mediates the inclusion of fucose into glycans on a variety of glycolipids and glycoproteins. Individuals homozygous for p.Trp154Ter have lower risk of several infectious diseases during childhood[25,26], a selective advantage. Indeed, we detected positive selection at this and other loci, including positive controls such as the *LCT* locus (Methods and Supplementary Table 21). The extended homozygosity of the haplotype carrying the minor, derived allele at the galactosylglycerol mQTL further supported positive selection (Fig. 5b).

Exploration of phenome-wide associations for fucosyltransferase 2 (FUT2) p.Trp154Ter in the UK Biobank (Methods) showed significant associations with dyslipidemia, hypertension and cholelithiasis (Fig. 5c). Colocalization confirmed a shared genetic basis of the two mQTL with these diseases, as well as of several plasma proteins

(Supplementary Tables 13 and 22). These observations suggest that higher urine levels of galactosylglycerol and 1,6-anhydroglucose could reflect increased risk for *FUT2* genotype-related cardiometabolic diseases of adult onset, motivating future studies.

**Urine-specific mQTLs from shared metabolites: the *AQP7* locus**

An interesting example of a urine-specific mQTLs arising from a matrix-shared metabolite (Fig. 5d) was detected at *AQP7* with urine glycerol levels. The signal (Fig. 5e, $P_{urine} = 9.93 \times 10^{-58}$; $P_{plasma} = 0.53$) was fine mapped to rs62542743, encoding the missense variant AQP-7 p.Gly264Val (NP_001161.1). The channel AQP-7 mainly transports water and glycerol and, in the rat and mouse kidney, localizes to proximal straight tubules[27]. *Aqp7*-knockout mice show glycerol loss in urine, supporting a role in glycerol reabsorption[28]. The minor A allele (p.264Val) was associated with higher urine glycerol levels, which is in agreement with the knockout mouse findings when assuming loss of function (Fig. 5f). The mutant valine carries two more methyl residues than wild-type glycine, which may decrease the channels' passing ability for glycerol. Moreover, a previous case report exists of three children homozygous for p.264Val who presented with normoglycerolemic hyperglyceroluria[29]. Indeed, we confirmed a recessive effect, with persons homozygous for the A allele showing >64-fold higher urine but not plasma glycerol levels (Fig. 5g), thereby confirming a single case report through evidence from population studies.

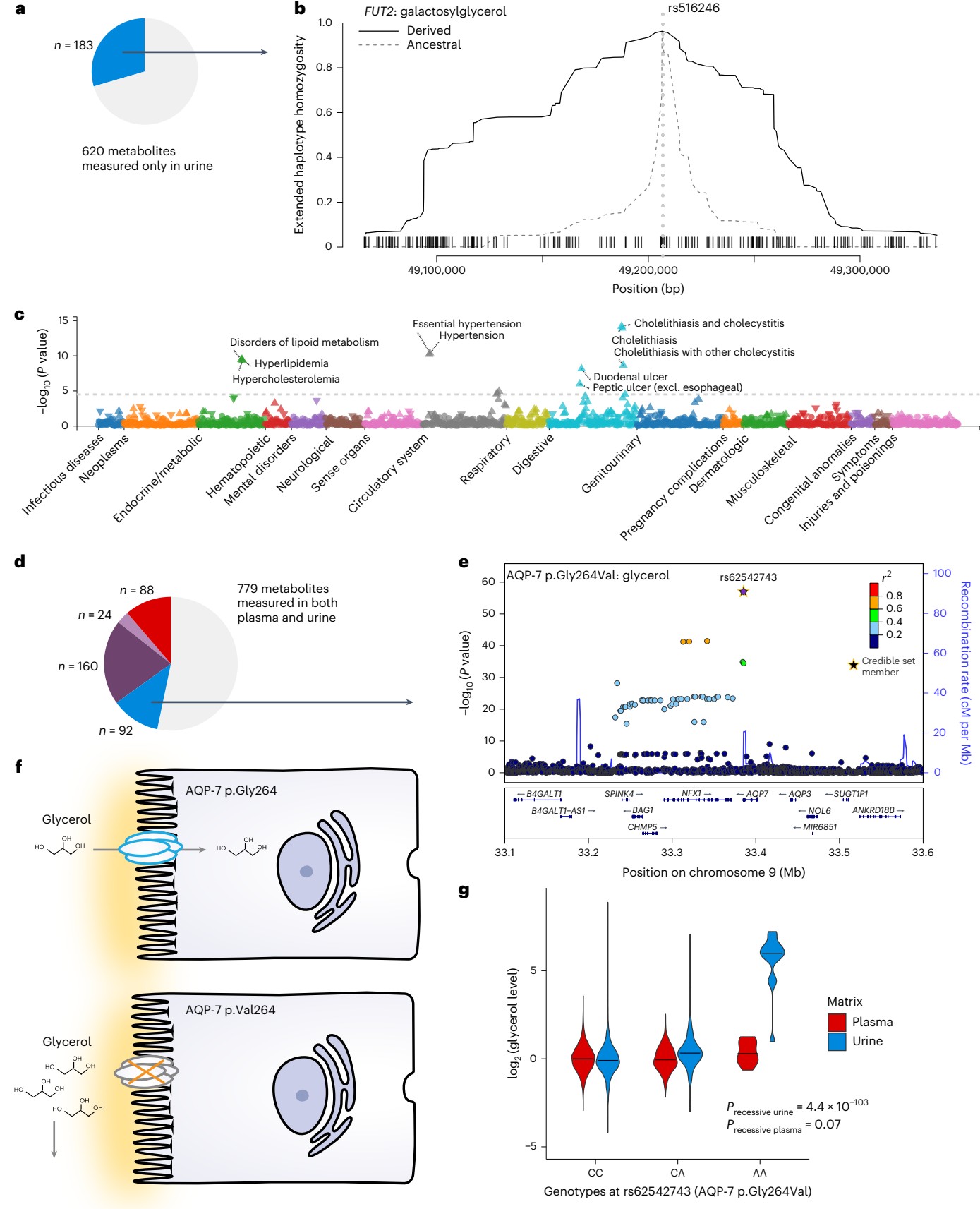

**Different matrices implicate distinct variants at *SLC10A2***

At *SLC10A2*, fine mapping revealed a single, yet different, underlying variant for the plasma mQTL of the secondary bile acid glycodeoxycholate

3-sulfate and the urine mQTL of the primary bile acid glycocholate (Supplementary Table 8). *SLC10A2* encodes the primary transporter for bile acid uptake in the distal ileum[30], ASBT, which is also responsible

**Fig. 5 | Urine-specific mQTLs deliver insights into systemic and kidney-specific processes. a**, The mQTLs highlighted in **b**,**c** belongs to the group arising from metabolites only measured in urine. **b**, The haplotype carrying the derived allele at the galactosylglycerol-associated mQTL at the *FUT2* locus shows extended homozygosity as compared to the haplotype carrying the ancestral allele; the *x* axis represents the genomic coordinates (bp in build 37) around the tested SNP rs516246, the proxy for the index SNP rs679574 ($r^2 = 1$); the *y* axis displays the extended haplotype homozygosity (EHH) statistic (Methods). The EHH around the derived allele is shown by the solid black line, whereas the one of the ancestral allele is shown by the gray dashed line. The dotted light gray line indicates the position of the tested SNP. Black dashes on the *x* axis represent the positions of SNPs that were used to compute the EHH statistic. **c**, Phenome-wide association study for rs516246 based on UK Biobank data (https://pheweb.org/UKB-TOPMed/variant/19:48702915-C-T)[51]. Excl., excluding. **d**, The mQTLs highlighted in **e**–**g** belongs to the group of urine-specific mQTLs arising from metabolites measured in plasma and urine. **e**, Regional association plot of the GWAS of urine glycerol levels (linear regression) around the most likely gene, *AQP7*. The index variant rs62542743 is shown in purple. **f**, Schematic representation of a presumed loss-of-function effect of AQP-7 p.Gly264Val on tubular reuptake of glycerol. **g**, Distribution of the $\log_2$-transformed glycerol levels in plasma (red) and urine (blue) by genotype at rs62542743. The black line in the center of each violin represents the median of the data.

for bile acid reabsorption in the proximal kidney tubules[31,32]. The fingerprints of the plasma index SNP rs55971546 and the urine index SNP rs16961281 on all quantified bile acid metabolites differed markedly (Fig. 6a,b): the minor T allele at rs55971546, encoding p.Val98Ile, was associated with higher plasma levels of several sulfated bile acids, which was propagated to urine (inner two bands). The ASBT 98Ile protein has experimentally been shown to result in a partial loss of function in one but not another in vitro system[33,34], and the corresponding T allele was associated with higher risk of gallstone disease[35].

On the other hand, the minor A allele at the urine index SNP rs16961281 showed a urine-specific fingerprint mainly as lower levels of glycocholate but also of cholate, glycoursodeoxycholate and glycodeoxycholate. These metabolites are known substrates of ASBT[36], whereas sulfated bile acids are not[37]. Thus, the urine bile acid profile may reveal genetic variants more directly related to ASBT function, whereas plasma levels may reflect secondary changes. The minor A allele at rs16961281 was significantly associated with higher *SLC10A2* gene expression in publicly available kidney expression quantitative trait locus (eQTL) data ($P = 8.7 \times 10^{-34}$)[38]. Colocalization supported a shared variant underlying lower urine glycocholate levels and higher renal *SLC10A2* expression (PP $H_4 = 1$; Fig. 6d).

We generated assay for transposase-accessible chromatin with sequencing (ATAC-seq) and RNA-seq data from manually dissected primary human kidney tissues to annotate prioritized variants (Methods) and found that rs16961281 mapped into highly accessible chromatin in the kidney cortex (Fig. 6c) and specifically in proximal tubular cells (Extended Data Fig. 8), supporting its regulatory function. Higher ASBT abundance should increase substrate reabsorption and result in lower urine levels, as observed (Fig. 6d). Colocalization of genetic associations with gallstone disease and urine glycocholate levels (concordant direction, PP $H_4 = 1$) as well as *SLC10A2* expression (inverse direction, PP $H_4 = 1$) supported the idea that higher ASBT abundance results in lower risk of gallstone disease (Fig. 6d). The occurrence of gallstones as a potential adverse effect of the new class of SLC10A2 inhibitors, such as odevixibat to treat cholestasis or elobixibat to treat constipation (Supplementary Table 12), therefore deserves attention. In fact, data from emerging clinical trials describe increased rates of cholelithiasis in the treatment group (https://clinicaltrials.gov identifier NCT03566238).

## Metabolome footprints of kidney-enriched proteins: *SLC13A3*

NaDC3, a kidney-enriched transport protein encoded by *SLC13A3*, also exemplifies a different metabolic 'footprint' in plasma and urine (Fig. 7a,b). We observed significant genetic associations with levels of plasma ($P = 1.2 \times 10^{-23}$) and urine ($P = 3.7 \times 10^{-25}$) methylsuccinoylcarnitine as well as plasma malate ($P = 4.7 \times 10^{-14}$) and fumarate ($P = 1.1 \times 10^{-11}$; Supplementary Tables 3). Functional annotation using our ATAC-seq and RNA-seq data and publicly available single-nucleus ATAC-seq data from the human kidney (Methods) showed that only rs6124828 of eight fine-mapped SNPs (Supplementary Tables 8 and 9) mapped into highly accessible chromatin, specifically in the kidney cortex (Fig. 7c) and in proximal tubule cells[39] (Extended Data Fig. 9). A potential regulatory function of rs6124828 was supported by histone chromatin immunoprecipitation followed by sequencing (ChIP–seq)-based chromatin state prediction from primary human kidney tissue (Methods) that showed active enhancer function at the variant's position. Screening of the presence of binding motifs of 517 kidney-expressed transcription factors at this position showed an intersection only with hepatocyte nuclear factor (HNF)1A and HNF1B. These master regulators of renal gene expression programs have been shown to bind at this position based on publicly available ChIP–seq data (Fig. 7c and the Methods). The minor A allele at rs6124828 was predicted to significantly reduce the binding probability of HNF1A and HNF1B (Fig. 7c), motivating an investigation of allele-specific binding in future ChIP datasets from primary human tissue.

NaDC3 is an Na$^+$–dicarboxylate transporter in the basolateral membrane of proximal tubule cells[40,41]. It transports a variety of substrates when overexpressed in cellular assays, including tricarboxylic acid cycle intermediates such as α-ketoglutarate, (methyl)succinate, malate and fumarate, that are used for mitochondrial energy generation[42,43]. Thus, genetic variants leading to lower expression of NaDC3 and hence lower intracellular substrate uptake, for example, via the presumed mechanism involving the regulatory A allele at rs6124828, are consistent with our observation of higher plasma levels of malate and fumarate and of lower levels of the resulting intracellular downstream metabolites such as methylsuccinoylcarnitine (Fig. 7d). The metabolomic signatures of *SLC13A3* shed light on physiological functions of NaDC3 in humans and permit identification of a likely causal regulatory allele.

**Fig. 6 | Plasma and urine implicate distinct causal variants and bile acid metabolites in the *SLC10A2* locus. a**, The *SLC10A2* locus contains two metabolite- and matrix-specific mQTLs. **b**, Systematic exploration of the effect of the urine mQTL rs16961281 (outer two bands) and the plasma mQTL rs55971546 (inner two bands) on levels of 39 bile acids quantified in plasma (red frames) and urine (blue frames) from their respective GWAS showed a urine-specific inverse association of the urine mQTL with glycocholate as well as other known substrates of the bile acid transporter encoded by *SLC10A2* in urine but not in plasma. The plasma mQTL was positively associated with specific, sulfated bile acids in plasma, and this metabolomic footprint was propagated to urine likely via glomerular filtration. The direction and magnitude of the modeled minor alleles on bile acid levels is color coded; dot size corresponds to significance levels. **c**, RNA-seq shows that *SLC10A2* expression is specific to the kidney cortex (plotted using pyGenomeTracks version 3.7). ATAC-seq highlights cortex-pronounced active chromatin that directly intersects with the fine-mapped urine mQTL rs16961281 (credible set size = 1), located in the 5′ untranslated region, flanking the active transcription start site (TSSFlnk, chromatin state band). RNA-seq and ATAC-seq tracks are an overlay of signal from three different tissue donors; chromatin states were derived from histone ChIP–seq data (see Extended Data Fig. 9 for the chromatin state legend; Methods). **d**, The findings for the urine-specific mQTL suggest that urine is the appropriate matrix to detect the effect of a minor allele at a regulatory variant that increases expression of *SLC10A2* in the kidney cortex, leading to lower urine levels of the ASBT substrate glycocholate through reabsorption, which translates into lower risk of gallstone disease. GoF, gain of function with respect to ASBT-mediated transport.

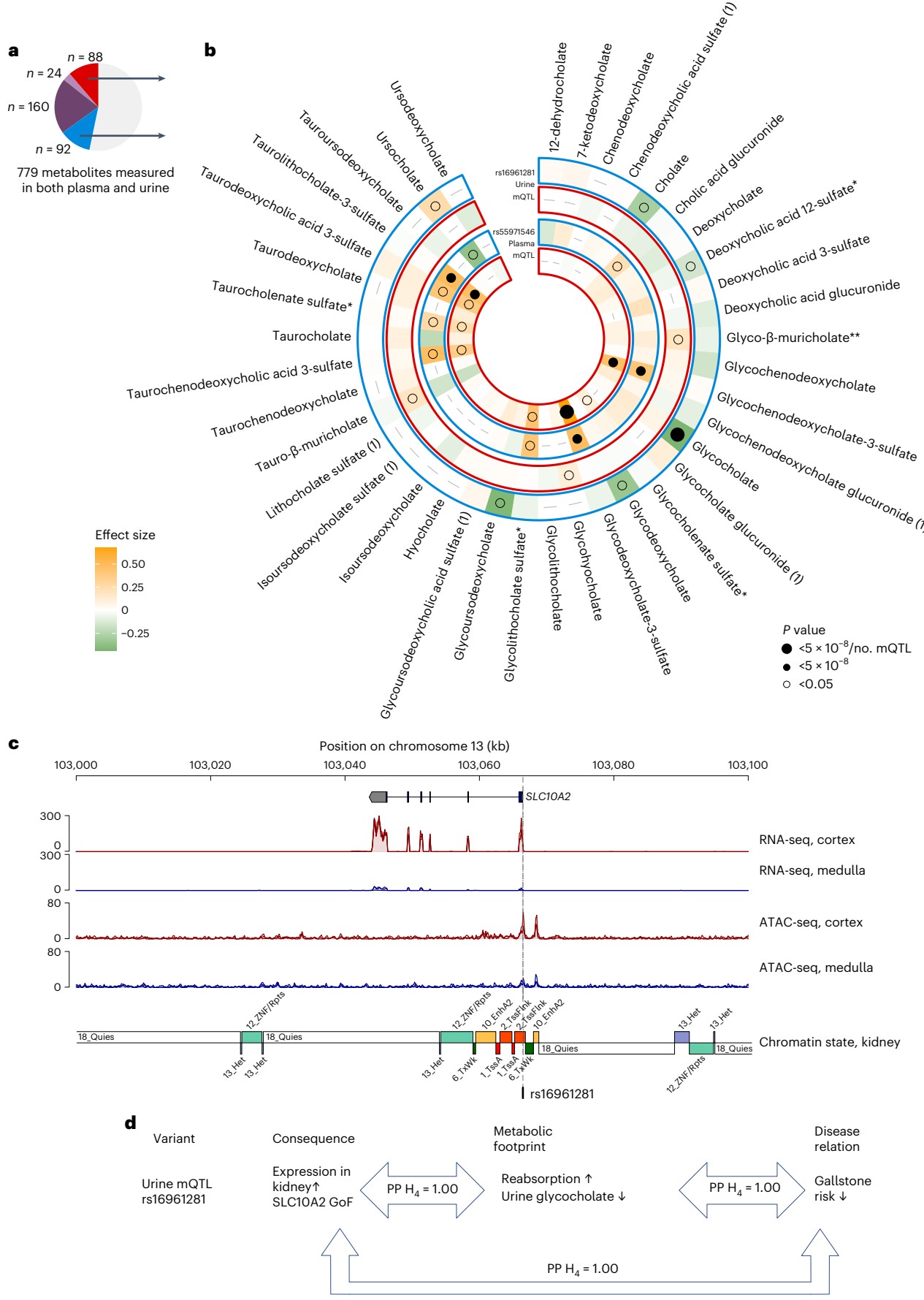

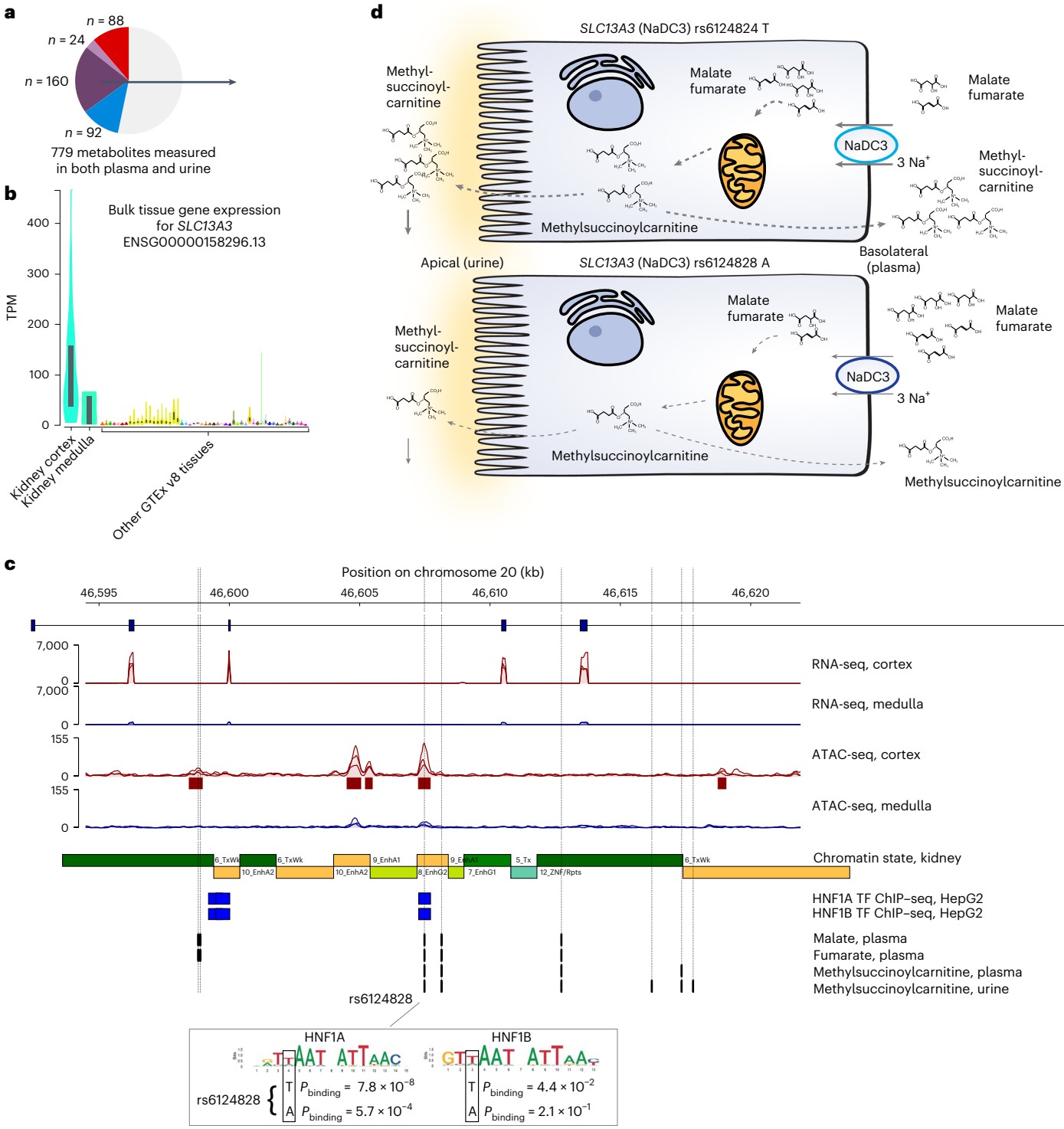

**Fig. 7 | Primary human kidney tissue permits prioritization of causal variants in kidney-enriched genes implicated by mQTLs. a**, The locus highlighted in this figure contains an mQTL identified with both plasma and urine measurements of a metabolite. **b**, *SLC13A3* transcript levels are particularly high in the kidney cortex and medulla among Genotype–Tissue Expression (GTEx) version 8 samples ($n_{\text{kidney cortex}} = 85$, $n_{\text{kidney medulla}} = 4$, $n_{\text{others}} = 9$–803; Methods). The dark bars in the violin plots mark the 25th and 75th percentiles. TPM, transcripts per million. **c**, RNA-seq shows that *SLC13A3* is predominantly expressed in the kidney cortex. ATAC-seq highlights cortex-specific active chromatin around the rs6124828 index SNP, which was associated with malate, fumarate (both in plasma) and methylsuccinoylcarnitine (in plasma and urine). RNA-seq and ATAC-seq tracks are an overlay of signal from three different tissue samples (donors). Chromatin

states derived from histone ChIP–seq data show an active enhancer state at the rs6124828 position (see Extended Data Fig. 9 for the chromatin state legend). The transcription factor (TF) motifs for HNF1A and HNF1B overlap rs6124828, and transcription factor ChIP–seq from HepG2 cells shows that the motifs are bound by both transcription factors. The minor A allele results in a higher predicted binding $P$ value for HNF1A and HNF1B. **d**, Schematic representation of the effect of genotype at the mQTL rs6124828 on NaDC3-mediated metabolite transport and subsequent intracellular metabolism. Intermatrix colocalization of genetic associations with methylsuccinoylcarnitine suggests that its levels in urine may reflect filtration from plasma, but an exit at the apical membrane cannot be excluded.

### Systemic roles of dipeptidase 1: digestive enzymes and diseases

*DPEP1* encodes dipeptidase 1 (DPEP1), an ectoenzyme localized on the apical membrane of tubular kidney cells. It has a role in dipeptide hydrolysis, including glutathione metabolism breakdown products such as cysteinyl-*bis*-glycine[44]. DPEP1 has been well studied in the kidney because it metabolizes several β-lactam antibiotics[45]. We detected several *DPEP1* mQTLs for glutathione pathway metabolites as well as for dipeptides in urine. For example, the index SNP for urine prolylglycine ($P = 9.0 \times 10^{-369}$) explained 25% of its variance. Although the high renal expression of DPEP1 and its apical localization may be expected to primarily affect the urine metabolome, we also observed six plasma mQTLs for glutathione-related (cysteinylglycine, oxidized cysteinylglycine, cysteinylglycine disulfide*) and other (picolinate, picolinoylglycine, X-25244) metabolites, suggesting extra-renal roles (Fig. 8a and Supplementary Table 3).

*DPEP1* is also highly expressed but less well studied in the small intestine, pancreas and testis (Fig. 8b). We therefore explored additional, systemic roles of DPEP1 through colocalization analysis using data from a recently published GWAS of the circulating plasma proteome (Methods)[46]. Eight of 4,907 protein readouts contained significant associations with SNPs in the *DPEP1* locus, including the DPEP1 protein itself. Strikingly, all of the seven other proteins are digestive enzymes or zymogens produced primarily in the exocrine pancreas and secreted into the small intestine (Supplementary Table 23). Positive colocalizations supported a shared genetic basis of *DPEP1*-related metabolites and of circulating readouts of DPEP1 and the digestive enzymes (Fig. 8c and Supplementary Fig. 3). Higher urine prolylglycine, that is, lower inferred DPEP1 function, was associated with lower plasma levels of DPEP1 and all seven digestive enzymes. These observations point toward an underappreciated role of DPEP1 and motivate experimental studies to identify the underlying mechanisms.

We also detected multiple, pleiotropic colocalizations of *DPEP1*-related mQTL, especially with osteoarthrosis, hypertension and intake of blood pressure medication (Supplementary Table 13 and Fig. 8d). Colocalization supported inverse associations between genetically higher levels of glutathione-related metabolites, that is, lower DPEP1 activity, and lower risk of arthropathies, consistent with a reported beneficial effect of glutathione on osteoarthritis[47]. Conversely, genetically predicted higher levels of urine and plasma picolinate and picolinoylglycine showed a positive relationship with osteoarthrosis. An opposite pattern was observed with respect to hypertension (Fig. 8d). Modification of DPEP1 function, for example, with its specific inhibitor cilastatin, is therefore expected to have opposing effects on osteoarthritis and hypertension risk.

## Discussion

This large-scale comparative study of the genetic footprint on the plasma and urine metabolomes uncovered numerous associations not reported previously and yielded several principal findings: first, the number of detected mQTLs is similarly large in plasma and urine, while the underlying metabolites show differences. Second, multi-matrix studies deliver many more associations than the individual analysis of similarly sized plasma or urine studies. Third, differences in metabolomic footprints between plasma and urine can deliver insights into the physiological function and localization of proteins operating at compartment interfaces and implicate different disease-related mechanisms. Fourth, the detected mQTLs and their colocalizing traits and diseases constitute a rich resource for the formulation of biologically plausible hypotheses regarding the in vivo physiological function of transporters and enzymes for future experimental studies.

Genetic studies of the metabolome using multiple matrices can provide information that cannot be obtained from studies using a single matrix such as plasma. Not only were ~60% of the metabolites quantified in just one matrix, but the combined study of paired metabolomes also allowed for the distinction of kidney-specific and systemic processes. In fact, 49% of mQTLs arising from a metabolite quantified in both matrices were detected exclusively in plasma or in urine, underscoring the fact that plasma and urine contain complementary information on the handling of metabolites by different organs. This is exemplified by the effect of AQP-7 p.Gly264Val on glycerol levels: urine is in direct contact with the apical membrane of tubular epithelial cells, where this glycerol transporter is expressed. Urine therefore is the appropriate matrix to capture the function of this transporter in the kidney in vivo, as was true for a urine-specific association between bile acids and a regulatory variant affecting renal *SLC10A2* expression. Conversely, the detection of plasma-specific effects of *SLC13A3* variants on malate and fumarate levels can be explained by the basolateral localization of the encoded NaDC3 transporter in kidney epithelial cells. More generally, the study of similarities and differences of the paired plasma and urine metabolome is especially informative for functions of the kidney. Paired studies of the plasma metabolome and other matrices such as intestinal fluids or breath air could provide new insights about specific functions of the digestive organs and lungs, respectively.

Our study confirms that common genetic variants, mQTLs, sometimes explain >50% of the observed metabolite variance. Although this translates into much smaller effects on complex diseases such as hypertension, arthropathies or gallstone disease, colocalization can nominate shared pathophysiological mechanisms and inform about potential therapeutic targets, repurposing opportunities and potential side effects of approved drugs. Our study includes numerous such examples, supported by the recent launch of new drugs such as evinacumab, a monoclonal antibody targeting angiopoietin-like 3 (ANGPTL3) to treat dyslipidemia, or the SLC10A2 inhibitor odevixibat to treat cholestasis. Even if a target implicated by metabolites in our study is not desirable or amenable for therapeutic modulation, disease-associated metabolites may represent valuable intermediate biomarkers for risk prediction or response to treatment.

Some limitations warrant mention: while we show here and in prior work[5] that genetic effects on metabolites are of comparable direction and magnitude in persons with and without reduced eGFR, future studies are required to examine whether our findings are generalizable to persons of non-European ancestry. Our study did not test the effects of rare and ultra-rare coding variants that may have particularly large effects, which could address remaining uncertainties in the assignment of the underlying causal gene(s) inherent to GWAS. Our gene-prioritization workflow incorporated information on gene expression from dozens of tissues. In addition to differences in tissue sample size, such prioritization can implicate several genes and tissues in a given locus, including scenarios in which different genes in one locus receive support from different tissues[48]. Moreover, our workflow prioritized coding genes over noncoding ones such as long noncoding RNA. Although many GWAS loci in which a causal gene has been experimentally validated implicate coding genes, noncoding genes are also recognized mediators of association signals with complex human traits such as cardiovascular diseases[49,50]. Lastly, our study employed semi-quantitative metabolite quantification, while additional targeted studies with absolute quantification are required to study fractional metabolite excretion and for clinical translation.

In conclusion, this genetic study of the paired plasma and urine metabolome emphasizes the role of multi-matrix studies to gain new insights into in vivo metabolic processes in general and the function of the kidney in particular. The results provide a rich resource for the experimental validation of yet unknown enzymatic and transport processes that may represent a molecular link between genetic variants and human traits and diseases.

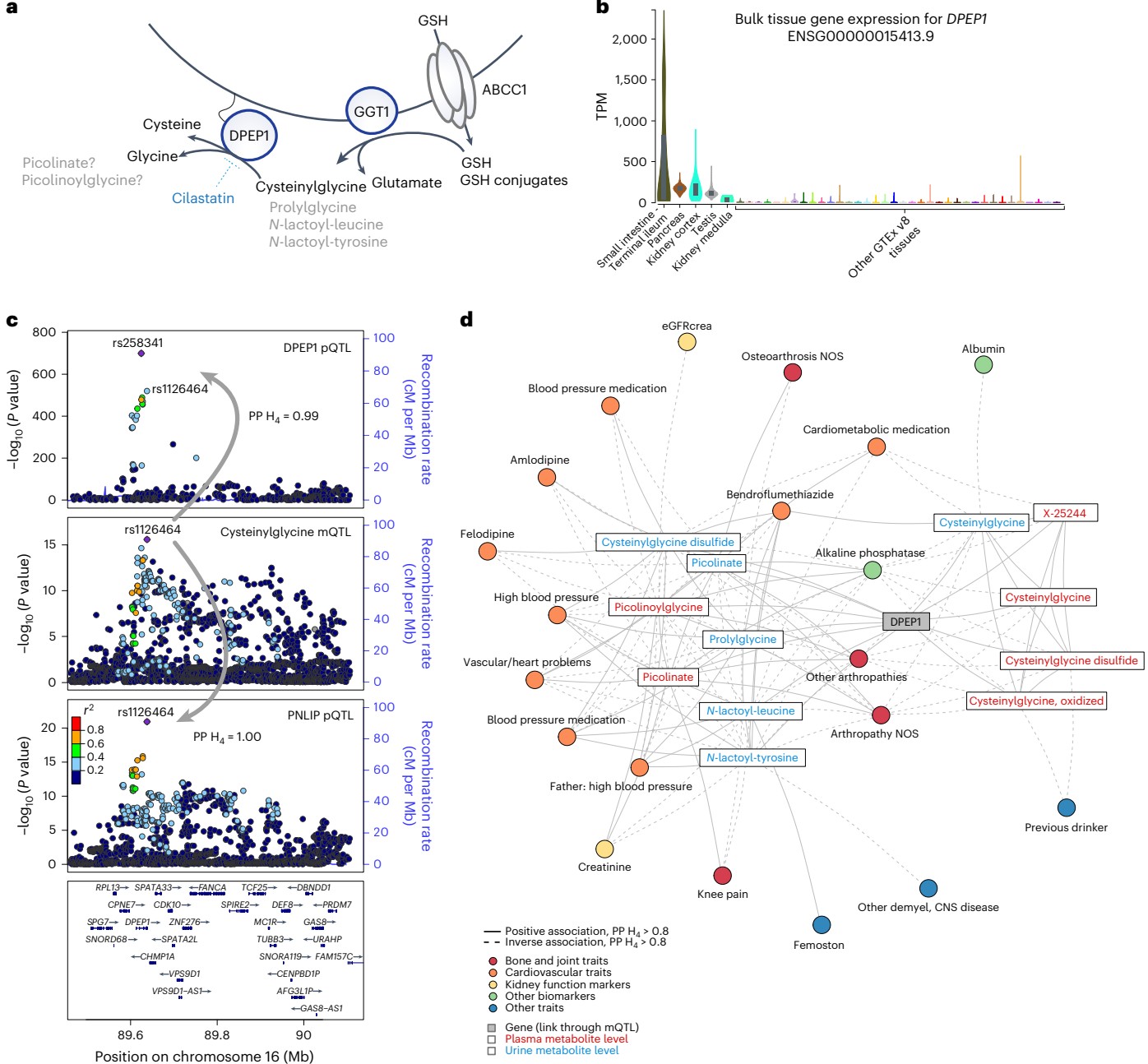

**Fig. 8 | DPEP1 influences plasma levels of major digestive enzymes.**
**a**, Schematic representation of the role of DPEP1, encoded by *DPEP1*, and several other genes in glutathione (GSH) metabolism, highlighting identified metabolites and genes. ABCC1, ATP-binding cassette subfamily C member 1; GGT1, γ-glutamyltransferase 1. **b**, *DPEP1* transcript levels are particularly high in the small intestine (terminal ileum), pancreas, kidney and testis among GTEx version 8 samples ($n_{\text{small intestine, terminal ileum}} = 187$, $n_{\text{pancreas}} = 328$, $n_{\text{kidney cortex}} = 85$, $n_{\text{testis}} = 361$, $n_{\text{kidney medulla}} = 4$, $n_{\text{others}} = 9{-}803$; Methods). The dark bars in the violin plots mark the 25th and 75th percentiles. **c**, Regional association plots of association patterns at the *DPEP1* locus (linear regression). SNPs are plotted by position (build 38) versus $-\log_{10}$ (association *P* values) of plasma DPEP1 levels (top; conditional independent protein quantitative trait locus (pQTL) statistics with the index SNP rs258341), urine cysteinylglycine (middle; mQTL) and plasma levels of the digestive enzyme pancreatic triacylglycerol lipase (PNLIP) (bottom;

conditional independent pQTL statistics with the index SNP rs1126464). The purple diamond highlights the index SNP for each association. SNPs are color coded to reflect their LD with this SNP (pairwise European-ancestry $r^2$ values from the 1000 Genomes Project phase 3). Genes, exons and the direction of transcription from the University of California at Santa Cruz Genome Browser are depicted. Plots were generated using LocusZoom[52]. **d**, Network representation of metabolites with a *DPEP1* mQTL as well as of all traits in the phenome-wide scan that are linked through positive colocalization for one of these. mQTLs are represented by the edge connecting the respective gene and metabolite, and all other edges are established through positive colocalization (PP $H_4 > 0.8$), with color coding representing the phenotype category. Effect directions are indicated by the line type (solid, positive association; dashed, inverse association). CNS, central nervous system; NOS, not otherwise specified.

## Online content

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

[1]Institute of Genetic Epidemiology, Faculty of Medicine and Medical Center - University of Freiburg, Freiburg, Germany. [2]Department of Epidemiology, Johns Hopkins Bloomberg School of Public Health, Baltimore, MD, USA. [3]Spemann Graduate School of Biology and Medicine, University of Freiburg, Freiburg, Germany. [4]Institute of Human Genetics, University Hospital Erlangen, Friedrich-Alexander-Universität Erlangen–Nürnberg, Erlangen, Germany. [5]Membrane Transport Discovery Lab, Department of Nephrology and Hypertension and Department of Biomedical Research, University of Bern, Bern, Switzerland. [6]Metabolon, Inc., Morrisville, NC, USA. [7]Department of Medicine IV—Nephrology and Primary Care, Faculty of Medicine and Medical Center, University of Freiburg, Freiburg, Germany. [8]Research and Early Development, Pharmaceuticals Division, Bayer AG, Wuppertal, Germany. [9]Department of Clinical Sciences in Malmö, Lund University Diabetes Centre, Lund University, Lund, Sweden. [10]Epidemiology, Human Genetics and Environmental Sciences, School of Public Health, University of Texas Health Science Center at Houston, Houston, TX, USA. [11]New York University Grossman School of Medicine, New York, NY, USA. [12]Human Genome Sequencing Center, Baylor College of Medicine, Houston, TX, USA. [13]Centre for Integrative Biological Signalling Studies (CIBSS), Albert-Ludwigs-University Freiburg, Freiburg, Germany. [14]Institute of Genetic Epidemiology, Department of Genetics, Medical University of Innsbruck, Innsbruck, Austria. [15]Department of Nephrology and Hypertension, University Hospital Erlangen, Friedrich-Alexander-Universität Erlangen–Nürnberg, Erlangen, Germany. [16]Department of Laboratory Medicine and Pathology, University of Washington, Seattle, WA, USA. [17]Freiburg University Faculty of Medicine, Center for Pediatrics and Adolescent Medicine, University Hospital Freiburg, Freiburg, Germany. [18]Department of Nephrology and Medical Intensive Care, Charité—Universitätsmedizin Berlin, Berlin, Germany. [19]Institute of Functional Genomics, University of Regensburg, Regensburg, Germany. [20]These authors contributed equally: Pascal Schlosser, Nora Scherer. *A list of authors and their affiliations appears at the end of the paper. ✉e-mail: pascal.schlosser@uniklinik-freiburg.de; anna.koettgen@uniklinik-freiburg.de

## GCKD Investigators

**Fruzsina Kotsis**[1,7]**, Florian Kronenberg**[14]**, Heike Meiselbach**[15]**, Ulla T. Schultheiss**[1,7]**, Kai-Uwe Eckardt**[15,18]**, Peter J. Oefner**[19] **& Anna Köttgen**[1,2,13]

## Methods

### Study design and participants

The GCKD study is an ongoing prospective observational study that enrolled 5,217 adult persons with CKD between 2010 and 2012. Patients regularly seen by nephrologists with eGFR between 30 and 60 ml min⁻¹ per 1.73 m² or eGFR >60 ml min⁻¹ per 1.73 m² with UACR > 300 mg per g (or urinary protein/creatinine ratio > 500 mg per g) were included[53]. This study used biomaterials collected at the baseline visit, shipped frozen to a central biobank and stored at −80 °C[54]. A more detailed description of the study design, standard operating procedures and the recruited study population has been published[53,55]. The GCKD study was registered in the national registry for clinical studies (DRKS 00003971) and approved by local ethic committees of the participating institutions (universities or medical faculties of Aachen, Berlin, Erlangen, Freiburg, Hannover, Heidelberg, Jena, München and Würzburg)[53]. All participants provided written informed consent. For this project, metabolites were quantified from stored EDTA plasma and spot urine. Information on genome-wide genotypes, covariates and metabolites was available for 4,960 (plasma) and 4,912 (urine) persons.

### Genotyping and imputation

Genotyping and data cleaning in the GCKD study were conducted as follows[5,56]. Genomic DNA from GCKD participants was genotyped at 2,612,357 variants using Illumina Omni2.5Exome BeadChip arrays and imputed using minimac3 version 2.0.1 at the Michigan Imputation Server[57] and the Haplotype Reference Consortium haplotype version r1.1 and Eagle 2.3 for phasing. On the variant level, SNPs with <96% call rate, imputation quality of $r^2 \leq 0.3$, MAF < 1% or deviating from Hardy–Weinberg equilibrium ($P < 1 \times 10^{-10}$) and all multi-allelic SNPs were removed. The cleaned genotype dataset contained 5,034 individuals and 7,724,508 high-quality autosomal variants for GWAS. Genotyping of ARIC samples was performed on the Affymetrix 6.0 DNA microarray and filtered for call rates <90% and Hardy–Weinberg equilibrium $P$ values < $10^{-6}$. SNPs were then imputed to the TOPMed Freeze 5b reference panel and filtered for $r^2 \leq 0.1$ (imputation quality).

### Metabolite identification and quantification

Non-targeted mass spectrometry analysis was performed at Metabolon, and sample preparation was carried out as published by Schlosser et al.[5]. Automated comparison of the ion features in the experimental samples to a reference library of chemical standard entries (>4,500 purified standards) was used for metabolite identification. Known metabolites reported in this study conformed to confidence level 1 (the highest confidence level of identification) of the Metabolomics Standards Initiative[58,59], unless otherwise denoted with an asterisk. Additional mass spectral entries have been created for compounds of unknown structural identity (unnamed biochemicals; >2,750 in the Metabolon library), which have been identified by virtue of their recurrent nature (both chromatographic and mass spectral). Peaks were quantified using the area under the curve and normalized to correct for variation resulting from instrument interday tuning differences by the median value for each run day. Likewise, metabolites in the ARIC replication sample were also quantified with the Metabolon HD4 platform.

### Data cleaning of quantified metabolites

An in-house pipeline was set up for data quality control, filtering and normalization of metabolite concentrations. No plasma specimens and four pairs of urine specimens with a Pearson correlation coefficient greater than 0.9 and differing sample IDs were removed. Four plasma specimens and no urine specimens were removed for >50% missing data. A total of 130 plasma and 131 urine metabolites were removed, as less than 300 genotyped samples were available.

To account for urine dilution, concentrations of each metabolite were pq normalized based on endogenous metabolites with <1% missing values ($n_{\text{metabolites}} = 309$)[60]. Of the log₂-transformed metabolites,

15 plasma metabolites were excluded for low variance (<0.01), and none were excluded for too many outliers (>5% of samples outlying >5 s.d.). Three plasma samples and one urine sample represented an outlier >5 s.d. along one of the first 15 principal components based on metabolites with complete information. The final dataset consisted of 1,296 plasma and 1,401 urine log₂-transformed traits for subsequent GWAS. Supplementary Table 2 provides detailed annotation of the metabolites, including heritability estimates for metabolites with at least one genetic association. Over the course of this project, two formerly different urine metabolites were merged because they represented the same molecule: X-12739 and X-24527 to the glutamine conjugate of $C_6H_{10}O_2$ (1)* and X-23667 and X-24759 to (2-butoxyethoxy)acetic acid.

### Definition of additional variables

In the GCKD study, an IDMS-traceable enzymatic assay (Creatinine Plus, Roche) was used to measure serum creatinine levels, for estimating GFR by means of the CKD-EPI formula[61], and to measure urine creatinine levels. The Tina-quant Albumin assay (Roche) was used to measure serum and urine albumin, for adjustment and calculation of the UACR, respectively. The GFR was estimated in the ARIC study from serum creatinine and cystatin C using the CKD-EPI formula[62].

### Genome-wide association study of metabolite levels

Based on log₂-transformed metabolite levels, residuals adjusted for age, sex and the first three genetic principal components were generated (similar to previous mGWAS[5,6,56,63,64]), with plasma levels additionally adjusted for ln(eGFR) and serum albumin. GWAS analyses of these residuals were performed with SNPTEST version 2.5.2 (https://www.well.ox.ac.uk/~gav/snptest/), using imputed genotype dosages and linear regression under additive modeling. Statistical significance was defined as genome-wide significance after correcting for multiple testing by a Bonferroni procedure ($3.9 \times 10^{-11} = 5 \times 10^{-8} \div 1,296$ plasma traits; $3.6 \times 10^{-11} = 5 \times 10^{-8} \div 1,401$ urine traits).

Significantly associated SNPs were assigned to mQTL by selecting, for each trait, the SNP with the lowest $P$ value as the index SNP, defining the corresponding locus as a 1-Mb interval centered on the index SNP and repeating the procedure for unassigned SNPs until no further genome-wide significant SNP remained. For each trait, overlapping intervals were combined into mQTL. The extended MHC region (chromosome 6, 25.5–34 Mb) was treated as one region. For associations with MAF < 3%, mQTLs were only kept if the index SNP remained significant with inverse-normal-transformed metabolite data. A regional association plot centered on the index SNP for each mQTL was generated using LocusZoom (version 1.3) and LD information from GCKD study genotypes[52]. Circular plots were created using Circos version 0.69-6 (ref. 65). The variance in metabolite levels explained by the index SNP of an mQTL was computed independently of other covariates.

We compared our findings to those from seven large studies of the plasma–serum metabolome that were published in peer-reviewed journals and shared their summary statistics[6–12]. These studies were selected to maximize overlap with our findings as studies of EA participants with large sample size that examined the effects of common SNPs on plasma–serum metabolite levels quantified with the Metabolon assay, rather than on rare variant association studies, or GWAS of metabolites quantified by different methods and/or in other populations, for example, refs. 66–70. Metabolites were matched by compound or chemical ID, if available, and biochemical name (ones not identical were checked manually). First, for each mQTL identified in one of the published plasma–serum studies mentioned above, available index SNPs were extracted from GWAS of the corresponding metabolite in plasma and urine, and effect direction and statistical significance were assessed at different levels of statistical significance ($P$ value < 0.05, <0.05 ÷ no. mQTLs in the previous study, <$5 \times 10^{-8}$ and <$5 \times 10^{-8}$ ÷ no. mQTLs in the previous study). Validation required effect-direction consistency for comparisons involving results from

the GCKD plasma mGWAS. Second, for each mQTL identified in this study in plasma or urine, availability of the corresponding index SNP and metabolite in the summary statistics of the previously published plasma–serum studies was assessed. If the index SNP was missing, we searched for proxy SNPs in high LD ($r^2 > 0.8$) within a window of ±500 kb around the index SNP based on genetic data from the 1000 Genome Project phase 3 version 5 of European ancestry using https://snipa.helmholtz-muenchen.de/snipa/?task=proxy_search. For each study, the best available proxy SNP in terms of maximal LD and minimal distance was selected. Summary statistics were downloaded from https://metabolomics.helmholtz-muenchen.de/gwas/index.php?task=download (Shin et al.[6]), http://www.hli-opendata.com/Metabolome (Long et al.[7], only summary statistics with $P$ value $< 10^{-5}$), https://omicscience.org/apps/crossplatform/ (Lotta et al.[8]), https://pheweb.org/metsim-metab/ (Yin et al.[10]), https://omicscience.org/apps/mgwas/mgwas.table.php (Surendran et al.[11]) and http://ftp.ebi.ac.uk/pub/databases/gwas/summary_statistics/; accession numbers for European GWAS are GCST90199621–GCST90201020 (Chen et al.[12]). Hysi et al.[9] shared their summary statistics upon request.

To determine the number of urine mQTLs not reported in our earlier study[5], we examined for each mQTL from this study whether an associated SNP within a window of ±500 kb for the corresponding metabolite was identified in the earlier study.

Replication analyses in the ARIC study were performed using $\log_2$-transformed metabolite levels and the same covariables. Statistical significance was defined by a Bonferroni procedure ($P$ value $< 0.05 \div 459$ and $0.05 \div 430$ association tests with matching data for EA and AA, respectively) and consistent effect directions as in the GCKD study.

We included an interaction term between the mQTL and sex in a linear regression model with the same adjustments as before to test for potential differences of the 1,299 mQTLs in men and women. For significant interactions ($P$ value $< 0.05 \div 1,299$), we performed sex-stratified analyses (Supplementary Table 7).

## Heritability estimation

A genetic relationship matrix was calculated from all autosomal SNPs with an imputation quality of $r^2 > 0.6$ using GCTA-GRM[71]. GCTA-GREML[72] was then used to estimate the proportion of variation in $\log_2$-transformed and, in the case of urine, pq-normalized metabolite levels that can be explained by the SNPs for all metabolites that gave rise to an mQTL.

## Independent SNP selection and statistical fine mapping

We identified independent signals within mQTL using approximate conditional analyses, with LD information estimated from our study sample. The fine-mapping regions of mQTL were aligned within matrices across metabolites, if index SNPs were in LD ($r^2 > 0.8$). For each mQTL, the GCTA-COJO Slct algorithm version 1.91.6 (ref. 73) was used to identify independent genome-wide significant SNPs ($P_{conditional} < 3.9 \times 10^{-11}$), using a collinearity cutoff of 0.1. For mQTL with multiple independent SNPs, approximate conditional analyses were carried out conditioning on the other independent SNPs in the region using the GCTA-COJO Cond algorithm to estimate conditional effect sizes. Statistical fine mapping was performed for all independent SNPs per mQTL. In loci with a single independent SNP, approximate Bayes factors (ABFs) were calculated from the original GWAS effect estimates using Wakefield's formula[74] with a standard deviation prior of 1.33. For mQTL with multiple independent SNPs, ABFs were derived from the conditional effect estimates. The SNP's ABF was used to calculate the posterior probability for the variant driving the association signal (PPA, 'causal variant'). Credible sets were calculated by summing the PPA across PPA-ranked variants until the cumulative PPA was >99%. $\log_2$-transformed credible set sizes were regressed on the MAFs of independent index SNPs.

## Pairwise colocalization tests of plasma and urine mQTL

To examine whether association patterns with metabolites measured in plasma and/or urine are shared across or within matrices, we conducted pairwise colocalization analyses between mQTL. When the windows of ±500 kb around the index SNPs for two mQTLs overlapped, colocalization was performed within the region of the merged windows using a version of Giambartolomei's colocalization method[75] as implemented with the 'coloc.fast' function from the R package 'gtx' (https://github.com/tobyjohnson/gtx) with default parameters and prior definitions. To visualize the effect sizes and explained variance for colocalizing signals for mQTLs detected for the same metabolite across matrices (Extended Data Fig. 4), we used the R package 'circlize' (ref. 76).

## Annotation

SNP annotation was performed by querying the SNiPA database version 3.4 (released 13 November 2020)[13], based on the 1000 Genomes phase 3 version 5 and Ensembl version 87 datasets. The retrieved combined annotation-dependent depletion (CADD) score was based on CADD version 1.3. The Ensembl VEP tool was used for the effect prediction of SNPs. SNiPA was used to collect the following annotations for each index SNP: gene hit or close by, regulated genes, CADD score, SnpEff effect impact (exonic and noncoding), mQTL, pQTL, GWAS Catalog, $cis$ eQTL, disease genes (based on ClinVar, OMIM, HGMD and Drugbank) and UK Biobank associations.

To select the most likely causal gene for each mQTL, the following steps were carried out: first, we compiled the 'genes' and 'evidence' information based on SNiPA[13]. Index SNPs were queried for association with differential expression of a nearby gene in tubulointerstitial kidney portions ($cis$ eQTL) from 187 patients with CKD using the NephQTL browser[77] and GTEx version 8 eQTL data[78]. Similarly, SNPs were queried for associations with differential levels of nearby proteins in plasma (2,751 unique proteins represented by 3,022 SOMAmers) in data from Sun et al.[79] downloaded from http://www.phpc.cam.ac.uk/ceu/proteins/. Second, when one or more $cis$ eQTL or $cis$ pQTL associations with $P < 0.05 \div 409$ (plasma, 409 unique index SNPs) or $P < 0.05 \div 410$ (urine, 410 unique index SNPs), respectively, was identified within ±100 kb of an index SNP, colocalization analyses of the respective metabolite(s)' mQTL and each of the eQTL and/or pQTL association(s) were performed within the eQTL–pQTL $cis$ window in the underlying study (gene region ±500 kb) using the method outlined above. Positive colocalizations with gene expression received equal weight for all investigated tissues to maximize the opportunity to detect processes in tissues interacting with blood and being filtered to urine. Sensitivity analyses assigning 1.5-fold and twofold greater weight to colocalizations arising from kidney or liver tissue or from kidney tissue only yielded almost identical results. The evidence codes h, r, e, p, m and c based on SNiPA[13] correspond to gene hit or close by, regulated genes, $cis$ eQTL, $cis$ pQTL, missense variants and disease genes based on pathogenic variants known to cause monogenic diseases, respectively. The evidence code E designated genes with evidence for colocalization with gene expression genome-wide association, and P designated those with protein-level genome-wide association. Evidence codes were collected and summed for each gene, where Ee and Pp only counted as one. The gene with the highest sum of scores within each locus was assigned as the most likely causal gene. In the case of ties, genes with evidence for gene expression colocalization were prioritized, followed by protein-level colocalization, followed by genes for which an inborn error of metabolism with the corresponding metabolite is known. When ties still remained, Ee scores were prioritized over E scores and Pp scores were prioritized over P scores. In all other cases, ties were resolved by prioritizing the closest gene; prioritization by distance determined the assigned most likely causal gene at 17% (221 of 1,299) of mQTLs. Lastly, the prioritized gene list was manually reviewed for biological plausibility based on published evidence and at colocalizing mQTLs as outlined in the Supplementary Methods. In case of a clear biological fit to another scored

gene (that is, corresponding monogenic disease or animal model), the prioritized gene was reassigned. This final gene list (*n* = 282) was used as input for downstream gene-based analyses. Known drugs were annotated for each gene and the corresponding indication and status of approval based on https://platform.opentargets.org/.

**Relation of mQTLs to plasma proteins in *trans* and phenotypes**
We also performed colocalization analyses of mQTLs with disease outcomes and biomarker measurements in the UK Biobank, with two representative kidney function traits and with *trans* pQTLs using the precomputed pQTL data from Sun et al.[79] to gain insights into clinical consequences and potential molecular mediators of mQTLs. Association summary statistics between SNPs and 30 biomarkers from the UK Biobank baseline examination, including the liver function markers AST, ALT, GGT, bilirubin and albumin, were computed using BOLT-LMM[80] (application no. 20272) in the same subset of European-ancestry participants as previous studies[81]. Precomputed GWAS summary statistics of diseases as ascertained in the UK Biobank and analyzed using phecodes were obtained from https://www.leelabsg.org/resources (1,403 binary traits) and from https://yanglab.westlake.edu.cn/data/ukb_fastgwa/imp_binary/ (2,325 of 2,989 binary traits[82]; traits containing job-coding terms were excluded from the analysis). There were 816 phecodes analyzed in both, but only unique phecodes were counted for positive colocalizations. We used GWAS summary statistics of creatinine-based and cystatin C-based eGFR (eGFRcrea and eGFRcys) from Stanzick et al.[83], who meta-analyzed kidney function GWAS from the CKDGen Consortium and the UK Biobank. The GWAS summaries were downloaded from the CKDGen data website at https://ckdgen.imbi.uni-freiburg.de. Colocalization testing between mQTL and *trans* pQTL was performed within a window of ±500 kb around the mQTL's index SNP when at least one *trans* pQTL association with $P < 0.05 \div 409 \div 3{,}000$ for plasma and $P < 0.05 \div 410 \div 3{,}000$ for urine was present within a window of ±100 kb around the index SNP. Similarly, colocalization analysis between mQTL and biomarkers, diseases and kidney function traits was performed within ±500 kb of the index SNP when there were one or more associated variants with $MAF > 0.01$ and $P < 0.05 \div 409$ or $P < 0.05 \div 410$, respectively, within ±100 kb of the index SNP, using the method outlined above.

Moreover, we investigated whether the most likely mQTL-related genes contained rare, putatively damaging variants that in aggregate are associated with clinical traits and diseases. Gene–phenotype associations based on whole-exome-sequencing data from ~450,000 UK Biobank participants were obtained on 4 February 2022 from the AstraZeneca PheWAS Portal (https://azphewas.com/) for the 274 available genes of the 282 mQTL-related genes[22]. We identified 2,745 distinct suggestive ($P < 1 \times 10^{-5}$) gene–phenotype associations for 115 of those genes. The significance threshold as derived for the PheWAS was $2 \times 10^{-9}$ (ref. [22]). Only the most significant collapsing model per trait was retained for Fig. 4b. In addition, the exome-wide variant-level results were downloaded on 26 August 2022. The 17,493 analyzed phenotypes were queried for significant ($P$ value $< 0.05 \div 17{,}493$) associations with mQTL index SNPs (Supplementary Table 16).

We further performed colocalization testing of independent signals for all the 12 identified mQTLs within the *DPEP1* genomic region and plasma proteins with a reported pQTL in the *DPEP1* locus[46]. Metabolite and plasma protein summary statistics were extracted with a 500-kb flanking region around *DPEP1* and the *DPEP1* mQTL index SNP for the proteins CPB1, AMY2B, PNLIP, AMY2A, REG3G, CTRB2 and PNLIPRP1. First, independent association signals were identified based on approximate conditional analyses via the GCTA-COJO Slct algorithm ($P$ value $< 5 \times 10^{-8}$; collinearity threshold = 0.1)[73]. For each conditionally independent SNP, conditional summary statistics were computed by conditioning on all other independent SNPs in the region using the GCTA-COJO Cond algorithm (collinearity threshold = 0.1)[73]. Subsequently, colocalization analyses were conducted

for all pairwise combinations of the conditionally independent mQTL and pQTL associations as outlined above. For the gallstone disease GWAS[84] and urine glycocholate, we performed colocalization analysis of signals conditioning on the plasma index SNP (rs55971546) within ±500 kb of the *SLC10A2* urine mQTL index SNP (rs16961281). The same conditional mQTL summary statistics were colocalized with kidney eQTL[38]. Marginal statistics were used for these, as rs55971546 was not available (FDR > 0.01).

**Processing of gene expression data from tissue and cell types**
To test for over-representation of plasma or urine mQTL-related genes among those highly expressed in specific tissues and cell types, we compiled bulk and single-cell gene expression (RNA-seq) datasets. These included GTEx version 8 (ref. [78]), the Human Liver Cell Atlas[85], a single-cell dataset and a single-nucleus dataset from the human kidney[86,87], a single-cell dataset from the mouse kidney[88], a single-cell dataset from the human intestine[89] and a single-nucleus dataset from the kidneys of patients with CKD from the Kidney Precision Medicine Project (KPMP)[90]. Except for the KPMP, data sources and processing followed the workflow published by Cheng et al.[91]. KPMP data were downloaded from the KPMP Kidney Tissue Atlas repository at https://atlas.kpmp.org/repository as Seurat-format files and were subsequently processed in Seurat[92] similar to the other datasets. For generation of the top 10% highly expressed genes for each tissue and cell type in each dataset, we followed the workflow published by Schlosser et al.[5].

**GO, KEGG, tissue and cell type enrichment analyses**
Enrichment testing of the 282 identified genes was performed as follows. The number of independent SNPs per gene was computed using GCKD genotypes (PLINK version 1.90 (ref. [93])), and a database of Entrez gene identifiers based on org.Hs.eg.db version 3.8.2 was generated. Gene annotation included the number of independent SNPs per gene, gene length, GO terms[94] and KEGG pathways[95], as well as being Human Protein Atlas tissue or group enriched[96]; Human Protein Atlas cell type enhanced, enriched or group enriched[97]; being a VIP gene from PharmGKB (accessed 5 December 2020)[98]; being a gene with an actionable drug interaction from the Clinical Pharmacogenetics Implementation Consortium (levels A, A/B and B; accessed 13 January 2021)[99]; and being among the top 10% highly expressed genes in each GTEx version 8 tissue[78] and human[85–87,89,90] and murine cell types[88]. We performed 100 million random draws of an equal number of genes as contained in the respective source list (combined mQTLs, 282; plasma mQTLs, 214; urine mQTLs, 195; plasma-only mQTLs, 87; urine-only mQTLs, 68), matched for deciles of the number of independent SNPs and deciles of gene length and compared any overlap with cell types, tissues and terms with the ones identified for the original source list. Multiple-testing correction was performed using the Benjamini–Hochberg procedure[100].

Lastly, we tested for over-representation of certain phenotypes among mice in which the implicated genes had been genetically manipulated. The phenotype terms 'abnormal homeostasis' (MP:0001764) and 'abnormal metabolism' (MP:0005266), all of their child terms and all genes associated with these terms were downloaded from MouseMine[101] on 9 December 2021. Mouse genes were mapped to their human homologs using the getLDS function from the biomaRt package[102]. Human and mouse genes that did not map to a homolog in the respective other species were excluded from the analysis. This excluded 861 of 6,051 abnormal homeostasis genes, 61 of 952 abnormal metabolism genes and ten of 282 mGWAS genes (*PYCRL*, *GBA3*, *PPDPFL*, *CETP*, *NAT16*, *ZNF680*, *ENOSF1*, *ACSM6*, *FUT3*, *ZNF675*). The genes identified from urine, plasma or both were tested for over-representation among the genes belonging to each of the phenotype terms using Fisher's exact test (with the universe set to 13,151 genes, the number of mouse genes that mapped to human homologs in the Mouse Genome Informatics database), followed by Benjamini–Hochberg correction for multiple testing.

## Reporting summary

Further information on research design is available in the Nature Portfolio Reporting Summary linked to this article.

## Data availability

Genome-wide summary statistics are available through the NHGRI-EBI GWAS Catalog (GCST90264176–GCST90266872, https://www.ebi.ac.uk/gwas/).

## Code availability

We have clearly indicated each software whenever applicable and provided information on options (Methods and Reporting Summary).

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

## Acknowledgements

The work of Y.L. was supported by grant KO 3598/4-2 (to A.K.). The work of P. Schlosser was supported by the German Research Foundation (DFG) project ID 1050086601 (SCHL 2292/2-1, Walter Benjamin Fellowship) and the EQUIP Program for Medical Scientists, Faculty of Medicine, University of Freiburg. The work of S.H., M.W., P. Sekula, M.K., M.S. and A.K. was supported by the DFG project ID 431984000 (SFB 1453). The work of F. Kotsis and U.T.S. was supported by the German Federal Ministry of Education and Research (BMBF) within the framework of the e:Med research and funding concept (grant 01ZX1912B). The work of A.K., M.K. and M.S. was supported by Germany's Excellence Strategy (CIBSS, EXC-2189, project ID 390939984). The work of M.S. was supported by the ERC starting grant TREATCilia, grant agreement no. 716344. The work of P. Sekula was supported by the DFG SE 2407/3-1. The work of Y.C. was supported by the DFG SFB 1479 (project ID 441891347-S1). The work of M.E.G. and J.C. was supported by grant R01DK124399. Genotyping and urine metabolomics were supported by Bayer Pharma. This project has received funding from the Innovative Medicines Initiative 2 Joint Undertaking (JU) under grant agreement no. 115974. The JU receives support from the European Union's Horizon 2020 research and innovation program and the EFPIA and the JDRF. Any dissemination of results reflects only the authors' view; the JU is not responsible for any use that may be made of the information it contains. The GCKD study was and is supported by the BMBF (FKZ 01ER 0804, 01ER 0818, 01ER 0819, 01ER 0820 and 01ER 0821) and the KfH Foundation for Preventive Medicine. Unregistered grants to support the study were provided by corporate sponsors (listed at https://gckd.org). We are grateful for the willingness of the patients to participate in the GCKD study. The enormous effort of the study personnel of the various regional centers is highly appreciated. We thank the large number of nephrologists who provide routine care for the patients and collaborate with the GCKD study. The GCKD investigators are listed in the Supplementary Note. The ARIC study has been funded in whole or in part with federal funds from the National Heart, Lung, and Blood Institute, National Institutes of Health, Department of Health and Human Services, under contract nos. 75N92022D00001, 75N92022D00002, 75N92022D00003, 75N92022D00004 and 75N92022D00005. Funding was also supported by R01HL087641 and R01HL086694, National Human Genome Research Institute contract U01HG004402 and National Institutes of Health contract HHSN268200625226C. Infrastructure was partly supported by grant no. UL1RR025005, a component of the National Institutes of Health Roadmap for Medical Research. We thank the staff and participants of the ARIC study for their important contributions. The metabolite data at ARIC visit 5 were supported by R01 HL141824. We thank M. Meier for her support with data transfer.

## Author contributions

Design of this study: A.K., P. Schlosser. Recruitment for and management of the study: K.-U.E., A.K., F. Kotsis, H.M., U.T.S., B.Y., M.E.G., J.C., E.B. Genotyping: A.B.E., E.B. Metabolite quantification: N.S., P. Schlosser, E.D.K., R.P.M. Bioinformatics and statistical analysis: A.K., B.G., F.G.-C., I.S., M.W., N.S., P. Schlosser, P. Sekula, S.A., S.H., S.M.-M., Y.L. Interpretation of results: A.K., M.K., N.S., P. Schlosser, P. Sekula, S.A., S.H., Y.C., G.G., E.D.K., F. Kronenberg, M.S., M.A.H., P.J.O. Wrote the manuscript: A.K., N.S., P. Schlosser. Critically read and approved the manuscript: P. Schlosser, N.S., F.G.-C., S.H., S.M.-M., I.S., B.G., M.W., Y.C., A.B.E., G.G., E.D.K., F. Kotsis, J.M., M.F.G., M.K., F. Kronenberg, H.M., R.P.M., S.A., M.S., M.A.H., P. Sekula, U.T.S., K.-U.E., P.J.O., Y.L., A.K., B.Y., M.E.G., J.C., E.B.

## Competing interests

R.P.M. and E.D.K. are employees of Metabolon and, as such, have affiliations or financial involvement with Metabolon. J.M. is an employee of Bayer. All other authors declare no competing interests.

## Additional information

**Extended data** is available for this paper at https://doi.org/10.1038/s41588-023-01409-8.

**Correspondence and requests for materials** should be addressed to Pascal Schlosser or Anna Köttgen.

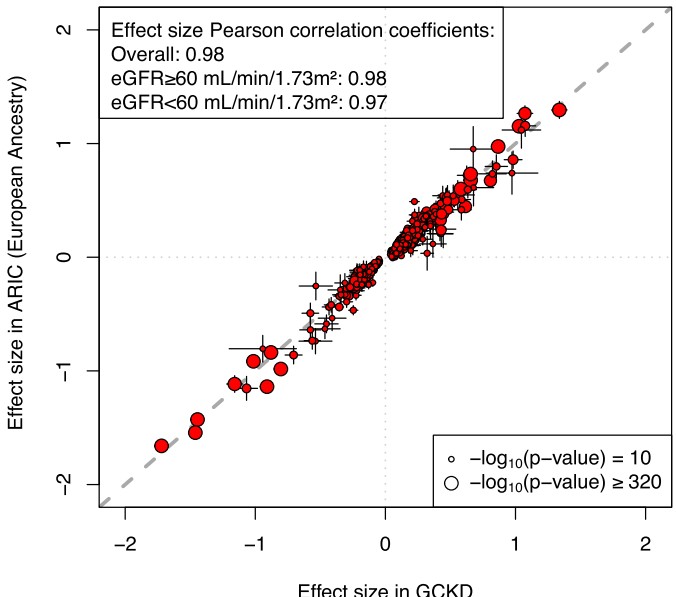

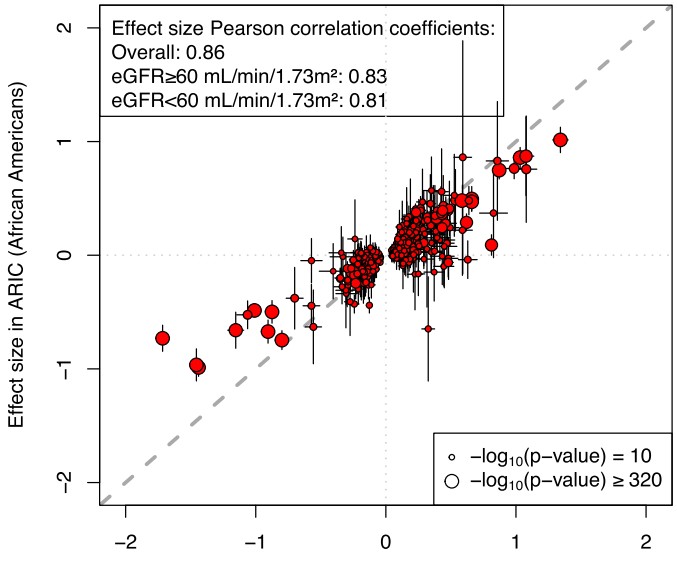

**Extended Data Fig. 1 | Evaluation of genetic associations of plasma mQTLs from CKD patients in a multi-ethnic, population-based sample.** Each point represents the index SNP of one of 459 (EA) and 430 (AA) associations that could be matched between the Metabolon platforms of the GCKD and ARIC studies (see Supplementary Table 6). Data are presented as effect size estimate +/- 1.96x standard errors in each study and the dot size is proportional to the two-sided -log$_{10}$(P-value) in GCKD (N$_{GCKD}$ = 4960, N$_{ARIC EA}$ = 3603, N$_{ARIC AA}$ = 818).

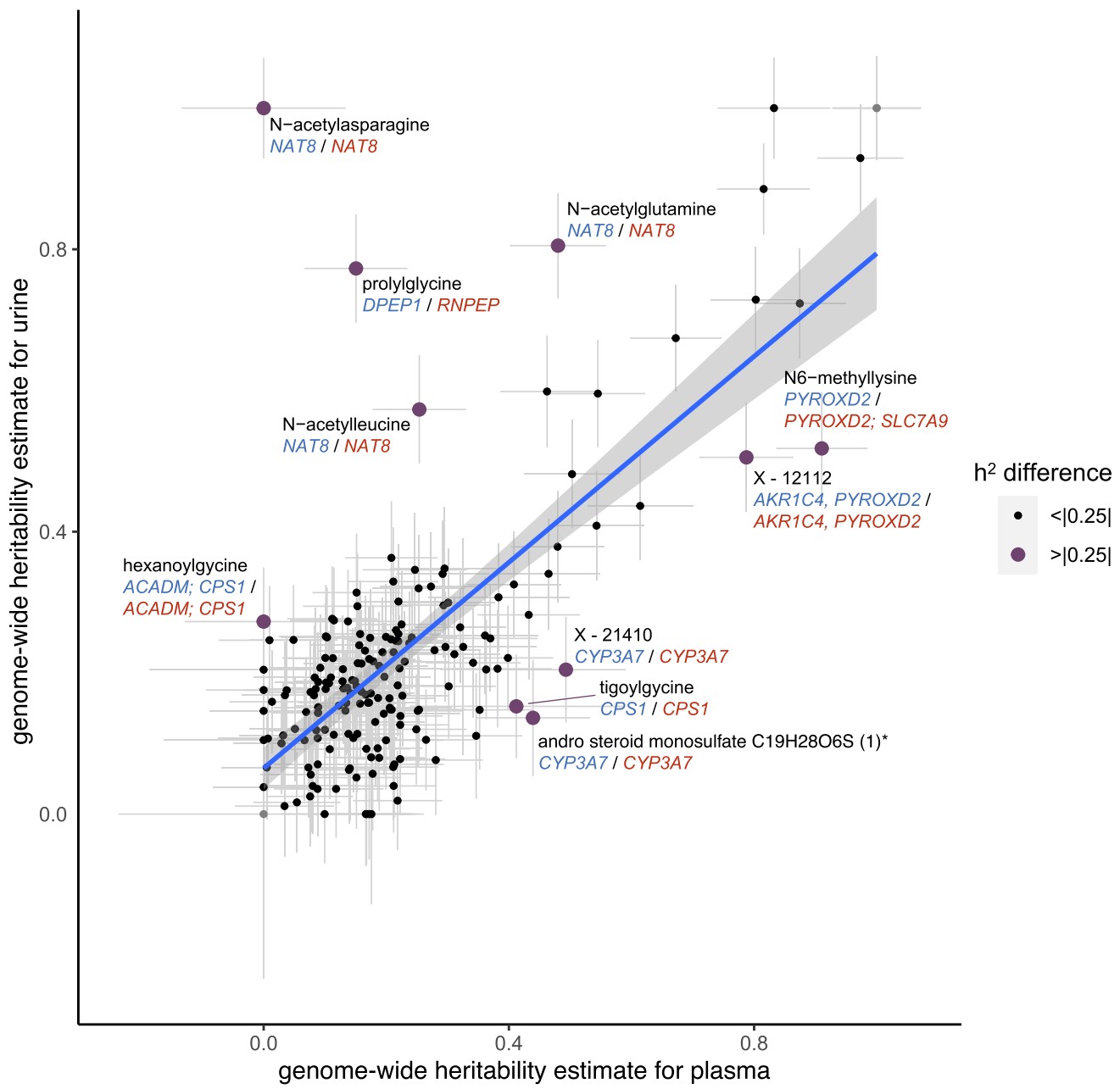

**Extended Data Fig. 2 | Comparison of the heritability for 184 matched plasma and urine metabolites with at least one mQTL.** The positive correlation between the estimated heritabilities for a given metabolite's plasma and urine levels is consistent with the metabolites' filtration from plasma to urine, without substantial additional genetic influences on their tubular handling. The blue line is the linear regression line and the gray shaded area represents the 95%-confidence interval. Differences in estimated heritability for plasma and urine (instances with >25% are labeled with the associated metabolite and most likely gene; error bars represent h² variance) can contain interesting biological information: for example, three metabolites with larger estimated heritabilities in urine than in plasma are N-acetylated amino acids, all of which have an mQTL at *NAT8*. *NAT8* is highly and selectively expressed in the kidney, where the encoded enzyme N-acetylates molecules to make them water soluble for subsequent excretion.

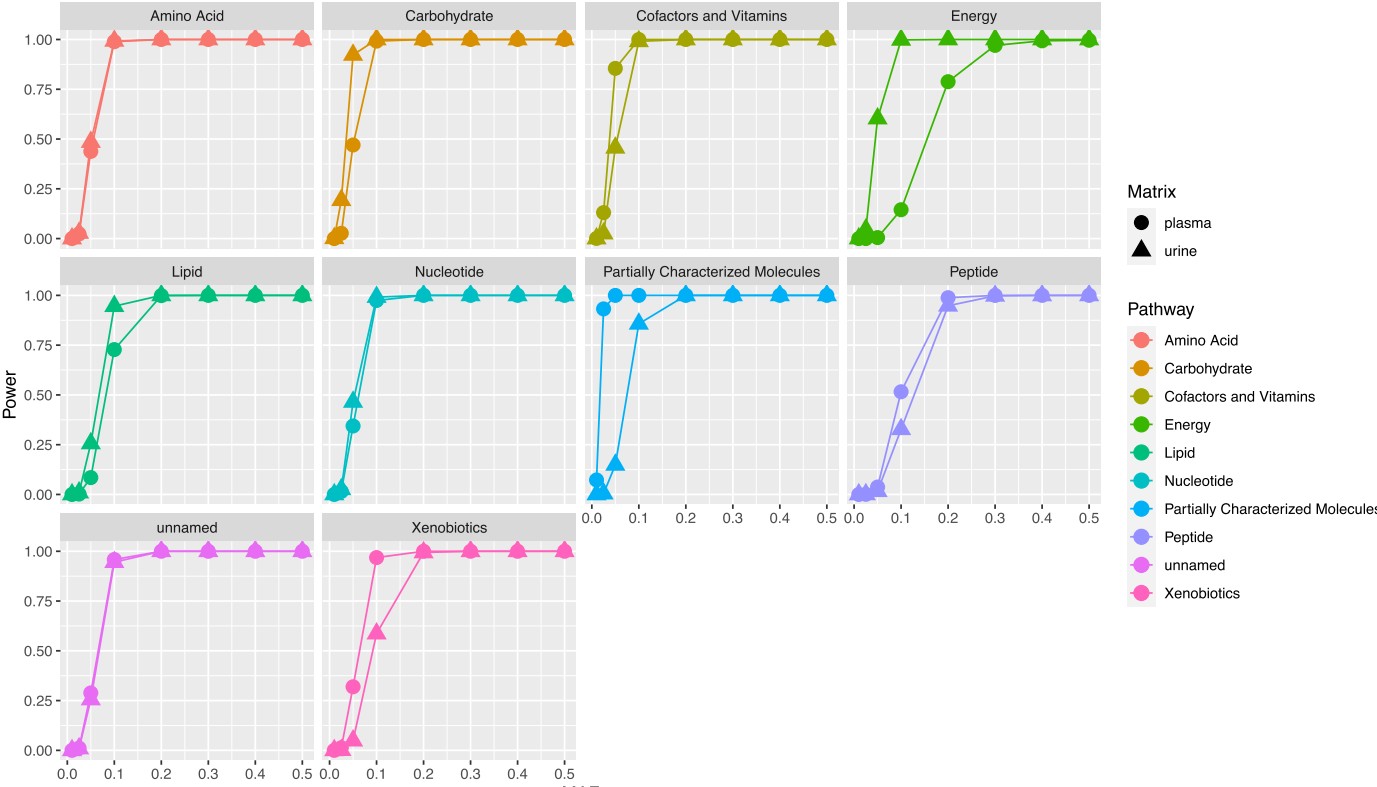

**Extended Data Fig. 3 | Post-hoc power analyses for plasma and urine mQTLs by metabolite super-pathway.** Power analyses are based on a sample size of 5,000, the genome-wide statistical significance thresholds used in our study, and are conducted across a range of minor allele frequencies. For each matrix-super-pathway subgroup, the median observed effect size across mQTLs as well as the median standard deviation of the metabolites with an mQTL within the group were used.

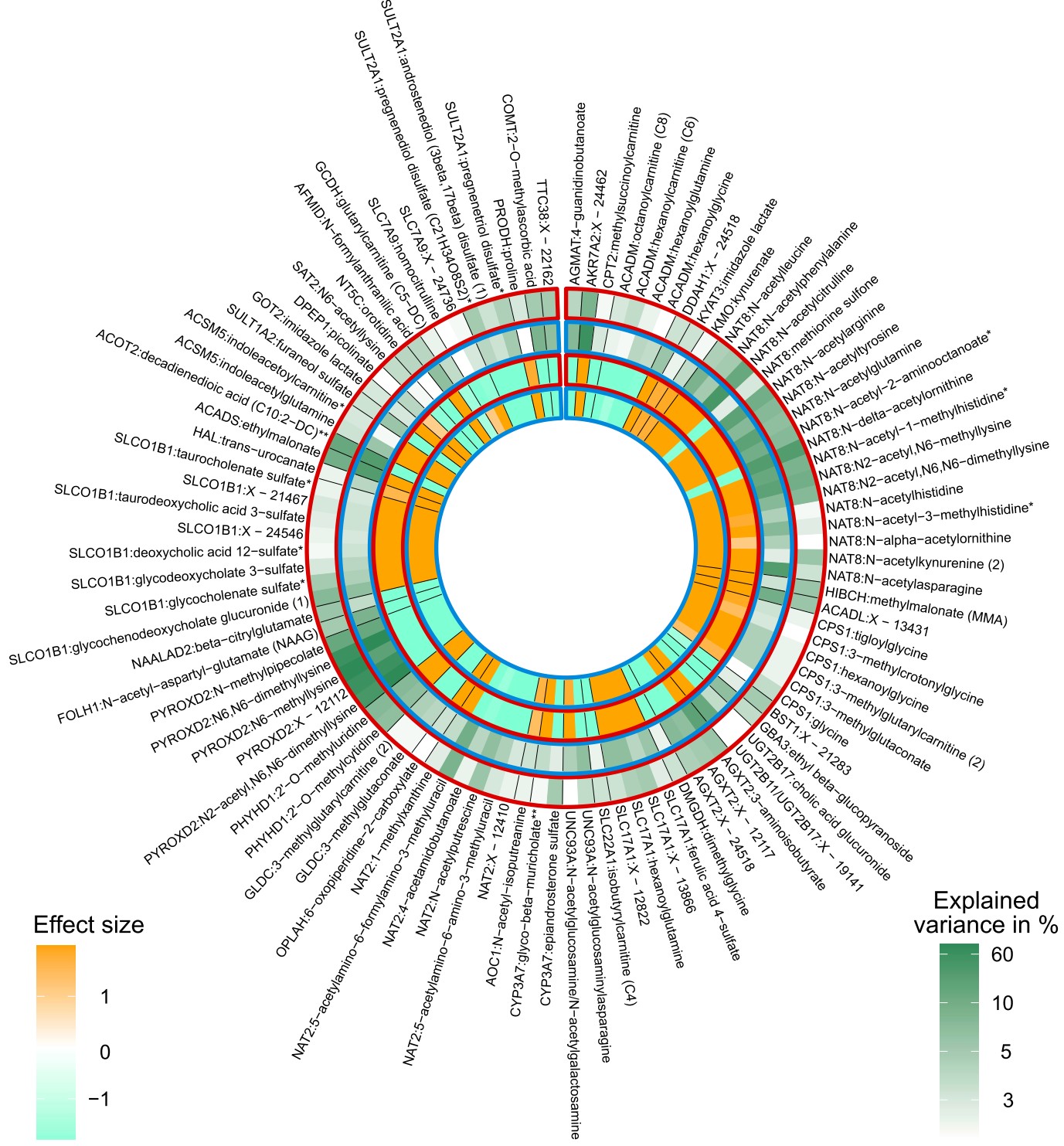

**Extended Data Fig. 4 | Comparison of direction of genetic associations and explained variance at inter-matrix mQTLs.** Comparison of effect sizes and explained variance for colocalization signals for mQTLs detected for the same metabolite in both plasma and urine (N = 204; only the 99 mQTLs for which the explained variance in metabolite levels in at least one of both matrices is >3% are shown). The two inner bands represent the effect size of the mQTL in plasma (framed in red) and urine (framed in blue). Shades of orange indicate positive effect sizes, shades of aquamarine negative ones. The two outer bands represent the variance in metabolite levels in plasma and urine explained by the index SNP of the corresponding mQTL, where a darker shade of green corresponds to a greater explained variance.

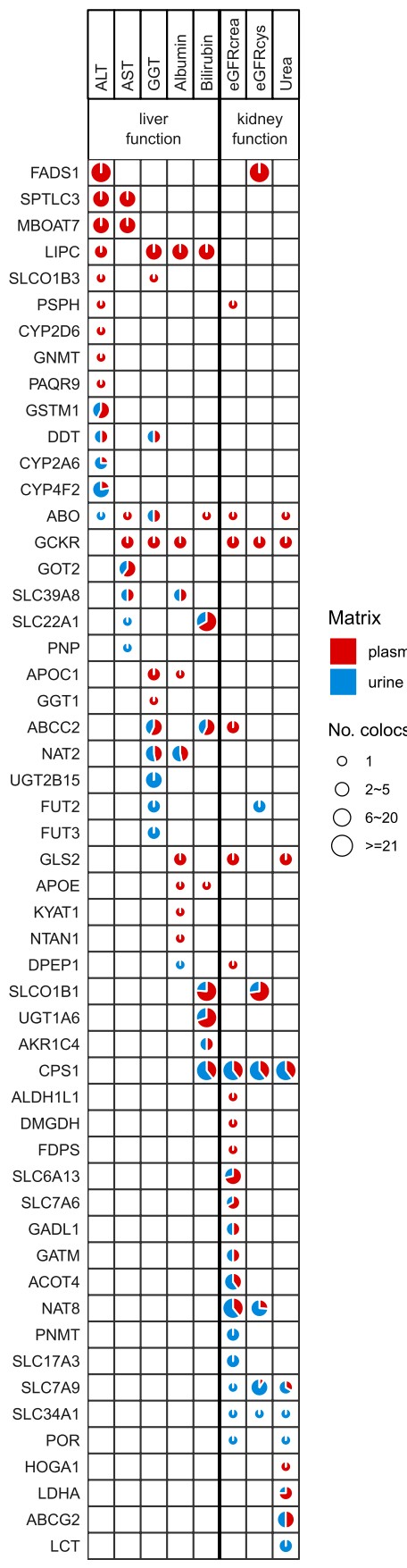

**Extended Data Fig. 5 | See next page for caption.**

**Extended Data Fig. 5 | Colocalization of mQTLs with selected clinical markers of kidney and liver function.** The mQTLs are represented by the implicated genes on the rows and the colocalized clinical markers are on the columns. Liver function markers include alanine aminotransferase (ALT), aspartate aminotransferase (AST), gamma glutamyltransferase (GGT), albumin and bilirubin. Kidney function markers include eGFRcrea, eGFRcys and urea. The size of pie represents the total number of colocalizations grouped into four categories. The slices in each pie colored in red and blue represent the proportion of colocalizations of plasma and urine mQTLs with the respective clinical markers.

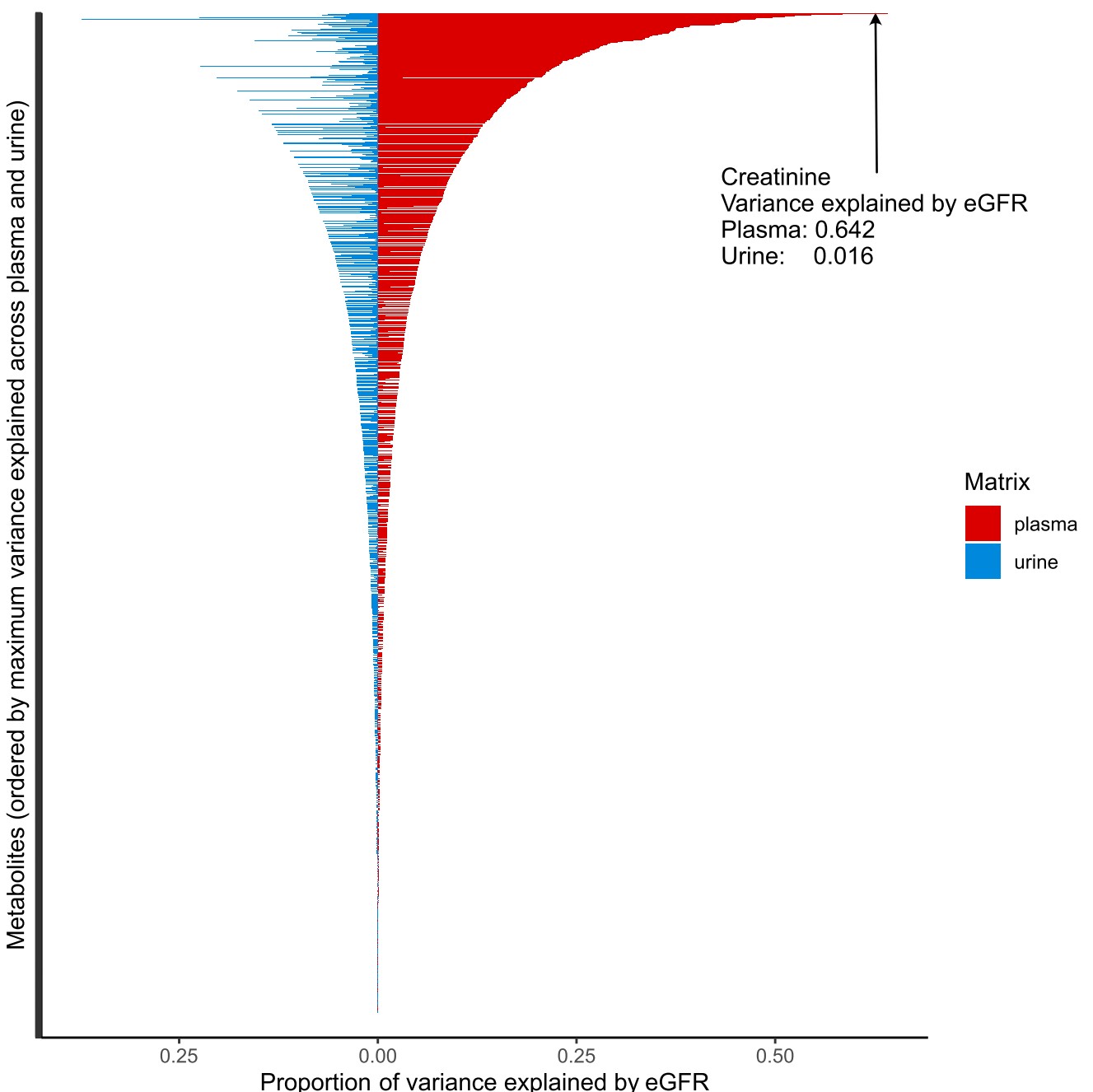

Creatinine
Variance explained by eGFR
Plasma: 0.642
Urine: 0.016

**Matrix**

plasma

urine

**Extended Data Fig. 6 | Proportion of metabolite variance explained by eGFR.** The proportion of a metabolite's variance explained by eGFR is represented on the x-axis. All metabolites quantified from plasma and urine are shown along the y-axis, ordered by the maximum variance explained across plasma (red color) and urine (blue color). The metabolite with the largest amount of variance explained by eGFR was plasma creatinine.

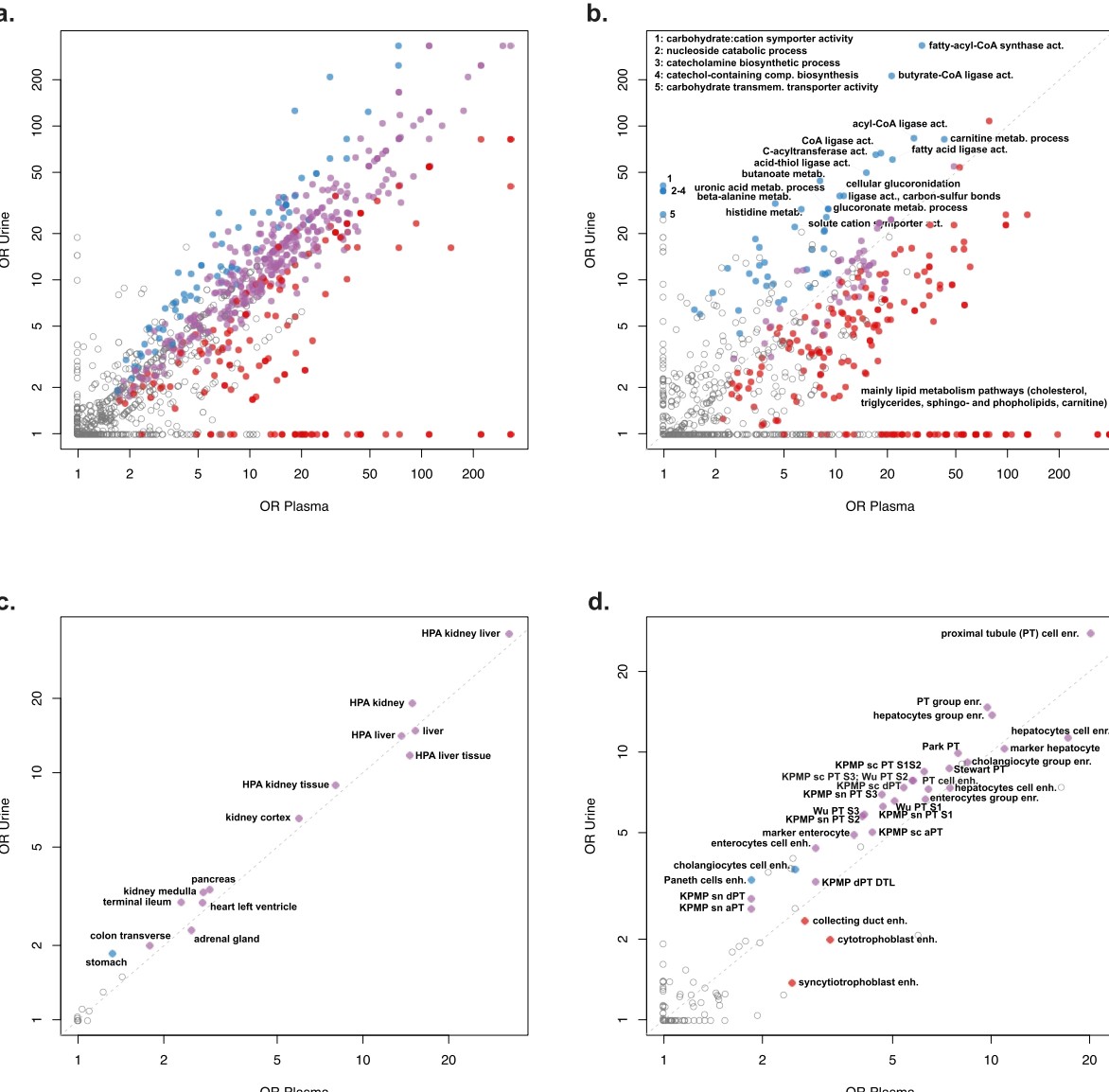

**Extended Data Fig. 7 | Enrichment of mQTL-related genes among GO terms, KEGG pathways, tissues, and cell types. (a**) Similarities and differences between terms and pathways enriched for genes identified by all plasma vs. all urine mQTLs; **(b)** mQTLs exclusively identified in plasma and urine; **(c)** between tissues enriched for genes identified by all plasma vs. all urine mQTLs, and **(d)** between cell types enriched for genes identified by all plasma vs. all urine mQTLs. Terms significantly (adjusted P-value < 0.05) enriched for genes identified by mQTLs from only one matrix are colored in red and blue respectively and terms significantly enriched for genes from both matrices are colored in purple. OR: odds ratio.

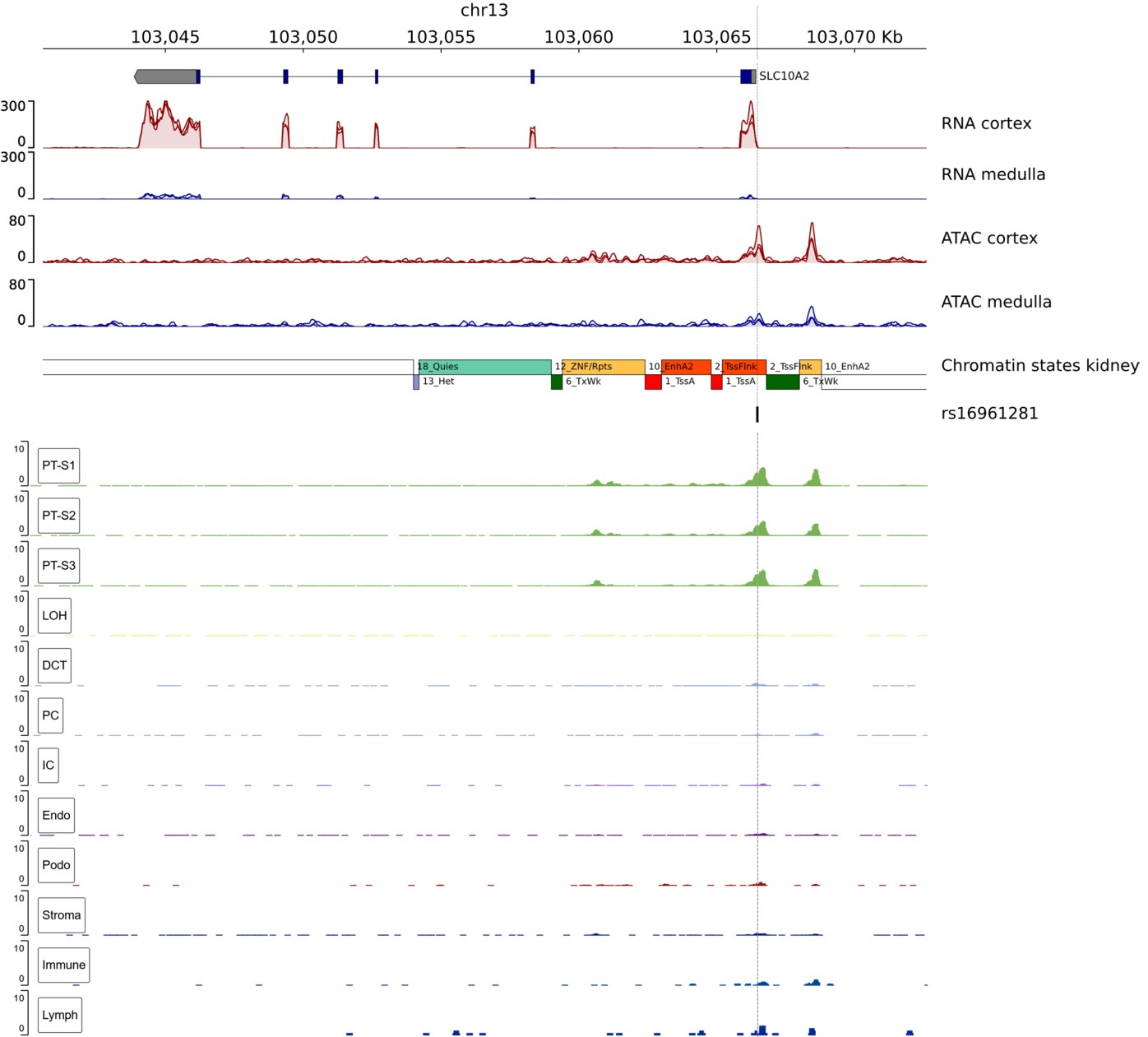

**Extended Data Fig. 8 | Extended view of the *SLC10A2* region.** The upper part of the figure shows the same RNA-seq, ATAC-seq, chromatin state and histone ChIP-seq tracks as Fig. 6. The RNA-seq and ATAC-seq tracks show the overlayed signal from tissue of three different donors. The index SNP rs16961281, that is associated with urine glycocholate, is located at the vertical dashed line. The bottom part shows publicly available single nucleus (sn)ATAC-seq data for different kidney cell types, which was derived from primary human kidney samples[38]. The position of rs16961281 is nearly exclusively accessible in cells of all proximal tubule segments (PT-S1, PT-S2, PT-S3). PTs are the predominant cell type in kidney cortex, underscoring the consistency of the snATAC-seq data and the bulk ATAC-seq data. Other cell types shown include: Endothelial cells (Endo), podocytes (Podo), loop of Henle cells (LOH), distal convoluted tubule cells (DCT), collecting duct principal cells (PC), collecting duct intercalated cells (IC), stroma cells (Stroma), immune cells (Immune), lymph cells (Lymph).

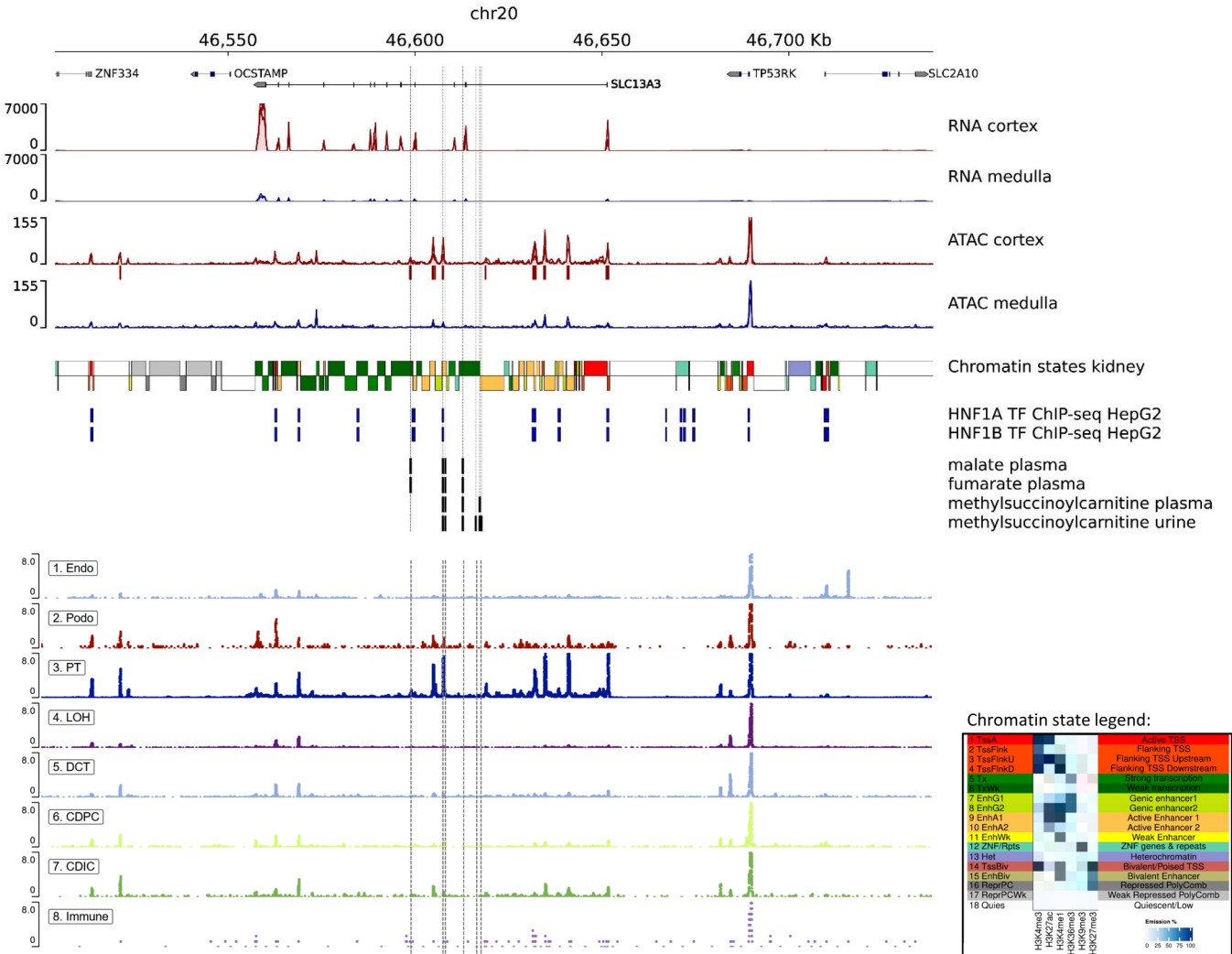

**Extended Data Fig. 9 | Extended view of the *SLC13A3* region.** The upper part of the figure shows the same RNA-seq, ATAC-seq, chromatin state and histone ChIP-seq tracks as Fig. 7. The index SNP rs6124828, that is associated with malate, fumarate, and methylsuccinoylcarnitine in plasma as well as with methylsuccinoylcarnitine in urine is located at the second vertical dashed line from the left. The bottom part shows single nucleus (sn)ATAC-seq data for different kidney cell types, which was derived from primary human kidney samples[39]. The position of rs6124828 is nearly exclusively accessible in proximal tubule cells (PT). PTs are the predominant cell type in the kidney cortex, underscoring the consistency of the snATAC-seq data and the bulk ATAC-seq data. Other cell types shown include: Endothelial cells (Endo), podocytes (Podo), loop of Henle cells (LOH), distal convoluted tubule cells (DCT), collecting duct principal cells (CDPC), collecting duct intercalated cells (CDIC), immune cells (Immune).

# Reporting Summary

## Statistics

For all statistical analyses, confirm that the following items are present in the figure legend, table legend, main text, or Methods section.

| n/a | Confirmed | |
|---|---|---|
| ☐ | ☒ | The exact sample size ($n$) for each experimental group/condition, given as a discrete number and unit of measurement |
| ☒ | ☐ | A statement on whether measurements were taken from distinct samples or whether the same sample was measured repeatedly |
| ☐ | ☒ | The statistical test(s) used AND whether they are one- or two-sided *Only common tests should be described solely by name; describe more complex techniques in the Methods section.* |
| ☐ | ☒ | A description of all covariates tested |
| ☐ | ☒ | A description of any assumptions or corrections, such as tests of normality and adjustment for multiple comparisons |
| ☐ | ☒ | A full description of the statistical parameters including central tendency (e.g. means) or other basic estimates (e.g. regression coefficient) AND variation (e.g. standard deviation) or associated estimates of uncertainty (e.g. confidence intervals) |
| ☐ | ☒ | For null hypothesis testing, the test statistic (e.g. $F$, $t$, $r$) with confidence intervals, effect sizes, degrees of freedom and $P$ value noted *Give P values as exact values whenever suitable.* |
| ☐ | ☒ | For Bayesian analysis, information on the choice of priors and Markov chain Monte Carlo settings |
| ☒ | ☐ | For hierarchical and complex designs, identification of the appropriate level for tests and full reporting of outcomes |
| ☐ | ☒ | Estimates of effect sizes (e.g. Cohen's $d$, Pearson's $r$), indicating how they were calculated |

*Our web collection on statistics for biologists contains articles on many of the points above.*

## Software and code

Policy information about availability of computer code

| Data collection | Data preparation, quality control and statistical analyses of the data presented in this manuscript were performed at the Institute of Genetic Epidemiology, Medical Center - University of Freiburg, Freiburg (Germany) and the Department of Epidemiology, Johns Hopkins University, Baltimore, MD, USA. |
|---|---|
| Data analysis | - Software-tools for genotype imputation:<br>    Michigan Imputation Server (incl. minimac3 v2.0.1, Eagle v2.3)<br>- Software-tools for association analyses: in-house pipeline based on SNPTEST v2.5.2<br>- Software-tools for postprocessing of results: LocusZoom v1.3, Circos v0.69-6, R-package gtx v2.1.6, Netboost v1.0.0, Seurat v2.3.4, GCTA v1.91.6 beta, pyGenomeTracks v3.7, locuscompare v1.0.0, R package MendelianRandomization (v0.6.0), PLINK v1.90<br>- Data bases, publicly available: SNiPA v3.3 (incl. CADD version 1.3, Ensembl VEP tool), NephQTL browser, GTEx V8, Ensembl Biomart, GO, KEGG, GeneAtlas (UK Biobank project), PhenoScanner V2<br>- Miscellaneous: R v3.5.3<br>References or website addresses are provided in manuscript. |

For manuscripts utilizing custom algorithms or software that are central to the research but not yet described in published literature, software must be made available to editors and reviewers. We strongly encourage code deposition in a community repository (e.g. GitHub). See the Nature Portfolio guidelines for submitting code & software for further information.

## Data

Policy information about availability of data

All manuscripts must include a data availability statement. This statement should provide the following information, where applicable:
- Accession codes, unique identifiers, or web links for publicly available datasets
- A description of any restrictions on data availability
- For clinical datasets or third party data, please ensure that the statement adheres to our policy

> Genome-wide summary statistics are available through the NHGRI-EBI GWAS Catalog (GCST90264176-90266872, https://www.ebi.ac.uk/gwas/).

# Field-specific reporting

Please select the one below that is the best fit for your research. If you are not sure, read the appropriate sections before making your selection.

☒ Life sciences   ☐ Behavioural & social sciences   ☐ Ecological, evolutionary & environmental sciences

For a reference copy of the document with all sections, see nature.com/documents/nr-reporting-summary-flat.pdf

# Life sciences study design

All studies must disclose on these points even when the disclosure is negative.

| | |
|---|---|
| Sample size | We included all 5023 GCKD participants with available plasma or urine metabolite quantification. A previous analysis of a subsample of 1627 GCKD participants with urine metabolite measurements has identified 240 mQTLs. |
| Data exclusions | Metabolites were excluded for high proportions of missingness (>80%). Samples were excluded if no genotypes were available. This is clearly described in the methods. |
| Replication | We evaluated reproducibility of our findings by three means :<br>- We assessed the presence and correlation of genetics effects using data from seven large studies of common variants and the plasma/serum metabolome as quantified with the Metabolon assay in European ancestry populations that were published in peer-reviewed journals, to maximize overlap with our findings. We observed excellent consistency of effect directions and validation rates, as outlined in the article.<br>- For urine mQTLs, 98.2% of 226 published urine mQTLs showed significant P-values (<5×10-8/226) and consistent effect direction.<br>- Independent replication testing was performed using genetic and plasma metabolome data from 3,603 European ancestry and 818 African American participants of the population-based Atherosclerosis Risk in Communities (ARIC) study (Methods). The results are described in the manuscript and include high replication rates for plasma mQTLs (94%) in the larger European ancestry sample. |
| Randomization | Not relevant to this study because this is an observational study |
| Blinding | Not relevant to this study because this is an observational study |

# Reporting for specific materials, systems and methods

We require information from authors about some types of materials, experimental systems and methods used in many studies. Here, indicate whether each material, system or method listed is relevant to your study. If you are not sure if a list item applies to your research, read the appropriate section before selecting a response.

### Materials & experimental systems

| n/a | Involved in the study |
|---|---|
| ☒ | ☐ Antibodies |
| ☒ | ☐ Eukaryotic cell lines |
| ☒ | ☐ Palaeontology and archaeology |
| ☒ | ☐ Animals and other organisms |
| ☐ | ☒ Human research participants |
| ☐ | ☒ Clinical data |
| ☒ | ☐ Dual use research of concern |

### Methods

| n/a | Involved in the study |
|---|---|
| ☒ | ☐ ChIP-seq |
| ☒ | ☐ Flow cytometry |
| ☒ | ☐ MRI-based neuroimaging |

## Human research participants

Policy information about studies involving human research participants

| | |
|---|---|
| Population characteristics | Please see Supplementary Table 1.<br>Characteristic Overall (N=5,023) |

Female sex 39.9% (2003)
Hemoglobin A1C (mmol/mol) 45.78 (11.21)
Age (years) 60.09 (11.97)
eGFRcr (mL/min/1.73m²) 49.45 (18.13)
Urinary albumin-to-creatinine ratio (mg/g) 50.52 [9.55;386.19]
Systolic blood pressure (mm Hg) 139.45 (20.35)
BMI (kg/m²) 29.8 (5.95)
Albumin (g/l) 38.34 (4.44)

| | |
|---|---|
| Recruitment | The GCKD study is an ongoing prospective observational cohort study of participants with CKD. Between 2010 and 2012, 5,217 adult persons with CKD under regular care by nephrologists provided written informed consent and were enrolled into the study at nine participating study centers across Germany (Aachen, Berlin, Erlangen, Freiburg, Hannover, Heidelberg, Jena, München, Würzburg). Participants could not self-select into the study, but it cannot be excluded that eligible persons with many or severe comorbidities were less likely to participate than other eligible participants. For this project, all participants with available plasma or urine collected at the baseline visit were selected (N=5,023). |
| Ethics oversight | The GCKD Study was registered in the national registry for clinical studies (DRKS 00003971) and approved by all local ethic committees of the participating centers (Universities or Medical Faculties of Aachen, Berlin, Erlangen, Freiburg, Hannover, Heidelberg, Jena, München, Würzburg). |

Note that full information on the approval of the study protocol must also be provided in the manuscript.

# Clinical data

Policy information about clinical studies

All manuscripts should comply with the ICMJE guidelines for publication of clinical research and a completed CONSORT checklist must be included with all submissions.

| | |
|---|---|
| Clinical trial registration | This study is an observational study (DRKS 00003971). |
| Study protocol | The study protocol and design has been published (PMID: 21862458). |
| Data collection | Data was collected during GCKD study visits by trained personnel following a published pre-specified protocol and standard operating procedures, and captured with the software Askimed (https://www.askimed.com/). The participants are currently followed for clinical outcomes for more than 10 years. |
| Outcomes | The primary outcomes of this study were metabolite levels, which was defined before study initiation by the authors. Non-targeted MS analysis was performed at Metabolon, Inc. from plasma and urine samples collected at the study's baseline visit, as described in detail in the publication. |

