## [Peer Review File · Nature Genetics]

Peer Review Information

Manuscript Title: Genetic studies of paired metabolomes reveal enzymatic and transport processes at the interface of plasma and urine

Corresponding author name(s): Dr Pascal Schlosser Professor Anna Köttgen

Reviewer Comments & Decisions:

Decision Letter, initial version:
--

15th Jul 2022

Dear Professor Kottgen,

Your Article, "Genetic studies of paired metabolomes reveal enzymatic and transport processes at the interface of plasma and urine" has now been seen by 3 referees. You will see from their comments below that while they find your work of interest, some important points are raised. We are interested in the possibility of publishing your study in Nature Genetics, but would like to consider your response to these concerns in the form of a revised manuscript before we make a final decision on publication.

To guide the scope of the revisions, the editors discuss the referee reports in detail within the team with a view to identifying key priorities that should be addressed in revision. In this case, we think all three referees have provided constructive reviews aimed at strengthening the analyses and improving the presentation, and we particularly ask that you address their technical comments as thoroughly as possible with appropriate revisions. Please do not hesitate to get in touch if you would like to discuss these issues further.

We therefore invite you to revise your manuscript taking into account all reviewer and editor comments. Please highlight all changes in the manuscript text file. At this stage we will need you to upload a copy of the manuscript in MS Word .docx or similar editable format.

*2) If you have not done so already please begin to revise your manuscript so that it conforms to our Article format instructions, available [here](http://www.nature.com/ng/authors/article_types/index.html). Refer also to any guidelines provided in this letter.

[redacted]

We hope to receive your revised manuscript within three to six months. If you cannot send it within this time, please let us know.

Sincerely,
Wei

Wei Li, PhD
Senior Editor
Nature Genetics
New York, NY 10004, USA

www.nature.com/ng

Reviewers' Comments:

Reviewer #1:
Remarks to the Author:

Thank you for the opportunity to review this paper which presents genome-metabolome-wide analyses of plasma and urine in the same individuals, all part of the German Chronic Kidney Disease (GCKD) study.

Overall, this is a clearly written paper with sound, standard methodology that raises no major concerns.

The paper falls short in that it lacks sufficient novelty. Much larger and more comprehensive analyses exist for the plasma metabolome. It is therefore disappointing that a) large parts of the plasma elements of this paper are known, and b) the urine part includes a large subset of the same participants that were part of the previous urine focused analysis published by the same group. Thresholds used for establishing 'replication' and novelty appear chosen in favour of this study, rather than reflecting the number of comparisons performed and acknowledging the contribution of the body of previous work.

The genome-metabolome-wide analysis (of blood and urine) remains rather descriptive and is written quite linearly (a bit like a counting exercise), which makes the paper quite long and not the most interesting read.

The authors expand on specific examples, which is a strength of this paper, but possibly more of interest to a specialist nephrology audience.

The comparative element of this paper is its main feature but is restricted by the power that comes from the number of people with paired samples, which is not necessarily required for these comparative genetic analyses. Since the main contribution of this paper lies in the comparative part, it is a limitation that this is outdated given larger plasma analysis already underway and unnecessarily restricted by the power based on paired samples only.

The fact that a subset of genetic associations is specific or restricted to urine versus plasma is neither surprising nor novel but largely reflects known biology/ biochemistry. Specificity of metabolites being present and detectable in plasma versus urine, respectively, is a key driver of (groups of related) matrix specific metabolites and associated mQTLs, due to the glomerular filtration barrier – something the authors acknowledge. Related to this, the large number of observed colocalizations is driven by the fact that the authors included markers of kidney and liver function.

The follow-up and examination of shared genetic drivers of urine vs plasma-specific mQTLs is a valuable part of this work, and it might be useful to be more focused on relevant diseases only and complications of CKD rather than being too explorative without very clear translational gain.

The authors are generally very clear and careful in their wording, nevertheless some of the statements appear to jump to conclusions too quickly. For example, while the FUT2 section is solid, the suggestion of using urine galactosylglycerol and 1,6-anhydroglucose as surrogates of cardiometabolic risk is unsubstantiated by the existing evidence.

Reviewer #2:

Remarks to the Author:

A very comprehensive and well-conducted analysis of GWAS of plasma and urine metabolites in patients with chronic kidney disease with a series of interesting downstream analyses and exciting illustrative case studies. The paper is very well-written but due to inherent complexity of this ambitious undertaking certain parts are perhaps not immediately easy to follow and may require additional explanations as per information with comments to authors. Overall, a very important contribution to our understanding of genetic determinants of plasma and urine metabolites.

1. When intersecting GWAS with QTL-like datasets, the identity of specific genes that mediate the association between genetic variants and the phenotype (i.e., plasma level of metabolite or urine level of metabolite) is of utmost importance. The authors used different sources of information including colocalisation with eQTL input from multiple tissues (GTEx) to prioritise the most "likely" driver genes. It does appear that (i) in many cases different genes came up in eQTL analyses conducted in different tissues and (ii) tissues were not prioritised, i.e. input from kidney eQTLs were not treated preferentially i.e. when it came to analysis of urine metabolites, (iii) in some cases the input from eQTL and pQTL may have been overwritten and a different gene assigned ["In case of a clear biological fit to another scored gene (i.e., corresponding monogenic disease or animal model), the prioritized gene was reassigned"], (iv) there were differences in the input between eQTL and blood pQTL analysis. It does appear that in most cases, there was one gene "most likely" mediating the association with plasma and urine metabolites. Could it be that different genes may be mediating the uncovered associations in different tissues (mQTL, eQTL are known to operate in a tissue specific manner – see for example PMID: 30467309). This deserves some contemplation in the discussion. Should the input from kidney/liver eQTL have a preference in this analysis (or at least in the analysis of urine metabolites (kidney))?
2. I think it is important to provide the list of 282 genes in relation to each metabolite in a dedicated table. How many of these genes were non-coding?
3. Have the authors considered using MR to provide further validation of the identified 282 genes as the drivers of the identified GWAS associations? At least for urine, an input from kidney eQTL could be used as the source of information on "exposure" (i.e., gene expression)?
4. Was there an enrichment for kidney-related phenotypes in the analysis of urinary mQTLs and 4,422 biomarkers and diseases in UK Biobank?
5. Would it be possible to use this rich dataset to see which blood and urinary metabolites show the strongest associations with the biochemical measure of kidney function? Would it be possible to comment on the causality of the detected associations?
6. Suppl Figures 1 and 2 did not display well in the Supplement.
7. Is mQTL an appropriate abbreviation of metabolite QTLs (eg. vs methylation QTL)?

Reviewer #3:

Remarks to the Author:

Schlosser et al examine the genetic basis of an important biological phenomenon, glomerular filtration, though paired examination of metabolite levels in serum and urine in CKD patients. While the incredibly extensive results are very interesting and the dataset has significant potential, the current analysis is somewhat limited in terms of broad effects on the genome wide association landscape (specifically, using tools that examine effects below genome wide significance). Some specific comments below:

Please put the exact N in (5023?) on line 104.

How does the heritability differ for matched serum and urine metabolites?

What is the distribution of genetic correlations across metabolites? This would enhance the paragraph starting line 162.

Please clarify on line 130 that this is *across metabolites* (I assume).

The suggestion behind the statement on line 153 is intriguing, but it is not actually tested in the manuscript. Please evaluate this claim or remove it.

On line 157, were mQTLs associated with multiple metabolites more or less likely to fine map to single SNPs? And what about by allele frequency?

In the paragraph starting on line 162, it's unclear how much the starting and specificity are driven by global differences in power or the pathways that these substrate specific metabolites belong to. These global comparisons would greatly aid interpretation by the reader and should be included.

The paragraph starting on line 195 confuses me. As these highly pleiotropic signals are counted multiple times in the list of pairwise colocalizations, then please include the number of distinct mQTLs underlying the lines 190-194 as well. Am I understanding your implication correctly that pleiotropy is evidence of these genes having key functionality? I would provide additional evidence for this.

Please provide the effect size and p value for both serum and urine on line 234.

Please provide your evidence for SLC10A2 being the causal gene on line 229.

On line 241, is egfrcrea from your analysis of UKBB or from Stanzick et al.? Do these agree? Do they differ from using creatinine directly?

The paragraph starting on line 240 takes a nice approach for evaluating shared signals with liver function traits, but misses the potential to consider regulatory annotations. Using tools like stratified LDSC, do you see evidence of this corresponding serum vs urine enrichments in kidney vs liver? While there is some evidence this might be the case, the manuscript and dataset provide a really nice opportunity to quantify this phenomenon and support the results here and in SF5.

For line 256, please provide evidence that there is an enrichment in the target traits across your 115 targets at the mQTLs themselves, not just at the genes they putatively target.

What do the associations look like for non-urine-specific (or even serum specific, and clinical tests) metabolites at rs601338? And where is FUT2 expressed? Given the disease outcome associations presented, it would be quite interesting if these two metabolites were the only associations.

Similarly, what does the phewas look like at rs62542743? There are previously reported associations

at AQP7, though your proposed mechanism seems quite plausible.

The SLC13A3 story is very nice. Do you see experimental evidence of allele specific binding at rs6124828 in heterozygous individuals, either in HNF ChIP or ATAC/histone chip? Is there evidence of epistasis with variants at HNF genes?

Suggestion to run mendelian randomization to support the (very neat!) results in the paragraph starting on line 393. Can you use other associations with these metabolites genome wide to tease out which might be causal (or is that what you did on line 398? I was confused)? That would likely simplify 7d as well, which I had trouble understanding.

I would consider running GWAS for the ratio of urine to serum levels of each metabolite as well, which should pinpoint effects on the kidneys even further.

The statement on line 469 only applies to urine; these results should be extended to serum in this manuscript (which I, for one, would find useful) or the statement should be clarified.

Author Rebuttal to Initial comments

Point-by-point response

Editorial comments:

To guide the scope of the revisions, the editors discuss the referee reports in detail within the team with a view to identifying key priorities that should be addressed in revision. In this case, we think all three referees have provided constructive reviews aimed at strengthening the analyses and improving the presentation, and we particularly ask that you address their technical comments as thoroughly as possible with appropriate revisions. Please do not hesitate to get in touch if you would like to discuss these issues further.

Response: We thank the Editors for their guidance, and agree that all three Reviewers have provided constructive reviews. As suggested, we have thoroughly addressed the technical comments through the conduct of additional analyses, as outlined below. We believe that the results are interesting and significant additions to our work, and we thank the Reviewers and the Editorial Board for giving us the opportunity to make these revisions.

Reviewer #1:

Thank you for the opportunity to review this paper which presents genome-metabolome-wide analyses of plasma and urine in the same individuals, all part of the German Chronic Kidney Disease (GCKD) study.

Overall, this is a clearly written paper with sound, standard methodology that raises no major concerns.

1. The paper falls short in that it lacks sufficient novelty. Much larger and more comprehensive analyses exist for the plasma metabolome. It is therefore disappointing that a) large parts of the plasma elements of this paper are known, and b) the urine part includes a large subset of the same participants that were part of the previous urine focused analysis published by the same group. Thresholds used for establishing 'replication' and novelty appear chosen in favour of this study, rather than reflecting the number of comparisons performed and acknowledging the contribution of the body of previous work.

Response: The aim of our study was to detect and compare genetic associations with metabolite levels across plasma and urine, and to thereby gain insights not only into systemic metabolism, but also into functions of the kidney at the interface of plasma and urine. To minimize differences introduced by variation in measurement technology, coverage, analytical choices including transformations and covariate adjustments, imputation reference panels, as well as differences between study samples, we therefore generated such data from paired plasma and urine samples, and analyzed it with a consistent workflow. We believe that this is a unique feature of our study, such that any differences in genetic associations with metabolites between plasma and urine, for example as provided for *AQP7*, can be interpreted confidently. Moreover, even if associations of a metabolite in plasma have been reported previously, its study in paired samples permits additional insights. For example, Shin *et al*¹ previously reported associations at the *SLC7A9* locus and plasma/serum homocitrulline from ~8,000 persons of European ancestry. Our study confirms the direction and significance of this association in plasma, but also adds a novel, inverse association with urine homocitrulline levels. Since *SLC7A9* reabsorbs dibasic amino acids from urine at the apical membrane of kidney tubular epithelial cells, the observed inverse association is biologically plausible and suggests that homocitrulline may represent a new substrate of *SLC7A9*. This information may be of interest for experimental transport studies and in the context of hyperornithinemia-hyperammonemia-homocitrullinemia syndrome, an inborn error of metabolism.

We agree with the Reviewer that a more comprehensive comparison to findings from published studies of the plasma/serum metabolome is of interest. We have therefore added a

systematic comparison to findings of five large GWAS of participants of European ancestry¹⁻⁵ that studied metabolites quantified with the same technique (Shin *et al* 2014, Long *et al* 2017, Lotta *et al* 2021, Yin *et al* 2022, Hysi *et al* 2022). New **Supplementary Table 4** summarizes the effects of these published plasma mQTLs on plasma levels of the corresponding metabolite in GCKD. The table footnote shows that validation rates are inversely related to the sample size of the discovery study for a range of different statistical significance thresholds, and that the correlation of genetic effects between our and the other studies is very high (Spearman coefficient >0.9 for all Metabolon studies). A description of this new comparison is now included on page 6, lines 135-38. Secondly, we have added an investigation of the plasma mQTLs detected in the GCKD study in the summary statistics of the published plasma/serum mGWAS (**Supplementary Table 5**). A graphical summary (new **Supplementary Figure 2**) shows that there are plasma mQTLs from the GCKD study that did not reach statistical significance in previous studies of the plasma/serum metabolome. Moreover, there were QTLs arising from 44 metabolites not reported in these previous studies. We conclude that continued studies of the plasma metabolome are informative and are referring to this comparison and its related display items on page 6, lines 138-140.

To more clearly delineate the contribution of investigating a substantially larger sample size than in our previous study of the urine metabolome, we quantified the absolute number of discoveries in the urine part of this mGWAS in comparison to our earlier publication (page 6f, line 150-52): we now detect 195 genes (90 previously), 622 mQTLs (240 previously), and 459 unique metabolites with at least one urine mQTL (211 previously).

Lastly, we have followed the Reviewer's excellent suggestion of investigating the results from larger plasma studies in the urine part of this mGWAS and vice versa, the findings of which are outlined in response to point 3, below.

2. The genome-metabolome-wide analysis (of blood and urine) remains rather descriptive and is written quite linearly (a bit like a counting exercise), which makes the paper quite long and not the most interesting read. The authors expand on specific examples, which is a strength of this paper, but possibly more of interest to a specialist nephrology audience.

Response: During the revisions, we have addressed the Reviewer's comment by restructuring parts of the manuscript, as well as by including new sections that expand on selected findings outside of nephrology. One example are metabolite associations at the *SLC10A2* locus (described on page 10f, line 246-63), which we now connect to increased risk of gallstone disease not only through PheWAS and conditional colocalization, but also through Mendelian

Randomization analyses. This example motivates further studies of the implicated bile acids in gallstone disease, and highlights a potential adverse effect of SLC10A2 inhibitors. This novel class of medications is just gaining market approval and in fact, increased rates of cholelithiasis are reported from the first clinical trials. The drug information provided in new **Supplementary Table 11** can serve as a resource for readers to search for additional examples of target genes with approved treatments. Moreover, we have attempted to maximize the usefulness to readers outside nephrology by adding several other global analyses, such as heritability estimation, power analyses, comparison to published mGWAS studies, and comparison of genetic effect sizes on plasma metabolite levels to the ones from a new replication sample of participants from the general population, and to the subset with normal kidney function. We recognize that the initial part of the results section contains a lot of numeric summaries, but believe that this is important context for the interpretation of the more detailed findings in the subsequent parts of the manuscript.

3. The comparative element of this paper is its main feature but is restricted by the power that comes from the number of people with paired samples, which is not necessarily required for these comparative genetic analyses. Since the main contribution of this paper lies in the comparative part, it is a limitation that this is outdated given larger plasma analysis already underway and unnecessarily restricted by the power based on paired samples only.

Response: We appreciate this helpful comment and agree that comparative investigations of findings from larger association studies of the plasma metabolome in our urine data are a useful and interesting addition. We therefore performed two new analyses: first, we investigated our urine mQTLs in available summary statistics from the large studies of the plasma/serum metabolome mentioned above. This comparison underscored the additional discovery potential of urine and showed that both correlation of genetic effects as well as validation rates were higher for the GCKD study than for larger, published GWAS of the circulating metabolome, highlighting the value of a paired study design (**Supplementary Figure 5, Supplementary Table 5**). The comparison also highlighted examples of inverse associations in plasma and urine that are biologically plausible, such as an association at *SLC5A11* with myo-inositol, a tubular transporter responsible for the reabsorption of this metabolite⁶, as well as for other transport proteins (**Supplementary Figure 5**).

Second, we also investigated the mQTLs from the five large studies of the plasma/serum metabolome in the summary statistics of our urine mGWAS (**Supplementary Table 4**). Again,

we observed that correlation and validation are highest for the GCKD study, although there are larger plasma mGWAS available. Of note, even when plasma mQTLs are already known, their “footprint” across matrices contains interesting biological information: when direction-consistent, it can be interpreted as arising from systemic metabolic processes that are captured in urine due to the metabolite’s glomerular filtration, whereas inverse associations can reflect tubular transport processes. Paired samples allow for interpretation of direction consistency with confidence.

We have added a description of the investigations of published plasma/serum mQTLs in our urine mGWAS and our urine mQTLs in summary statistics from published plasma/serum mGWAS in the Results (page 7, lines 154-62) and the Methods (page 39, lines 897-918). We note that comparisons between different mGWAS studies need to be interpreted in light of potential differences in metabolite annotation and analytical decisions such as transformations and covariate adjustments (see **Supplementary Tables 4 and 5** footnote).

4. The fact that a subset of genetic associations is specific or restricted to urine versus plasma is neither surprising nor novel but largely reflects known biology/ biochemistry. Specificity of metabolites being present and detectable in plasma versus urine, respectively, is a key driver of (groups of related) matrix specific metabolites and associated mQTLs, due to the glomerular filtration barrier – something the authors acknowledge. Related to this, the large number of observed colocalizations is driven by the fact that the authors included markers of kidney and liver function.

Response: We agree with the Reviewer that a substantial number of colocalizations with markers of kidney and liver function can be expected, as the levels of many metabolites are related to the function of these organs. As shown in **Reviewer Figure 1**, 14.8% (1,044 of all 7,073) pairwise colocalizations between mQTLs and clinical biomarkers and traits were attributable to colocalization with kidney function markers (creatinine, cystatin C, urea, urate, eGFR_{crea}, eGFR_{cys}), and 7.7% (545/7,073) were attributable to colocalizations with liver function markers (ALT, AST, GGT, albumin, total and direct bilirubin). Thus, while colocalizations between mQTLs and classical markers of liver and kidney function accounted for ~22% of the pairwise colocalizations, they did not account for the majority of them. We have included these numbers in the manuscript on page 11, line 265-67.

The intended purpose of our main **Figure 5a** and **Supplementary Figure 9** was to address the question whether the genetic architecture of mQTLs that leave a signature in the urine and/or plasma metabolome is shared with the function of the kidney, the liver, or both.

Loci containing lipid-metabolism related genes such as *FADS1* or *LIPC* show evidence of shared genetic architecture (positive colocalization) between plasma lipid metabolite levels and the function of the liver, but not the kidney. Likewise, genetic variants in loci encoding several solute transporters with important roles in the kidney such as *SLC34A1* or *SLC7A9* support a shared genetic architecture of the associated metabolites in urine with kidney function. While this is not unexpected (now included on page 11f, lines 269-75), the fact that it is observed from the systematic, unbiased integration of data supports its biological information content. The majority of mQTL-related loci showed a mixed picture, illustrating that the plasma and the urine metabolome contain complementary information about genetic effects that are shared with the function of an important metabolic organ such as liver or kidney, which we have now attempted to clarify by rephrasing on page 11f, lines 265-79.

5. The follow-up and examination of shared genetic drivers of urine vs plasma-specific mQTLs is a valuable part of this work, and it might be useful to be more focused on relevant diseases only and complications of CKD rather than being too explorative without very clear translational gain.

Response: We appreciate the Reviewer's comment and have added additional information that highlights connections between detected mQTLs and kidney diseases and/or complications

thereof (page 10, line 242-45). Based on results from colocalization analyses with diseases and symptoms assessed in the UK Biobank, this includes *GSTM1* and kidney cancer, *FMO4* and hypertensive CKD, *ALPL*, *SLC34A1*, *ABCG2*, and *CYP2D6* and nephrolithiasis, as well as *ABCC4* and urinary retention. Moreover, in response to point 13 by Reviewer 3, we have now investigated the relationship of all exonic index SNPs from our project and diseases, biomarkers and symptoms in the UK Biobank based on the genetic information from whole-exome sequencing (new **Supplementary Table 15**). This investigation shows associations between index SNPs at *ABCG2*, *APOE*, *CPS1*, *DMGDH*, *DPEP1*, *FDPS*, *IDI2*, *KYAT3*, *POR*, *SLC17A1*, *SLC25A45*, *SLC39A8*, *SLCO1B1*, and *SRD5A2* with urinary/renal traits. While p.Gln141Lys in *ABCG2* and serum urate can be considered an experimentally confirmed positive control,⁷ other missense variants represent interesting new candidates for further experimental studies (now mentioned on page 12, lines 283-89).

In order to enhance the focus, as suggested by the Reviewer, we have reduced the number of outcomes investigated for colocalization by not counting redundant endpoints, and removing those not clearly related to diseases, such as history of employment. We do believe, however, that the exploration across a broad spectrum of diseases and traits, enabled by the systematic integration of different comprehensive datasets as performed in our study, can also lead to valuable insights. This is, for example, illustrated by the relation of a plasma glycodeoxycholate 3-sulfate associated mQTL at *SLC10A2* and gallstone disease, for which we are now providing additional support through Mendelian Randomization analyses (please compare detailed response 10 to Reviewer 3).

6. The authors are generally very clear and careful in their wording, nevertheless some of the statements appear to jump to conclusions too quickly. For example, while the *FUT2* section is solid, the suggestion of using urine galactosylglycerol and 1,6- anhydroglucose as surrogates of cardiometabolic risk is unsubstantiated by the existing evidence.

Response: We agree with the Reviewer that the investigation of galactosylglycerol and 1,6-anhydroglucose as surrogates of cardiometabolic risk is best conducted in a future project, once urine metabolite data in very large population-based studies with cardiometabolic outcomes become available. We have therefore removed their denomination as potential surrogates of cardiometabolic risk, and revised the statement to motivate the investigation of their relationship to cardiometabolic diseases in future, larger studies (page 14, lines 330-33).

Reviewer #2:

A very comprehensive and well-conducted analysis of GWAS of plasma and urine metabolites in patients with chronic kidney disease with a series of interesting downstream analyses and exciting illustrative case studies. The paper is very well-written but due to inherent complexity of this ambitious undertaking certain parts are perhaps not immediately easy to follow and may require additional explanations as per information with comments to authors. Overall, a very important contribution to our understanding of genetic determinants of plasma and urine metabolites.

Response: Thank you for the positive assessment of our work.

1. When intersecting GWAS with QTL-like datasets, the identity of specific genes that mediate the association between genetic variants and the phenotype (i.e., plasma level of metabolite or urine level of metabolite) is of utmost importance. The authors used different sources of information including colocalisation with eQTL input from multiple tissues (GTEx) to prioritise the most “likely” driver genes. It does appear that (i) in many cases different genes came up in eQTL analyses conducted in different tissues and (ii) tissues were not prioritised, i.e. input from kidney eQTLs were not treated preferentially i.e. when it came to analysis of urine metabolites, (iii) in some cases the input from eQTL and pQTL may have been overwritten and a different gene assigned [“In case of a clear biological fit to another scored gene (i.e., corresponding monogenic disease or animal model), the prioritized gene was reassigned”], (iv) there were differences in the input between eQTL and blood pQTL analysis. It does appear that in most cases, there was one gene “most likely” mediating the association with plasma and urine metabolites. Could it be that different genes may be mediating the uncovered associations in different tissues (mQTL, eQTL are known to operate in a tissue specific manner – see for example PMID: 30467309). This deserves some contemplation in the discussion. Should the input from kidney/liver eQTL have a preference in this analysis (or at least in the analysis of urine metabolites (kidney)?

Response: We agree with the Reviewer that this point deserves contemplation. Our original rationale to assign equal weights to all investigated tissues when testing for colocalization between mQTLs and eQTLs was that metabolites in plasma, and – when filtered - also in urine, can originate from many tissues and organs and we wanted to maximize the opportunity to detect these colocalizations. For instance, many xenobiotic metabolites are detected in urine,

although they enter the body via the gastrointestinal tract, the skin, or the lung and their levels may be driven by transport or enzymatic processes in these tissues.

To empirically address the Reviewer’s question, we have performed re-analyses of our workflow to assign the most likely causal genes. A new analysis of all 1,299 mQTLs that assigned a weight of 1.5 to kidney tissue (as compared to 1 for all other tissues) for urine mQTLs and of 1.5 for kidney and liver tissue for plasma mQTLs yielded only five instances in which the prioritized gene was no longer among the prioritized ones from our previous workflow. In several of these five cases, the biological connection relating the newly assigned gene to the metabolites was not as plausible as before (**Reviewer Table 1**).

Reviewer Table 1: Sensitivity analyses: mQTLs for which a revised scoring algorithm no longer ranked the originally assigned gene among the top genes. There were no such changes for the remaining 1294 mQTLs.

Original gene	Top gene by the revised score	Involved metabolite levels	Potential biological connection between metabolite and prioritized gene
GATM	SPATA5L1	plasma homoarginine	The enzyme encoded by GATM is involved in homoarginine synthesis, whereas no connection to homoarginine is known for SPATA5L1 , encoding Spermatogenesis Associated 5 Like 1.
KYATI	SPOUT1	plasma 2-hydroxy-phenylacetate	The associated metabolite is a molecule in the metabolism of phenylalanine. KYATI encodes a metabolic enzyme in the pathway “phenylalanine, tyrosine and tryptophan biosynthesis”. SPOUT1 is involved in the association of the centrosomes with the poles of the bipolar mitotic spindle during metaphase, and has not been connected to amino acid metabolism. Thus, the connection of 2-hydroxyphenylacetate to KYATI has greater biological plausibility.
PTER	CIQL3	urine N-acetyl-beta-alanine and N-acetyltaurine	The protein encoded by PTER is reported to enable hydrolase activity by enzymes; CIQL3 encodes a protein predicted to be part of a collagen trimer with a role in synapse organization.
PYROXD2	HPS1	plasma N-methylpipecolate	The associated metabolite is a molecule in lysine metabolism. PYROXD2 encodes a mitochondrial oxidoreductase. Many other N-methylated amino acids and their derivatives are also associated with this locus. HPS1 encodes a protein that participates in a complex important for the biogenesis of organelles like melanosomes.

Additional sensitivity analyses that assigned a weight of 1.5 only to kidney but not to liver tissue, as well as analyses assigning a weight of 2 to either only kidney or to kidney and liver tissue provided very similar results. Based on these new analyses and the rationale laid out above, we decided to retain the original approach to the colocalization analyses of mQTLs and eQTLs. We have revised the manuscript to include the rationale to assign equal weight to all tissues in the Methods (page 42, line 984-88), and are now including a contemplation of the

valid points raised by the Reviewer in the Discussion (page 21, lines 500-2).

2. I think it is important to provide the list of 282 genes in relation to each metabolite in a dedicated table. How many of these genes were non-coding?

Response: Pairwise gene-metabolite relationships are included in **Supplementary Tables 3 and 10**. For the readers' convenience, we have now also generated a table at the level of the 282 genes, some of which are associated with more than one metabolite (new **Supplementary Table 11**). There were no noncoding genes among the prioritized ones, which might be related to the fact that it is more difficult for noncoding genes to achieve high evidence scores. Not only do they not have a chance to show colocalization at the protein level, but they are also not well represented in variant scoring algorithms, nor in databases like OMIM.

3. Have the authors considered using MR to provide further validation of the identified 282 genes as the drivers of the identified GWAS associations? At least for urine, an input from kidney eQTL could be used as the source of information on "exposure" (i.e., gene expression)?

Response: We had considered but decided not to perform MR analyses for the purpose of assigning the most likely causal genes, for several reasons: first, it is likely that not all mQTLs influence metabolite levels through differential gene expression. In fact, there are 249 instances in which the SNPs in single or small (<5 SNPs) credible sets contain a missense variant, which may affect protein processing or function rather than gene expression. Second, the underlying assumptions of MR, especially the absence of associations with other traits that may introduce horizontal pleiotropy (in this case other transcripts), do not hold for many metabolite-associated genetic instruments. Many SNPs are associated with differential transcript expression of more than one gene in *cis*. A third reason complicating systematic MR analyses using gene expression as the exposure is that not all transcripts can be comprehensively assessed in each tissue because of various reasons (true missingness, low sample size of some GTEx tissues, other). Fourth, as opposed to powerful GWAS meta-analyses of complex traits that often identify hundreds of associated loci and hence potential instruments, most GWAS of gene expression are only conducted in the *cis* region, which reduces the possibility to perform methodological sensitivity analyses to check the underlying assumptions of MR that require the presence of many independent instruments. For these reasons, we believe that systematic MR investigations of gene expression to assign the most likely causal gene at each locus is out of the scope of this

project. To nevertheless include information from eQTL studies of diverse tissues including kidney, we had therefore performed systematic colocalization analyses of genetic associations with metabolites and with gene expression to identify most likely driver genes. These analyses have fewer assumptions and can more easily be applied to all loci.

We do agree with the Reviewer that MR analyses can be useful to further investigate potential causal relationships at the identified mQTLs. We have therefore now performed MR analyses to investigate the relationship between selected mQTLs and health outcomes, for which we were able to check assumptions. This work is described in more detail in response to comment 5 below (for eGFR) and to comment 10 of Reviewer 3.

4. Was there an enrichment for kidney-related phenotypes in the analysis of urinary mQTLs and 4,422 biomarkers and diseases in UK Biobank?

Response: We examined the Reviewer's question by performing overrepresentation analyses of urine versus plasma mQTLs with at least one positive colocalization signal among kidney-related phenotypes in the UK Biobank (kidney biomarkers creatinine, cystatin C, urea, urate, eGFR_{crea}, and eGFR_{cys}, as well as all phenotypes containing "kidney" or "renal" (but not adrenal) or "nephro" or "calculus" or "urine"). No significant association was identified (odds ratio 1.02, p-value 0.90). When interpreting these findings, we recognize that many plasma metabolites are filtered to urine, and their associated mQTLs may therefore be detected in both plasma and urine. We therefore also examined an overrepresentation among urine-specific mQTLs and kidney diseases that originate in the tubule ("calculus of kidney", "calculus of ureter", or "urinary calculus"). Overrepresentation analysis of urine-specific as compared to other mQTLs with a positive colocalization showed a trend for enrichment (odds ratio 1.51), but confidence intervals were wide and the association hence not significant (p-value 0.59).

5. Would it be possible to use this rich dataset to see which blood and urinary metabolites show the strongest associations with the biochemical measure of kidney function? Would it be possible to comment on the causality of the detected associations?

Response: We agree with the Reviewer that these are pertinent questions and have performed a series of complementary analyses. First, we inferred, for each plasma and urine metabolite, the proportion of metabolite variance explained by eGFR based on linear models in the GCKD study. The new **Supplementary Figure 10** shows results for all investigated metabolites,

ordered by the maximum of explained variance across plasma and urine. Consistent with expectations, the metabolite for which the eGFR explained the largest proportion of variance was plasma creatinine.

Second, we have investigated the relationship between 424 index SNPs of a large GWAS meta-analysis of eGFR (Stanzick, et al. ⁸) with the mQTLs detected in our study, where 414 of these index SNPs were present. **Reviewer Table 2** (attached .xlsx file) lists the 25 and 27 eGFR index SNPs from Stanzick *et al.* that were significantly associated with metabolite levels in plasma ($p < 0.05 / (424 * 1296)$) and urine ($p < 0.05 / (424 * 1401)$), respectively, further divided into those with support from colocalization. This comparison implicates especially mQTLs at *CPS1*, *NAT8*, and *SLC6A13* as the metabolite loci with the strongest relationship to the most common measure of kidney function, the eGFR.

Third, we have performed MR analyses at loci implicated by positive colocalizations with eGFR_{crea} or eGFR_{cys}. For these mQTLs, we assessed the statistical support for altered kidney function (exposure) as a cause of altered metabolite levels (outcome). We concentrated on this direction because of the abundance of independent genetic instruments for eGFR, which enables checks of MR assumptions as well as sensitivity analyses, which is not the case for most metabolites. We thoroughly filtered genetic instruments for potential pleiotropy (see updated Methods), and now report 11 findings with significant support of altered kidney function causing a change in metabolite levels (**Supplementary Table 13**). Of note, many of the implicated metabolites are related to the function of a detoxification enzyme almost exclusively expressed in the kidney, NAT8. It is conceivable that altered kidney function may lead to changes in the levels of substrates or products of a central renal enzyme. As expected, better kidney function was related to higher levels of all NAT8 products in plasma and urine. We have included this new information in the results section of the **Supplementary Information**, to which we are referring in the manuscript on page 12, line 279-82, and in the Methods (page 46, lines 1066-79).

6. Suppl Figures 1 and 2 did not display well in the Supplement.

Response: We have generated new figures that should display in a better resolution while maintaining a much smaller file size.

7. Is mQTL an appropriate abbreviation of metabolite QTLs (eg. vs methylation QTL)?

Response: We have decided to use the term “mQTL” for metabolite QTL in order to stay consistent with previous literature: our own work (e.g., Schlosser, et al. ⁹) and work by others (e.g., Kraus, et al. ¹⁰, Demirkan, et al. ¹¹) that used “mQTL” for metabolite QTLs, the term has been coined by one of the very first genetic studies of the metabolome in 2011 (Nicholson, et al. ¹²). Although some studies of the methylome also use “mQTL”, many studies of methylation QTLs use the term “meQTL”, for example a recent multi-Omics paper published in *Nature Genetics* (Liu, et al. ¹³), as well as a recent large-scale genetic screen of DNA methylation published in *Nature Genetics* (Hawe, et al. ¹⁴). We would therefore like to retain the use of “mQTL” for metabolite QTL.

Reviewer #3:

Schlosser et al examine the genetic basis of an important biological phenomenon, glomerular filtration, though paired examination of metabolite levels in serum and urine in CKD patients. While the incredibly extensive results are very interesting and the dataset has significant potential, the current analysis is somewhat limited in terms of broad effects on the genome wide association landscape (specifically, using tools that examine effects below genome wide significance). Some specific comments below:

1. Please put the exact N in (5023?) on line 104.

Response: We have included that metabolite levels were measured for 5,023 GCKD participants.

2. How does the heritability differ for matched serum and urine metabolites?

Response: Thank you for raising this interesting question. We have now computed heritability for all metabolites that give rise to any of the 1,299 identified mQTLs as described in the expanded Methods on page 40, lines 928-33, and added the estimates to **Supplementary Table 2**. The new **Supplementary Figure 6** shows the requested comparison of the heritability for matched plasma and urine metabolites: there is a high, positive correlation between the estimates of heritability for a given metabolite’s plasma and urine levels. Since most metabolites that can be quantified in both plasma and urine are filtered by the kidney, this positive

correlation is expected. Moreover, there are instances in which the difference in estimated heritability for plasma and urine differed by more than 25%. Such differences can contain interesting biological information. For example, three metabolites with large differences in heritability between urine and plasma are N-acetylated amino acids, all of which have an mQTL at *NAT8*. *NAT8* is highly and selectively expressed in the kidney, where the encoded enzyme N-acetylates molecules to make them water soluble for subsequent excretion. We have added this example to the footnote of new **Supplementary Figure 6**.

3. What is the distribution of genetic correlations across metabolites? This would enhance the paragraph starting line 162.

Response: To address the Reviewer's question, we estimated the genetic correlation coefficients for all 11,524 pairs of metabolites that share at least one mQTL, as well as for all 779 plasma-urine pairs of metabolites that were quantified in both matrices, irrespective of shared mQTLs, using the recently published SCORE algorithm.¹⁵ We found that only one third (33.4%) of the analyses converged to genetic correlation coefficients that also included a standard error estimate. This in large part relates to statistically very strong associations between genetic variants and metabolite levels at many mQTLs. The removal of such loci is recommended before estimation of genetic correlations.¹⁶ However, many of the metabolites only showed a single or two very strong mQTLs, making this infeasible. We note that the estimation of genetic correlations is usually applied to results from large meta-GWAS of complex traits, which often contain dozens or even hundreds of significantly associated loci, i.e. display clear polygenicity. Furthermore, the majority of our converged analyses (64.9%; 2,670 of 4,110) had standard error estimates greater than 0.5, consistent with the observation that very large studies and polygenic architecture are required for precise estimates of genome-wide genetic correlations. When repeating these analyses using the LD Score regression method,¹⁷ we observed similarly low rates of model convergence, and also imprecise estimates. We conclude that much larger sample sizes would be required to infer precise and reliable genetic correlations of metabolites with each other, and decided not to include their estimation in the manuscript.

4. Please clarify on line 130 that this is *across metabolites* (I assume).

Response: We have now clarified our workflow on lines 132f: we merged, for each metabolite, overlapping 1 Mb intervals that showed at least one significant association, and then selected the SNP with the lowest p-value in each region as the index SNP. We used colocalization of

mQTLs to assess whether associations with different metabolites mapping to the same region were likely driven by the same underlying variant.

5. The suggestion behind the statement on line 153 is intriguing, but it is not actually tested in the manuscript. Please evaluate this claim or remove it.

Response: We agree with the Reviewer and have removed the sentence.

6. On line 157, were mqtls associated with multiple metabolites more or less likely to fine map to single SNPs? And what about by allele frequency?

Response: We performed linear regression analyses relating the average number of credible set size at a given locus to the number of associated metabolites and found no significant relationship ($p > 0.8$). Conversely, a significant association was identified for the relationship of credible set size and index SNP minor allele frequency ($p = 2.3E-13$), with lower MAF related to smaller credible set size. We have added these observations to the manuscript on page 7, lines 168-70, and extended the methods accordingly (page 41, lines 949f).

7. In the paragraph starting on line 162, it's unclear how much the starting and specificity are driven by global differences in power or the pathways that these substrate specific metabolites belong to. These global comparisons would greatly aid interpretation by the reader and should be included.

Response: We agree with the Reviewer that these are helpful comparisons to aid interpretation. We have therefore conducted post-hoc power analyses for urine and for plasma mQTLs, separately by metabolite super-pathway, for a sample size of 5,000 and across a range of minor allele frequencies. For each matrix-super-pathway subgroup, we assumed the median observed effect size across mQTLs as well as the median standard deviation of the metabolites with a mQTL within the group. As shown in the new **Supplementary Figure 7** to which we now refer on page 8, lines 179-81, the power to detect significant associations at the thresholds used in our study was very similar for urine and plasma for almost all metabolite super-pathways. The only clear difference was observed for energy metabolism, a category that contained only 0.8% of mQTLs.

8. The paragraph starting on line 195 confuses me. As these highly pleiotropic signals are counted multiple times in the list of pairwise colocalizations, then please include the number of distinct mQTLs underlying the lines 190-194 as well. Am I understanding your implication correctly that pleiotropy is evidence of these genes having key functionality? I would provide additional evidence for this.

Response: The numbers of distinct mQTLs for each of the four groups has now been added to the legend of **Figure 3**. The fact that a very high number of colocalizations is observed for some genetic loci indicates that they have key functionality in the sense that the same underlying genetic signal is likely to drive the association with many metabolites. When investigating colocalizations restricted to a certain matrix such as urine, we can therefore identify the metabolomic “footprint” of a genetic locus on this matrix. As suggested by the Reviewer, we have now added additional citations that support that the implicated genes influence multiple metabolites specifically in the investigated matrix (page 10, lines 224). Pleiotropy is further supported by the targeted investigation of exonic mQTL index SNPs in the UK Biobank whole-exome sequencing data (see below), where available signals in *CPS1*, *SLCO1B1*, and *SLC17A1* were associated with numerous other biomarkers, including metabolites.

9. Please provide the effect size and p value for both serum and urine on line 234.

Response: The effect sizes for glycocholate in plasma and urine were 0.07 and -0.44, respectively, and the corresponding p-values were 0.21 and 2.9E-13. Prompted by Reviewer comments that encouraged the study of potentially causal relationships (please also see response to the next point), we have however now revised the entire section of the manuscript related to *SLC10A2*.

10. Please provide your evidence for *SLC10A2* being the causal gene on line 229.

Response: *SLC10A2* is the only gene in the associated LD block (see **Supplementary Figures 1 and 4**, regional association plots). The encoded protein is the primary transporter for bile acid uptake in the distal ileum (Dawson, et al. ¹⁸), and both associated metabolites are bile acids. Moreover, the index SNP for plasma glycodeoxycholate 3-sulfate is a missense variant that we fine-mapped to a single-SNP credible set. The resulting valine to isoleucine substitution at position 98 encoded by rs55971546 has experimentally been shown to result in reduced bile

acid transport using one *in vitro* system,¹⁹ but not in an earlier study using another system²⁰. Moreover, we have now added conditional colocalization analyses (see updated Methods) and identified evidence for a shared genetic variant, rs55971546, giving rise to both the association with metabolite levels as well as with gallstone (PP H4 = 0.99; new **Figure 4a and b**). Across 1,400 EHR-derived broad PheWAS codes from the UK Biobank, rs55971546 was exclusively associated with cholelithiasis (**Figure 4c**). Together, we believe this is compelling evidence for *SLC10A2* as the likely causal gene.

To further strengthen this evidence, we have performed MR analyses. As shown in **Figure 4d**, we provide evidence for a causal role of this bile acid, or the mechanism which it reflects, in gallstone disease susceptibility. This motivates special attention to the occurrence of gallstones as a potential adverse effect of the new class of medications targeting *SLC10A2* for cholestasis. In fact, data from emerging clinical trials describes increased rates of cholelithiasis in the treatment group (ClinicalTrials.gov Identifier: NCT03566238). This observation may be especially relevant because *SLC10A2* inhibitors are currently explored in the context of other, more common liver, metabolic, and digestive diseases such as constipation. Lastly, with more than 30 bile acids quantified in plasma and urine each, our data represents an interesting opportunity to systematically explore which specific bile acids are altered in variant carriers (**Figure 4e**). We believe that this new finding significantly adds to our manuscript and have revised the corresponding part in the Results section on page 10f, lines 246-63.

11. On line 241, is *egfrcrea* from your analysis of UKBB or from Stanzick et al.? Do these agree? Do they differ from using creatinine directly?

Response: The *eGFRcrea* GWAS summary statistics used in our manuscript are from Stanzick, et al.⁸, which is a meta-analysis of summary statistics from the CKDGen Consortium (Wuttke, et al.²¹) and a GWAS of *eGFR* in the UKB. We therefore did not evaluate *eGFR* in the UKB only, since a separate assessment would have resulted in reduced power.

We did conduct a GWAS of serum creatinine in the UKB for the purpose of this manuscript. The new **Reviewer Figure 3** contains a Venn diagram showing the overlap of mQTLs that colocalize with *eGFR* and those that colocalize with serum creatinine. Because of differences in sample size and hence power of the two analyses, we are also showing a scatter plot that compares genetic effect sizes on creatinine-based *eGFR* with those on serum creatinine for each index SNP at a mQTL that colocalizes with at least one of both traits. We observe that genetic effects compare very well and are highly correlated (Spearman coefficient -0.99, p-value 1.6e-61).

Reviewer Figure 3: Comparison of genetic associations of mQTLs and their index SNPs with eGFRcrea and with serum creatinine as a Venn diagram (left) and a scatter plot (right).

Reviewer Figure 3: Comparison of genetic associations of mQTLs and their index SNPs with eGFRcrea and with serum creatinine as a Venn diagram (left) and a scatter plot (right).

12. The paragraph starting on line 240 takes a nice approach for evaluating shared signals with liver function traits, but misses the potential to consider regulatory annotations. Using tools like stratified LDSC, do you see evidence of this corresponding serum vs urine enrichments in kidney vs liver? While there is some evidence this might be the case, the manuscript and dataset provide a really nice opportunity to quantify this phenomenon and support the results here and in SF5.

Response: As suggested by the Reviewer, we performed stratified LDSC analyses for 944 metabolites with an mQTL. Across all 54 tissues and metabolites, the lowest p-value for a tissue was $1E-05$, which is not statistically significant after correction for multiple testing. Restricting the results to only liver and kidney tissue, the top 10 metabolites with the strongest enrichment for liver are two urine and eight plasma metabolites, the latter of which contain 5 bilirubin and one glucuronidated metabolite. The top 10 metabolites for kidney are mostly from urine (7/10) and often related to amino acid and nucleotide metabolism. These findings are consistent with existing biological knowledge about the functions of the liver and the kidney, but they are not statistically significant. We conclude that such analyses are currently underpowered and believe

that it will be interesting to revisit them once mGWAS become available that reach sample sizes of several 100,000s, as is the case for many complex traits including the ones examined in the original manuscript proposing stratified LDSC.²²

13. For line 256, please provide evidence that there is an enrichment in the target traits across your 115 targets at the mQTLs themselves, not just at the genes they putatively target.

Response: We agree that variant-level investigations of associated traits and diseases are an interesting addition to the gene-based associations we presented previously. We have performed these analyses and added the results as new **Supplementary Table 15**, as well as included them in the Results section (page 12, lines 284-9). Of note, the new results include the prioritized missense variant rs55971546 in *SLC10A2* (see above), which was exclusively associated with gallstone disease among 11,669 diseases queried.

14. What do the associations look like for non-urine-specific (or even serum specific, and clinical tests) metabolites at rs601338? And where is *FUT2* expressed? Given the disease outcome associations presented, it would be quite interesting if these two metabolites were the only associations.

Response: We have now clarified that rs601338 was exclusively associated with levels of the metabolites galactosylglycerol and 1,6-anhydroglucose, both of which happen to be urine-specific (page 13, lines 317f). With the clinical tests performed in the UK Biobank, which we evaluated for colocalization, the *FUT2* signal colocalized with alkaline phosphatase, CRP, gamma-GT, LDL and total cholesterol, apolipoprotein B, IGF-1, total bilirubin and protein, and eGFR_{cys} (**Supplementary Table 12**). As displayed in **Reviewer Figure 4**, *FUT2* shows a ubiquitous pattern of expression in the GTEx data, with highest levels in tissues of the digestive system. We have added this information to the manuscript on page 13, line 318.

Reviewer Figure 4: *FUT2* transcript levels are ubiquitously expressed among GTEx v8 samples, but particularly high in tissues of the digestive tract. Box limits mark the 25th and 75th percentile.

15. Similarly, what does the phewas look like at rs62542743?

Response: **Reviewer Figure 5** shows a PheWAS of rs62542743 in the *AQP7* locus, using the same dataset as for rs601338 in *FUT2*. No significant findings were identified after appropriate correction for multiple testing, which may not be surprising given the hypothesized mechanism (see below).

Reviewer Figure 5: PheWAS of rs62542743 in *AQP7* and 1,400 EHR-derived codes for 57 million TOPMed-imputed variants in 400,000 white British individuals as provided by the Lee lab (<https://pheweb.sph.umich.edu/>).

16. There are previously reported associations at *AQP7*, though your proposed

mechanism seems quite plausible.

Response: We have now investigated all four genome-wide significant associations reported for “AQP7” in the GWAS Catalog. These arise from three unique SNPs: rs144994089 (trait: urinary albumin excretion), rs35201538 (lung adenocarcinoma), and indel rs145465051 (blood glucose levels). Using the European ancestry samples included in the TOPMed Project, the variants rs144994089 and rs35201538 are not in LD with the index SNP identified in our manuscript, rs62542743 ($r^2 < 0.2$; <http://topld.genetics.unc.edu/>). The variant rs145465051 was not included in the TOPMed or other SNP reference databases such as LDlink, but its estimated frequency differs clearly from the one of our variant (0.18 vs. 0.06). Given the literature support of AQP7 as a glycerol transporter, including mouse models and a case report of a monogenic manifestation in a single family, as well as the absence of associations between rs62542743 and other diseases and traits, we agree with the Reviewer that the proposed mechanism is quite plausible.

17. The SLC13A3 story is very nice. Do you see experimental evidence of allele specific binding at rs6124828 in heterozygous individuals, either in HNF CHIP or ATAC/histone chip?

Response: We agree with the Reviewer that this is an interesting question, which we are looking forward to investigate further once suitable datasets become available. At the moment, we do not have approval to obtain genetic information from the donors of the kidney samples from which we generated the ATAC-seq data. We are also not aware of any publicly available individual-level datasets that contain both genotype and HNF CHiP or ATAC/histone chip information that we could use to identify persons heterozygous for rs6124828, in order to explore this question further. The HNF ChIP-seq information in our current manuscript is derived from a cell line, for which we do not know the genotype status of the donor at rs6124828. Therefore, we have included the investigation of this question as an interesting area of future investigation (page 16, line 379-81).

18. Is there evidence of epistasis with variants at HNF genes?

Response: We performed SNP*SNP interaction analyses for variants within +/- 500 kb around the *HNF1A* and *HNF1B* genes and the rs6124828 variant in *SLC13A3*, for each of the four associated metabolites. We did not identify any significant SNPs after accounting for testing of

502 independent SNPs across both regions (estimated with plink using $r^2 < 0.2$) and the number of traits (all interaction p-values $> 0.05 / (502 * 4)$). We conclude that there is no compelling evidence for epistasis of rs6124828 with common variants at *HNF1A* or *HNF1B*. This is not unexpected if rs6124828 at *SLC13A3* alters the binding motif of these transcription factors, whereas variation in the genes encoding for the transcription factors themselves may lead to different amounts or conformation of the transcription factors.

19. Suggestion to run mendelian randomization to support the (very neat!) results in the paragraph starting on line 393. Can you use other associations with these metabolites genome wide to tease out which might be causal (or is that what you did on line 398? I was confused)? That would likely simplify 7d as well, which I had trouble understanding.

Response: MR analyses at the *DPEP1* locus are complicated by its pleiotropic associations with multiple traits and diseases, which can introduce horizontal pleiotropy, thereby violating MR assumptions. We therefore instead carried out colocalization analyses of genetic associations at this locus with metabolites and multiple traits and diseases. These showed that the same underlying genetic signal gave rise to associations with osteoarthritis and hypertension in opposing effect directions. We realize that the graphic representation of these results may not have been straightforward, and have now revised **Figure 8d** to address this point.

20. I would consider running GWAS for the ratio of urine to serum levels of each metabolite as well, which should pinpoint effects on the kidneys even further.

Response: We agree with the Reviewer that the evaluation of metabolite ratios in the context of their renal handling is of special interest. We believe however that such analyses are most informative when performed with measurements that provide absolute instead of semi-quantitative metabolite levels, as only this enables the calculation of the metabolites' fractional excretions. In fact, we are working on such analyses as part of a dedicated, separate project that uses a different technique for metabolite quantification. We have therefore decided to include the study of metabolite ratios that use absolute quantification as a future endeavor of interest (page 21, lines 504).

21. The statement on line 469 only applies to urine; these results should be extended to serum in this manuscript (which I, for one, would find useful) or the statement should be clarified.

Response: We thank the Reviewer for this important comment. Indeed, while we had previously shown that genetic effects on urine metabolite levels were similar for participants of the GCKD study and those from a general population-based sample,⁹ we have now added new data to extend this evaluation to circulating metabolites. We find that the genetic effects on plasma metabolite levels are very similar between participants of the GCKD study and those of the population-based ARIC study, in which a comparable serum metabolomics dataset was available. For 459 mQTLs with available SNPs and metabolite levels, we observed a replication rate of 94.1% (P-value <0.05/459 and direction consistent). Moreover, genetic effects on metabolite levels were highly correlated (Pearson correlation coefficient of 0.98). A high correlation of genetic effect sizes was also observed when only ARIC participants with eGFR of ≥ 60 ml/min/1.73m² were included for the comparison (Pearson correlation coefficient of 0.98). We included these new findings in the manuscript on page 6, lines 140-148. A scatter plot of the genetic effects in the two studies is now included as Supplementary Figure 3, while the new Supplementary Table 6 includes the numeric results.

References:

1. Shin, S.Y. *et al.* An atlas of genetic influences on human blood metabolites. *Nat Genet* **46**, 543-550 (2014).
2. Long, T. *et al.* Whole-genome sequencing identifies common-to-rare variants associated with human blood metabolites. *Nat Genet* **49**, 568-578 (2017).
3. Lotta, L.A. *et al.* A cross-platform approach identifies genetic regulators of human metabolism and health. *Nat Genet* **53**, 54-64 (2021).
4. Hysi, P.G. *et al.* Metabolome Genome-Wide Association Study Identifies 74 Novel Genomic Regions Influencing Plasma Metabolites Levels. *Metabolites* **12**(2022).
5. Yin, X. *et al.* Genome-wide association studies of metabolites in Finnish men identify disease-relevant loci. *Nat Commun* **13**, 1644 (2022).
6. Lahjouji, K. *et al.* Expression and functionality of the Na⁺/myo-inositol cotransporter SMI2 in rabbit kidney. *Biochim Biophys Acta* **1768**, 1154-9 (2007).
7. Woodward, O.M. *et al.* Identification of a urate transporter, ABCG2, with a common functional polymorphism causing gout. *Proc Natl Acad Sci U S A* **106**, 10338-42 (2009).
8. Stanzick, K.J. *et al.* Discovery and prioritization of variants and genes for kidney function in >1.2 million individuals. *Nat Commun* **12**, 4350 (2021).
9. Schlosser, P. *et al.* Genetic studies of urinary metabolites illuminate mechanisms of detoxification and excretion in humans. *Nat Genet* (2020).
10. Kraus, W.E. *et al.* Metabolomic Quantitative Trait Loci (mQTL) Mapping Implicates the

- Ubiquitin Proteasome System in Cardiovascular Disease Pathogenesis. *PLoS Genet* **11**, e1005553 (2015).
11. Demirkan, A. *et al.* Insight in genome-wide association of metabolite quantitative traits by exome sequence analyses. *PLoS Genet* **11**, e1004835 (2015).
 12. Nicholson, G. *et al.* A genome-wide metabolic QTL analysis in Europeans implicates two loci shaped by recent positive selection. *PLoS Genet* **7**, e1002270 (2011).
 13. Liu, H. *et al.* Epigenomic and transcriptomic analyses define core cell types, genes and targetable mechanisms for kidney disease. *Nat Genet* **54**, 950-962 (2022).
 14. Hawe, J.S. *et al.* Genetic variation influencing DNA methylation provides insights into molecular mechanisms regulating genomic function. *Nat Genet* **54**, 18-29 (2022).
 15. Wu, Y. *et al.* Fast estimation of genetic correlation for biobank-scale data. *Am J Hum Genet* **109**, 24-32 (2022).
 16. 10-10-2022, <https://github.com/bulik/ldsc/issues/54>.
 17. Bulik-Sullivan, B.K. *et al.* LD Score regression distinguishes confounding from polygenicity in genome-wide association studies. *Nat Genet* **47**, 291-5 (2015).
 18. Dawson, P.A., Lan, T. & Rao, A. Bile acid transporters. *J Lipid Res* **50**, 2340-57 (2009).
 19. Ho, R.H. *et al.* Functional characterization of genetic variants in the apical sodium-dependent bile acid transporter (ASBT; SLC10A2). *J Gastroenterol Hepatol* **26**, 1740-8 (2011).
 20. Love, M.W. *et al.* Analysis of the ileal bile acid transporter gene, SLC10A2, in subjects with familial hypertriglyceridemia. *Arterioscler Thromb Vasc Biol* **21**, 2039-45 (2001).
 21. Wuttke, M. *et al.* A catalog of genetic loci associated with kidney function from analyses of a million individuals. *Nat Genet* **51**, 957-972 (2019).
 22. Finucane, H.K. *et al.* Heritability enrichment of specifically expressed genes identifies disease-relevant tissues and cell types. *Nat Genet* **50**, 621-629 (2018).

Decision Letter, first revision:

4th Jan 2023

Dear Professor Köttgen,

Your Article, "Genetic studies of paired metabolomes reveal enzymatic and transport processes at the interface of plasma and urine" has now been seen by 2 referees. You will see from their comments below that while they find your work of interest, some important points are raised. We are interested in the possibility of publishing your study in Nature Genetics, but would like to consider your response to these concerns in the form of a revised manuscript before we make a final decision on publication.

We therefore invite you to revise your manuscript taking into account all reviewer and editor

comments. Please highlight all changes in the manuscript text file. At this stage we will need you to upload a copy of the manuscript in MS Word .docx or similar editable format.

*2) If you have not done so already please begin to revise your manuscript so that it conforms to our Article format instructions, available [here](http://www.nature.com/ng/authors/article_types/index.html). Refer also to any guidelines provided in this letter.

[redacted]

We hope to receive your revised manuscript within 8 to 12 weeks. If you cannot send it within this time, please let us know.

Nature Genetics is committed to improving transparency in authorship. As part of our efforts in this direction, we are now requesting that all authors identified as 'corresponding author' on published papers create and link their Open Researcher and Contributor Identifier (ORCID) with their account on the Manuscript Tracking System (MTS), prior to acceptance. ORCID helps the scientific community achieve unambiguous attribution of all scholarly contributions. You can create and link your ORCID from the home page of the MTS by clicking on 'Modify my Springer Nature account'. For more information please visit please visit

<http://www.springernature.com/orcid>>www.springernature.com/orcid.

Sincerely,
Wei

Wei Li, PhD
Senior Editor
Nature Genetics
New York, NY 10004, USA
www.nature.com/ng

Reviewers' Comments:

Reviewer #1:
Remarks to the Author:

Thank you for the opportunity to assess the revised version of this manuscript. The authors have carefully attempted to respond to the reviewers' comments.

This is a fast-moving field, and the updates do still not account for the existing knowledge and recent studies that vastly exceed the current one in sample size (but are lacking the matrix comparison). Therefore, all results for the genetic analysis of plasma metabolites need to be updated to not make false claims of novelty. I cannot be sure, but strongly assume that the new results render much of the plasma discoveries of this work redundant. Examples highlighted in the rebuttal letter, such as rs55971546 in SLC10A2, are known and have been reported not only for glycochenodeoxycholate sulfate but long been linked to gallstones, so neither the variant, gene, nor the mechanism are new.

The genetic comparison between plasma and urine using all available mQTLs (rather than the matched analysis) is a helpful addition but also out of date. Please state more clearly how many of the 'novel' urine mQTLs (compared to the author's earlier work) are already captured by larger plasma studies, i.e. concordant?

I was struck by the fact that the correlation of genetic effects as well as validation rates were higher for the GCKD study than for larger, published GWAS of the circulating metabolome. Is this the case for the largest efforts? If yes, this is interesting, and the authors interpret this to demonstrate the value of a paired study design. However, it does beg the question how much of this this is driven by selected focus on CKD patients with reduced kidney function and not generalisable to a general population?

Given these limitations, a study more focused on the comparative element would not only help to reduce the still very lengthy technical description of GWAS loci (in plasma), which is not a unique feature of this study, but allow focussing on the main stated rationale: to use the plasma-urine comparison in CKD patients to identify transporters and enzymes with pharmaceutical relevance for

CKD and more broadly. This part does not stand out clearly enough and is what might make this work competitive.

Another point of note is the total absence of any investigation of sex differences.

A strength of the revised version is that replication is performed, although it would be more useful had this been extended to include an assessment of the generalisability of findings to ethnic groups represented in the ARIC replication cohort to compensate at least partly for the fact that this is yet another White European focussed effort.

Reviewer #2:

Remarks to the Author:

The authors have provided a very strong and detailed rebuttal to my comments and the revised version of the manuscript, and the Supplement look excellent. There are only a few remaining minor points worth addressing (as listed below).

1. Thank you for conducting an additional sensitivity analysis and providing a rationale for assigning equal weight to all tissues in the algorithm used to nominate the most likely gene drivers. I think that inclusion of Table S1 cataloguing these genes is most useful. While I am satisfied that these additional analyses and results make a very persuasive case for not making any further changes to the algorithm, I do think that the discussion requires a little bit more contemplation of its limitations. Indeed, the algorithm (i) identifies the highest scoring gene, but it does not specify the tissue through which the genes is likely to influence the metabolites (ii) it does not provide equal weights to protein-coding and non-coding (e.g. long non-coding) genes in scoring which may explain why the latter are not amongst the prioritised genes. This is important because (i) a proportion of GWAS signals operate through changes in expression of different genes in different tissues (please see CKD-defining traits as a relevant example - Nat Commun. 2018;9:4800.) and (ii) long non-coding RNAs are recognised gene mediators of associations signals with complex human traits. It would be most helpful to see this being covered in the discussion with appropriate references.
2. Can you please clarify for what percentage of the mQTLs you could not identify the most likely gene drivers?
3. I think that the title of the Results section "Systemic roles of renal dipeptidase 1 /DPEP1/: circulating enzymes, hypertension and arthropaties" needs changing to "Systemic roles of dipeptidase 1 /DPEP1/: circulating enzymes, hypertension and arthropaties". Indeed, DPEP1 encodes dipeptidase 1 also called Microsomal Dipeptidase or Dipeptidase 1 (Renal). In the current form the titles suggests that it is the kidney expression of this gene that determines the levels of digestive enzymes, hypertension and arthropaties while of course based on the provided data one cannot assign a specific tissue that is responsible for these findings. As the authors point out, DPEP1 is expressed not only in the kidney.

Reviewer #3:

None

Author Rebuttal, first revision:

Dear Editors and dear Reviewers,

Thank you for the evaluation of our revised manuscript. We have carefully considered your comments, performed new analyses, and responded to each comment in the point-by-point response, below.

We believe that your comments have helped us to further improve our manuscript. We hope that the implemented changes meet your requests, and that we have answered all open questions.

Sincerely,

Pascal Schlosser, Nora Scherer, and Anna Köttgen, on behalf of all authors

Point-by-point response

Reviewer #1:

Thank you for the opportunity to assess the revised version of this manuscript. The authors have carefully attempted to respond to the reviewers' comments.

This is a fast-moving field, and the updates do still not account for the existing knowledge and recent studies that vastly exceed the current one in sample size (but are lacking the matrix comparison). Therefore, all results for the genetic analysis of plasma metabolites need to be updated to not make false claims of novelty. I cannot be sure, but strongly assume that the new results render much of the plasma discoveries of this work redundant. Examples highlighted in

the rebuttal letter, such as rs55971546 in SLC10A2, are known and have been reported not only for glycochenodeoxycholate sulfate but long been linked to gallstones, so neither the variant, gene, nor the mechanism are new.

Response: We agree that the elucidation of the genetic basis of plasma metabolite levels is a fast-moving field. Since we had performed the requested comparisons for the first revision of our manuscript, two additional studies that met our selection criteria for maximizing overlap with the findings from our study were published in peer-reviewed journals (Surendran *et al*, Nature Medicine, November of 2022¹; Chen *et al*, Nature Genetics, January of 2023²). We were also meanwhile able to obtain summary statistics of the results published by Hysi *et al* in 2022³. We have therefore updated the comparison of our findings for plasma metabolites to include these new studies.

As in our previous comparison, we find excellent correlation of genetic effects on plasma metabolites detected in the GCKD study with the corresponding metabolites and variants (or their proxies) in the newly added population-based studies: Spearman correlation coefficients are 0.92 for all three studies. Our initial motivation for the comparative analyses of plasma findings to published studies was to assess whether genetic effects in the GCKD study and in population-based samples agree and how their effect sizes correlate, rather than claiming or emphasizing novelty. Based on our newly updated comparisons, the majority (92.6%) of plasma mQTLs from our study were already reported as genome-wide significant in one or more of the seven published studies of the circulating metabolome. This is not surprising, since their sample size exceeds that of our study up to 17-fold. In fact, it bolsters confidence in the results that we use to perform a comparative study of genetic footprints on the urine and plasma metabolomes, the main goal of our study. We note that we still detected 50 significant plasma mQTLs from our study that did not pass the significance threshold in any of the evaluated larger GWAS of the circulating metabolome, of which 20 arose from 17 metabolites not reported in any of the previous studies. We have added these new results and considerations to the Results (page 6f, lines 149-160), updated the methods section accordingly, and updated the respective display items (**Supplementary Figure 3, Supplementary Tables 5 and 6.**) Lastly, we have added citations of additional studies of the circulating metabolome that did not meet the selection criteria for our comparison to published studies (Results, page 43, lines 983-990).⁴⁻⁸

About *SLC10A2*, the association with plasma glycodeoxycholate 3-sulfate has been reported in the studies by Surendran *et al* and Chen *et al* in the meantime, including Mendelian randomization (MR) analyses in the study by Surendran *et al*. We agree with the Reviewer that

our previous figure needed revision, and have taken the Reviewer's advice to more clearly emphasize the comparative aspect of our study.

In fact, the *SLC10A2* locus represents an interesting example of how the study of different matrices can facilitate the detection of different likely causal variants in the same locus. By adding the information about the urine mQTL at *SLC10A2*, we show that the fine-mapped plasma and urine index SNPs leave different footprints on the bile acid profiles in the two matrices (new **Figure 6**). Whereas the effect of the plasma index SNP on levels of sulfated bile acids is propagated to urine by filtration and therefore observed in both matrices, the study of urine revealed a regulatory variant in the 5' UTR of the gene, rs16961281, that was fine-mapped to a single-SNP credible set. It maps into accessible chromatin in the cortex of our kidney functional genomics data, and more specifically in proximal tubular cells (new **Supplementary Figure 12**), where the encoded transporter ASBT is expressed and responsible for the reabsorption of bile acids from urine. The minor allele is associated with higher *SLC10A2* expression in the kidney, lower levels of its substrate glycocholate in urine, as well as with lower risk of gallstone disease, independently of the plasma index SNP and supported by positive pairwise colocalizations. This regulatory variant leaves a urine-specific fingerprint on several bile acids that are known substrates of the encoded transport protein ASBT such as glycocholate (new **Figure 6**), whereas sulfated bile acids, the metabolites associated with the plasma index SNP, have been reported not to be transported by ASBT⁹. The urine-specific variant likely represents a more direct readout of ASBT-mediated transport in a matrix directly interacting with the transporter, whereas plasma levels may reflect secondary changes. We have added these observations to the manuscript (page 15, lines 362-403).

The genetic comparison between plasma and urine using all available mQTLs (rather than the matched analysis) is a helpful addition but also out of date. Please state more clearly how many of the 'novel' urine mQTLs (compared to the author's earlier work) are already captured by larger plasma studies, i.e. concordant?

Response: Following the same workflow as for plasma, we have updated the comparison of urine mQTLs (**Supplementary Figure 5e-5f**). Across all 622 urine mQTLs, 352 (56.6%) were not reported as associated with the corresponding metabolite in any of the seven evaluated, large studies at their defined level of significance (updated **Supplementary Table 6**). When restricted to the 399 urine mQTLs not reported in our earlier work, this was the case for 243 mQTLs (60.9%). Of note, even when a urine mQTL was captured by larger plasma studies, the direction of association was not always concordant. These discordant effects were especially observed at loci that encode for transporters and enzymes expressed at the apical membrane of tubular

epithelial cells in the kidney. For example, genetic associations with myo-inositol levels in urine and plasma show opposite directions, reflecting its active reabsorption from urine by the implicated transport protein SLC5A11. We have included this information in the Results section (page 7, lines 165-181), and replaced the term “novel” with “not previously reported in urine”.

I was struck by the fact that the correlation of genetic effects as well as validation rates were higher for the GCKD study than for larger, published GWAS of the circulating metabolome. Is this the case for the largest efforts? If yes, this is interesting, and the authors interpret this to demonstrate the value of a paired study design. However, it does beg the question how much of this this is driven by selected focus on CKD patients with reduced kidney function and not generalisable to a general population?

Response: Our updated analyses show that the correlation coefficient for the genetic effect sizes of urine mQTLs with matching plasma metabolite levels is still somewhat higher for GCKD (Spearman correlation coefficient 0.81) than for the larger, published mGWAS (range of coefficients: 0.19-0.77; median: 0.74). The validation rates depended on the significance level: while they were similar at a nominal significance level (0.82 in GCKD, 0.84 in Surendran, 0.84 in Chen, 0.75 in Hysi), they were lower at more stringent levels (e.g., at $p < 5E-08$: 0.64 in GCKD, 0.70 in Surendran, 0.77 in Chen, 0.59 in Hysi). In the study of Yin *et al*, which had a similar sample size as GCKD ($N=6,000$ vs. $N=5,000$ in GCKD) and hence should have similar or slightly better power, these validation rates were somewhat lower than in GCKD even at genome-wide significance (Spearman coefficient 0.59). We have added these observations to the manuscript (**Supplementary Figure 5; Supplementary Table 6**; page 7, lines 169-77). The Reviewer’s second question can only conclusively be answered once well-powered GWAS of paired plasma and urine metabolites from a population-based study become available, which we have now emphasized as an interesting direction for future research (page 22, lines 525-28).

Given these limitations, a study more focused on the comparative element would not only help to reduce the still very lengthy technical description of GWAS loci (in plasma), which is not a unique feature of this study, but allow focussing on the main stated rationale: to use the plasma-urine comparison in CKD patients to identify transporters and enzymes with

pharmaceutical relevance for CKD and more broadly. This part does not stand out clearly enough and is what might make this work competitive.

Response: We agree with the Reviewer and follow the advice as outlined above. We have performed the updated comparisons, quantified overlap with previous studies, and focus more on the comparative element of our study, for which we have added a new example. Details related to the GWAS loci in plasma have been moved to the Supplementary Results.

Another point of note is the total absence of any investigation of sex differences.

Response: We agree that this is interesting information and have evaluated sex differences following a published workflow¹⁰. After correction for multiple testing, we detected 37 significant ($p < 3.8E-05$) interactions with sex. The SNPs that showed the strongest differences by sex are consistent with existing knowledge: for example, variants at the *CPS1* locus showed a stronger effect on the levels of plasma glycine^{11,12} and also other associated plasma and urine metabolites in women compared to men. When an mQTL for a given metabolite was detected in both plasma and urine at the *CPS1* locus, most significant sex differences detected in plasma translated to urine. A stronger genetic effect of variants at the *SULT2A1* locus on the plasma levels of androgen metabolites in men as compared to women is consistent with the gene's function in catalyzing dehydroepiandrosterone sulfation in the adrenal cortex, and could be explained by higher levels of these metabolites in men compared to women. To the best of our knowledge, the significantly (p -interaction = $8E-11$) larger effect of the index SNP at the *SLC28A2* locus on urine adenosine levels in men as compared to women has not been reported previously. The encoded protein operates as a nucleoside transporter in the apical membrane of kidney epithelial cells where it mediates nucleoside reabsorption, including adenosine.¹³ GTEx data show higher median expression levels of *SLC28A2* in men as compared to women, which may explain the observed differences. We have included a new **Supplementary Table 7** that summarizes the interaction analyses between our index SNPs and sex, and added the new findings to the Results (page 8, lines 183-7), Supplementary Results (Supplementary Information page 3f), and Methods sections (page 44, lines 1022-5).

A strength of the revised version is that replication is performed, although it would be more useful had this been extended to include an assessment of the generalisability of findings to ethnic groups represented in the ARIC replication cohort to compensate at least partly for the fact that this is yet another White European focussed effort.

Response: We agree and have now extended the replication effort to the African American (AA) participants of the ARIC Study. We find that genetic effects on metabolite levels show very good correlations with the discovery estimates in GCKD (Pearson correlation coefficient 0.86, as compared to 0.98 for ARIC EA; expanded **Supplementary Figure 2**, also shown below). These observations support that many findings from a discovery population of European ancestry and with reduced kidney function are generalizable to an African American sample from the general population. However, the sample size of ARIC AA participants with available metabolite measurements was substantially smaller (max. N=818) than for the ARIC EA sample (N=3,603), and fewer mQTLs pass the multiple-testing corrected significance threshold (124 mQTLs, $p < 0.05/430$ and direction consistent). We have added the new replication data to the manuscript, so that readers can compare the genetic effect size correlations between the GCKD discovery and the ARIC EA or ARIC AA replication samples, as well as explore effect estimates, allele frequencies and sample sizes for the individual SNPs in each of the three samples (**Supplementary Table 4**). The manuscript has been expanded in the Results (page 6, lines 141-8) and the Methods section (page 39, lines 905-7 and page 44, line 1020f).

Supplementary Figure 2: Replication of plasma mQTLs from GCKD in the ARIC EA (left) and AA (right) sample.

Reviewer #2:

The authors have provided a very strong and detailed rebuttal to my comments and the revised version of the manuscript, and the Supplement look excellent. There are only a few remaining minor points worth addressing (as listed below).

Response: Thank you for the positive assessment of our revised manuscript.

1. Thank you for conducting an additional sensitivity analysis and providing a rationale for assigning equal weight to all tissues in the algorithm used to nominate the most likely gene drivers. I think that inclusion of Table S1 cataloguing these genes is most useful. While I am satisfied that these additional analyses and results make a very persuasive case for not making any further changes to the algorithm, I do think that the discussion requires a little bit more contemplation of its limitations. Indeed, the algorithm (i) identifies the highest scoring gene, but it does not specify the tissue through which the genes is likely to influence the metabolites (ii) it does not provide equal weights to protein-coding and non-coding (e.g. long non-coding) genes in scoring which may explain why the latter are not amongst the prioritised genes. This is important because (i) a proportion of GWAS signals operate through changes in expression of different genes in different tissues (please see CKD-defining traits as a relevant example - Nat Commun. 2018;9:4800.) and (ii) long non-coding RNAs are recognised gene mediators of associations signals with complex human traits. It would be most helpful to see this being covered in the discussion with appropriate references.

Response: Thank you for the concrete and helpful suggestions. Regarding point i), **Supplementary Table 11**, columns L and N show in which tissues an eQTL was identified or in which tissue a positive colocalization with gene expression was detected, respectively. We agree with the Reviewer, however, that this information is not unambiguous: one the one hand, more than one tissue and gene can be implicated and on the other hand, not all tissues may be equally well represented. We have therefore added these observations to the limitations section of the discussion (pages 23f, lines 555-62), where we have also addressed the second point about protein coding and non-coding genes. We have included appropriate references, including the one above.

2. Can you please clarify for what percentage of the mQTLs you could not identify the most likely gene drivers?

Response: Our implemented workflow always assigns a most likely gene. Genes in the locus were evaluated and scored based on multiple sources of evidence as outlined in the Methods. In case there were several genes receiving the highest score in a locus, such ties were resolved by distance and the closest gene assigned as the most likely causal one. The most likely causal gene was assigned by distance for 17% of the mQTLs (221/1299). We have now added this information to the manuscript (page 47, lines 1098f). The number and sources of evidence that support each gene in each associated locus are shown for interested readers in **Supplementary Table 11**, column H.

3. I think that the title of the Results section “Systemic roles of renal dipeptidase 1 /DPEP1/: circulating enzymes, hypertension and arthropaties” needs changing to “Systemic roles of dipeptidase 1 /DPEP1/: circulating enzymes, hypertension and arthropaties”. Indeed, DPEP1 encodes dipeptidase 1 also called Microsomal Dipeptidase or Dipeptidase 1 (Renal). In the current form the titles suggests that it is the kidney expression of this gene that determines the levels of digestive enzymes, hypertension and arthropaties while of course based on the provided data one cannot assign a specific tissue that is responsible for these findings. As the authors point out, DPEP1 is expressed not only in the kidney.

Response: Indeed, we included “renal” as part of the enzyme’s name. We agree with the Reviewer’s comment and have now implemented the suggested changes.

References:

1. Surendran, P. *et al.* Rare and common genetic determinants of metabolic individuality and their effects on human health. *Nat Med* **28**, 2321-2332 (2022).
2. Chen, Y. *et al.* Genomic atlas of the plasma metabolome prioritizes metabolites implicated in human diseases. *Nat Genet* (2023).
3. Hysi, P.G. *et al.* Metabolome Genome-Wide Association Study Identifies 74 Novel Genomic Regions Influencing Plasma Metabolites Levels. *Metabolites* **12**(2022).
4. Konig, E. *et al.* Whole Exome Sequencing Enhanced Imputation Identifies 85 Metabolite Associations in the Alpine CHRIS Cohort. *Metabolites* **12**(2022).

5. Bomba, L. *et al.* Whole-exome sequencing identifies rare genetic variants associated with human plasma metabolites. *Am J Hum Genet* **109**, 1038-1054 (2022).
6. Feofanova, E.V. *et al.* A Genome-wide Association Study Discovers 46 Loci of the Human Metabolome in the Hispanic Community Health Study/Study of Latinos. *Am J Hum Genet* **107**, 849-863 (2020).
7. Yousri, N.A. *et al.* Whole-exome sequencing identifies common and rare variant metabolic QTLs in a Middle Eastern population. *Nat Commun* **9**, 333 (2018).
8. Li-Gao, R. *et al.* Genetic Studies of Metabolomics Change After a Liquid Meal Illuminate Novel Pathways for Glucose and Lipid Metabolism. *Diabetes* **70**, 2932-2946 (2021).
9. St-Pierre, M.V., Kullak-Ublick, G.A., Hagenbuch, B. & Meier, P.J. Transport of bile acids in hepatic and non-hepatic tissues. *J Exp Biol* **204**, 1673-86 (2001).
10. Pietzner, M. *et al.* Mapping the proteo-genomic convergence of human diseases. *Science* **374**, eabj1541 (2021).
11. Mittelstrass, K. *et al.* Discovery of sexual dimorphisms in metabolic and genetic biomarkers. *PLoS Genet* **7**, e1002215 (2011).
12. Hartiala, J.A. *et al.* Genome-wide association study and targeted metabolomics identifies sex-specific association of CPS1 with coronary artery disease. *Nat Commun* **7**, 10558 (2016).
13. Elwi, A.N. *et al.* Renal nucleoside transporters: physiological and clinical implications. *Biochem Cell Biol* **84**, 844-58 (2006).

Decision Letter, second revision:

9th Mar 2023

Dear Dr. Köttgen,

Thank you for submitting your revised manuscript "Genetic studies of paired metabolomes reveal enzymatic and transport processes at the interface of plasma and urine" (NG-A60156R1). It has now been seen by the original referees and their comments are below. The reviewers find that the paper has improved in revision, and therefore we'll be happy in principle to publish it in Nature Genetics, pending minor revisions to comply with our editorial and formatting guidelines.

Sincerely,
Wei

Wei Li, PhD
Senior Editor
Nature Genetics
New York, NY 10004, USA
www.nature.com/ng

Reviewer #1 (Remarks to the Author):

Thank you for this revised version, I appreciate the effort the authors have made to update their results to reflect recent publications (Chen et al) and have no further comments.

Reviewer #2 (Remarks to the Author):

I am satisfied with this revision.

Final Decision Letter:

26th Apr 2023

Dear Dr. Köttgen,

I am delighted to say that your manuscript "Genetic studies of paired metabolomes reveal enzymatic and transport processes at the interface of plasma and urine" has been accepted for publication in an upcoming issue of Nature Genetics.

Your paper will be published online after we receive your corrections and will appear in print in the next available issue. You can find out your date of online publication by contacting the Nature Press Office (press@nature.com) after sending your e-proof corrections. Now is the time to inform your Public Relations or Press Office about your paper, as they might be interested in promoting its publication. This will allow them time to prepare an accurate and satisfactory press release. Include your manuscript tracking number (NG-A60156R2) and the name of the journal, which they will need when they contact our Press Office.

Please note that *Nature Genetics* is a Transformative Journal (TJ). Authors may publish their research with us through the traditional subscription access route or make their paper immediately open access through payment of an article-processing charge (APC). Authors will not be required to make a final decision about access to their article until it has been accepted. [Find out more about Transformative Journals](https://www.springernature.com/gp/open-research/transformative-journals)

Authors may need to take specific actions to achieve [compliance with funder and institutional open access mandates](https://www.springernature.com/gp/open-research/funding/policy-compliance-faqs). If your research is supported by a funder that requires immediate open access (e.g. according to [Plan S principles](https://www.springernature.com/gp/open-research/plan-s-compliance)) then you should select the gold OA route, and we will direct you to the compliant route where possible. For authors selecting the subscription publication route, the journal's standard licensing terms will need to be accepted, including [self-archiving-and-license-to-publish](https://www.nature.com/nature-portfolio/editorial-policies/self-archiving-and-license-to-publish). Those licensing terms will supersede any other terms that the author or any third party may assert apply to any version of the manuscript.

Please note that Nature Portfolio offers an immediate open access option only for papers that were first submitted after 1 January, 2021.

If you have not already done so, we invite you to upload the step-by-step protocols used in this manuscript to the Protocols Exchange, part of our on-line web resource, natureprotocols.com. If you complete the upload by the time you receive your manuscript proofs, we can insert links in your article that lead directly to the protocol details. Your protocol will be made freely available upon publication of your paper. By participating in natureprotocols.com, you are enabling researchers to more readily reproduce or adapt the methodology you use. [Natureprotocols.com](http://natureprotocols.com) is fully searchable, providing your protocols and paper with increased utility and visibility. Please submit your protocol to <https://protocolexchange.researchsquare.com/>. After entering your [nature.com](http://www.nature.com) username and password you will need to enter your manuscript number (NG-A60156R2). Further information can be found at <https://www.nature.com/nature-portfolio/editorial-policies/reporting-standards#protocols>

Sincerely,
Wei

Wei Li, PhD
Senior Editor
Nature Genetics
New York, NY 10004, USA
www.nature.com/ng